

# LACK OF MARINE ENTRY INTO MARMARA AND BLACK SEA-LAKES INDICATE LOW RELATIVE SEA LEVEL DURING MIS 3 IN THE NORTHEASTERN MEDITERRANEAN

Anastasia G. Yanchilina[1], Celine Grall[2], William B.F. Ryan[2], Jerry F. McManus[2], Candace O. Major[3]

[1]Department of Earth and Planetary Sciences, Weizmann Institute of Science, Rehovot, Israel 7610001.
[2]Lamont-Doherty Earth Observatory, Columbia University, 61 Route 9W, Palisades, New York 10964.
[3]National Science Foundation, 2415 Eisenhower Ave., Alexandria, Virginia 22314

Correspondence: Anastasia Yanchilina (anastasia.yanchilina@weizmann.ac.il)

## Abstract

The Marine Isotope Stage 3 (MIS 3) is considered a period of persistent and rapid climate and sea level variabilities during which eustatic sea level is observed to have varied by tens of meters. Constraints on local sea level during this time are critical for further estimates of these variabilities. We here present constraints on relative sea level in the Marmara and Black Sea regions in the northeastern Mediterranean, inferred from reconstructions of the history of the connections and disconnections (partial or total) of these seas together with the global ocean. We use a set of independent data from seismic imaging and core-analyses to infer that the Marmara and Black Seas remained connected persistent freshwater lakes that outflowed to the global ocean during the majority of MIS 3. Marine water intrusion during the early MIS-3 stage may have occurred into the Marmara Sea-Lake but not the Black Sea-Lake. This suggests that the relative sea level was near the paleo-elevation of the Bosporus sill and possibly slightly above the Dardanelles paleo-elevation, ~80 mbsl. The Eustatic sea level may have been even lower, considering the isostatic effects of the Eurasian ice sheet would have locally uplifted the topography of the northeastern Mediterrranean.

## 1. Introduction

Marine isotope stage 3 (MIS 3) is identified as the period between 60 and 25 kyr B.P., when regional and global climate fluctuated over a broad range of temperatures on millennial time scales (Dansgaard et al. 1993, Members 2006). One particularly noteworthy and intriguing aspect of MIS 3 is its characteristic sequence of abrupt climate fluctuations including the iconic Dansgaard-Oeschger (D-O) bi-polar oscillations and Heinrich event iceberg discharges with corresponding fluctuations in global sea level on the order of ~20-40 m (Siddall et al. 2008). The eustatic sea level (ESL) during this period remains uncertain, with sea level elevations that range as low as 87 meters below today's sea level (mbsl) to as high as 25 mbsl (Siddall et al. 2008). The lack of existence of a rigorous constraint on ESL has implications



for understanding factors that control ice-sheet growth and collapse. Ice volume variations are
additionally an important input into glacial isostatic adjustment (GIA) models (Pico et al.
2016), that are in turn used to understand changes in the distribution of ice volume on the planet
and its effect on local sea level. It should be noted that ESL is different from Relative Sea Level
(RSL) as the RSL represents the elevation of the sea relative to the moving solid earth.  The
earth is moving notably because of isostatic adjustment of the earth's surface to the changes in
the distribution of ice and water and the corresponding changes of the gravitational potential
of the earth-ocean system (Lambeck et al. 2002b).  Also, the RSL can be obtained from
geological archives and ESL may be derived from these archives when the earth solid surface
motions are properly considered.

Several methodologies have been employed to deduce the height of past ESL: 1) ice

volume changes using oxygen isotope measurements of planktonic and benthic foraminifera
(Bintanja et al. 2005, Shakun et al. 2015) (Fig. 1a and b), 2) elevated coral terraces (Cutler et
al. 2003) (Fig. 1c), and 4) changes between brackish and fresh conditions in marginal basins
(Van Daele et al. 2011, Pico et al. 2016, Pico et al. 2017).  Reconstructing ESL from the benthic
foraminiferal LR04 $\delta^{18}O$ record using an inverse ice-sheet climate model gives sea level
estimates below 80 mbsl (Fig. 1a) (Bintanja et al. 2005).  Isolating the ice volume from a
compilation of planktonic foraminiferal $\delta^{18}O$ records gives a global sea level that decreases
from ~60 mbsl to 80 mbsl over the course of MIS 3 (Fig. 1b) (Shakun et al. 2015).  U/Th dated
coral terraces used as a relative sea level proxy (Lambeck et al. 2002b), suggest after
considering a GIA correction, that the ESL was below 60 mbsl (Fig. 1c) (Yokoyama et al.
2001, Peltier et al. 2006).  More recent studies suggest shallower estimates of ESL during MIS
3.  Sediment cores taken from the Yellow River Delta that similarly record fluctuations
between fresh and marine environments have been used to assert that ESL reached a peak of
38 mbsl during an interval between 50 and 37 kyr (Pico et al. 2016).  Similarly, records from
the Albermarle Embayment on the U.S. Mid-Atlantic coast show a ESL peak level of 40 mbsl
during MIS 3 (Pico et al. 2017).  Geological archives provide RSL indices.  RSL indices are
critical to constrain the local ESL and Global Mean Sea Level models.

In this paper, we provide highstand thresholds on RSL in the Black Sea and Marmara

Sea system to infer information about ESL during MIS 3 using a GIA model and provide
valuable geological constraints on regional RSL reconstructions.  In the modern configuration,
Black Sea and Marmara Sea are connected to the global ocean via shallow Bosporus and
Dardanelles Straits to form an interconnected lake-ocean system (Çagatay 2003).  During



glacial periods, this configuration can change if the RSL falls below the straits, cutting off
marine water entry.  Here we use observations from geochemical records and seismic images
to reconstruct water connection and disconnection histories between the Black Sea, Marmara
Sea, and the global ocean.  We show that the two lakes were freshwater, connected, and
outflowing excess freshwater to the East Mediterranean for a large fraction of MIS 3, with the
exception of a possible transient marine entry from 55 kyr B.P. to 44 kyr B.P.  We then provide
different independent estimates on paleo-lake elevations and sill elevations to provide a
database of RSL estimates and discuss the possible implication on the local ESL.  In this paper,
we will refer to the Sea of Marmara and the Black Sea as Marmara Sea-Lake and Black Sea-
Lake, taking into account their prior freshwater state.

**2. Inferred paleo-salinity in the Marmara Sea-Lake and Black Sea-Lake during MIS 3**
**from previous studies**

Previous studies have documented the $CaCO_3$ accumulation in the Marmara Sea

(Çağatay et al. 2015) and Ca (%) accumulation in the Black Sea (Nowaczyk et al. 2012),
porewater $Cl^-$ in the Marmara Sea (Aloisi et al. 2015) and Black Sea (Soulet et al. 2010), and
last, $\delta^{18}O$ composition of the Black Sea mollusk record (Major et al. 2006, Yanchilina et al.
2017) and Marmara Sea mollusk record (Vidal et al. 2010).

$CaCO_3$, although a proxy that cannot be used as a direct measurement of paleosalinity,

it can be used to interpret the connectivity between water bodies.  $CaCO_3$ in lakes reflects the
integration between sedimentation rate and $CO_3$-assimilation and pH-changes induced by
phytoplankton blooms.  Higher $CaCO_3$ becomes deposited during warmer periods when there
are more phytoplankton blooms relative to colder periods.  Specific to our case, the Sea of
Marmara is small in volume relative to the volume of the Black Sea and also, does not have a
significant amount of independent river inflow.  Hence, the temporal synchronicity of $CaCO_3$
variability between both basins is inferred to reflect connectivity of the two lakes and to
indicate outflows of the Black Sea-Lake into the Marmara Sea-Lake.  Although fresh, both
lakes are likely to have been somewhat alkaline in order to account for episodes of organic and
inorganic carbonate accumulation at times of rapid warming during each of the
Dansgaard/Oeschger events (Çağatay et al. 2015).

Porewater $Cl^-$ measurements were ~50 mmol/L in the Black Sea-Lake and 200-500

mmol/L in the Marmara Sea-Lake during MIS 3 before their connection to the global ocean at
9.3 kyr B.P. (Yanchilina et al. 2017) and 14 kyr B.P. (Aloisi et al. 2015), respectively.





Application of an advection/diffusion model allows to qualitatively deconstruct paleosalinity.
The modern values for the Sea of Marmara and the Black Sea are 620 mmol/L and 350 mmol/L
with the former corresponding to a salinity value of ~39 ppt (Soulet et al. 2010, Aloisi et al.
2015).  Freshwater bodies typically have low porewater Cl⁻ composition (i.e., <~ 50 mmol/L).
A decrease of porewater Cl⁻ towards these values would indicate that these bodies of water
were fresher in the past. After taking into account advection and diffusion, porewater Cl⁻ values
indicate both of the seas were fresh during MIS 3 and possibly also the late part of MIS 4
(Soulet et al. 2010, Aloisi et al. 2015).
The $\delta^{18}O$ composition of the mollusks from the Black Sea is measured to be -6±1 ‰
(Major et al. 2006, Yanchilina et al. 2017) and the $\delta^{18}O$ composition of the Sofular Cave
stalagmites is measured to be -12±1 ‰ during the MIS 3 (Fleitmann et al. 2009, Badertscher
et al. 2011).  The resolution of the U/Th ages for the Sofular Cave stalagmites for MIS 3 is, on
average, 24 years (Fleitmann et al. 2009).  $\delta^{18}O$ of -6 ‰ measured in the Black Sea mollusks
is the value that reflects the $\delta^{18}O$ composition of the mollusks during the last glacial period,
MIS 2, when the Black Sea was shown to be a freshwater lake.  The $\delta^{18}O$ of -12 ‰ of the
Sofular Cave stalagmites is shown to reflect the evaporation of Black Sea water with a constant
offset of -6 ‰, the difference argued to account for fractionation of water as a consequence of
distillation effects (Fleitmann et al. 2009, Badertscher et al. 2011).  Given the observation that
the $\delta^{18}O$ of the Sofular Cave stalagmites remains at -12 ‰ all throughout MIS 3 indicates the
Black Sea was also fresh for the entirety of this period. Furthermore, freshwater mollusk
*Dreissena rostriformis*, and dinoflagellates *S. cruciformis* and *P. psilata* persistently dominate
the MIS 3 faunal composition in the Black Sea-Lake (Rochon et al. 2002, Yanchilina et al.

2017).

We will further evaluate these observations and prior interpretations from comparison
of variations in $^{87}Sr/^{86}Sr$ measured in ostracod and mollusk shells from both Black Sea and
Marmara Sea basins.  Contrary to the $\delta^{18}O$ composition of water, a variable that incorporates
changes in the hydrological cycle of lake systems, $^{87}Sr/^{86}Sr$ of water responds exclusively to
changes in source of water and can differentiate between changes in input from the different
sources. $^{87}Sr/^{86}Sr$ is especially relevant to measure when making an attempt to identify inputs
of saline water into lake systems. Because of the low concentration of freshwater Sr from rivers,
even a small input of marine water, rich in Sr, will make an observable change in the $^{87}Sr/^{86}Sr$
composition of freshwater bodies of water.  $^{87}Sr/^{86}Sr$ is a very sensitive proxy used previously
to reconstruct the deglacial entry of marine water into the Black Sea-Lake during the Holocene





(Major et al. 2006, Yanchilina et al. 2017). The modern $^{87}Sr/^{86}Sr$ composition of river water
inflowing into the Black Sea is ~0.7088 (Palmer et al. 1989) whereas the $^{87}Sr/^{86}Sr$ composition
of the ocean water is 0.709155 (Henderson et al. 1994).

**3. Materials and Methods**
3.1 Geochemistry

We measured $^{87/86}Sr$ of ostracodes and mollusks from the Marmara Sea sediments and

compared these measurements with the published values from the Black Sea (Major et al. 2006,
Yanchilina et al. 2017).  The sediments were taken from cores ITU-C1 at 73 mbsl, MD01-2426
at 250 mbsl, ITU-C10 at 364 mbsl, and MD01-2430 at 580 mbsl (Fig. 2).  MD01-2426 and
MD01-2430 were retrieved in 2001 during the MD123/MARMACORE cruise (R/V Marion
Dufresne).  MD01-2426 was collected from north of the Imrali ridge and MD01-2430 was
collected on the Western High between the Central and the Tekirdag deep basins (Grall et al.
2013).  ITU-C1 and –C10 were retrieved in 2002 with R/V MTA Sismik 1 in and around the
Sarkoy Canyon in the Western Marmara Sea. The $^{87}Sr/^{86}Sr$ records for the Sea of Marmara
were measured at Lamont-Doherty Earth Observatory, Columbia University.  Sr was initially
leached to retrieve the Sr fraction (Bailey et al. 2000) which was then loaded upon tungsten
filaments with TaCl5 (Birck 1986).  $^{87}Sr/^{86}Sr$ ratios were measured using a dynamic multi-
collector on a VG thermal ionization mass spectrometer and normalized to $^{86}Sr/^{88}Sr = 0.1194$
to correct for mass fractionation.  Beam size was tuned to be close to 5.0 x 10-11 for $^{88}Sr$.
$^{87}Sr/^{86}Sr$ measurements were monitored to account for instrumental drift through periodically
running NBS987 which gave $^{87}Sr/^{86}Sr$ = 0.710288 (±0.000015) with a 2σ external
reproducibility, n = 16. The original age model was constructed from $^{14}C$ measurements and
calibrated to calendar age with a zero reservoir correction.  Although the original $^{14}C$
measurements have been misplaced, we compose our own age model from $^{14}C$ measurements
made from pieces of mollusks from MD01-2430 (Vidal et al. 2010) which we correct for
reservoir after tuning its $\delta^{18}O$ record  to that of the Black Sea and Sofular Cave $\delta^{18}O$ records.
This procedure follows from the observation that the $\delta^{18}O$ of the Sofular Cave $\delta^{18}O$ record
reflects the $\delta^{18}O$ composition of Black Sea surface water (Fleitmann et al. 2009, Badertscher
et al. 2011) which, before the connection of the Marmara Sea-Lake with the Mediterranean Sea
reflected predominantly the $\delta^{18}O$ composition of the Black Sea-Lake that flowed through the
Bosporus Strait into the Sea of Marmara.  The results are presented in Supplementary Materials
1 and illustrated in Fig. 3 a.




3.2 Chirp profiles

We present a chirp record of a perched pond/lake, lake Gemlik (Figs. 2, 4), from Sea

of Marmara in order to diagnose whether any marine entry occurred during MIS 3.  The chirp
profile was acquired during Sensing the Ocean with Marine Radars (SoMAR) cruise on the
R/V *K. Piri Reis* in 2013 with SyQwest-Bathy 2010 chirp profiler and operating frequency of
3.5 KHz.  This perched lake is observed on the southern shelf of the Marmara Sea (Fig. 2).  It
is separated from the deeper sections of the Marmara Sea by the Imrali ridge with a depth of
50 mbsl. Lake Bandirma is also indicated (Fig. 2) observed to lie to the east of Lake Gimlik
but a chirp profile for was not acquired.

**4. Results**

4.1 Paleosalinity and paleo-connectivity of Marmara and Black Sea during MIS 3

$^{87}Sr/^{86}Sr$ measurements of the Marmara Sea mollusks at the beginning of MIS 2 vary

around 0.7088 (Figure 3a), a value that is similar to the $^{87}Sr/^{86}Sr$ composition of the Black Sea
during this period and also to the average $^{87}Sr/^{86}Sr$ composition of river water flowing into the
Black Sea of 0.7088 (Palmer and Edmond 1989).  $^{87}Sr/^{86}Sr$ composition of Black Sea mollusks
also has a strictly lacustrine composition of 0.70880±5E-5 at the end of MIS 3 (Fig. 3a). The
$^{87}Sr/^{86}Sr$ of freshwater mollusks from the Marmara Sea-Lake follows closely that of the Black
Sea record through the deglaciation and is also lacustrine at the beginning of MIS 2.  For the
two lakes to share the identical $^{87}Sr/^{86}Sr$ composition similar to the composition of river inflow
into the Black Sea, the Black Sea must have been fresh and outflowing to the Sea of Marmara.
The Sea of Marmara must have subsequently outflowed to the Mediterranean Sea, as it's a
much smaller volume relative to the Black Sea.  We present supporting data to contest that the
two lakes were freshwater, connected, and outflowing excess freshwater to the East
Mediterranean for a large fraction of MIS 3.

Comparing together sediment Ca and $CaCO_3$ (Nowaczyk et al. 2012, Çağatay et al.

2015) (Fig. 3b) with the porewater Cl⁻ (Soulet et al. 2010, Aloisi et al. 2015) (Fig. 3c) from the
Black and Marmara Seas and Sofular Cave $\delta^{18}O$ of the stalagmites (Fleitmann et al. 2009,
Badertscher et al. 2011) (Fig. 3d) support this interpretation. The Ca and $CaCO_3$ of the Black
Sea-Lake and Marmara Sea-Lake are identical for all of MIS 3 with the exception of the period
from 55 to 44 kyr B.P., in which either the Black Sea-Lake suspended outflow to the Marmara
Sea-Lake and/or there was a potential marine entry into the Marmara Sea-Lake but not the





Black Sea-Lake. The latter is less likely to have occurred as every marine entry recorded into
both the Marmara and Black Sea-Lakes is followed by a formation of a sapropel, evidence of
which here is not observed. We still consider this possibility in case the transient marine entry
was minor and perhaps was just a small inflow.
Porewater Cl⁻ in Marmara Sea-Lake sediments differs from that of the Black Sea-Lake
sediments as a result of the earlier connection of the Marmara Sea with the global ocean and
higher post-connection salinity, leading to an earlier diffusion of marine water into the
previously lacustrine sediments. If there had been any marine inflow into either sea during
MIS 3, there would be observable remnant diffusion trends. In fact, Cl- decreases back through
time and into MIS 4 to 100 mmol/L, suggesting the Marmara Sea-Lake was also fresh during
this period and the freshening had to have happened even earlier. There is only one point
during which the Cl- is observed to increase, 50 kyr B.P., and we discuss this in this paper,
with the light of supplemental Sr analyses. Pore-water chloride, $\delta^{18}$O and $^{87}$Sr/$^{86}$Sr of mollusks
and ostracods do not indicate any significant rises in salinity in the Marmara and Black Sea-
Lakes.
$\delta^{18}$O composition of the Sofular Cave stalagmites, porewater Cl- and sediment $CaCO_3$
results support this interpretation. The $\delta^{18}$O of Black Sea-Lake and Marmara Sea-Lake
carbonate reflects the composite of hydrological balance in the basin through integration of
inputs in the form of river and rain water and outputs in the form of evaporative processes
(Major et al. 2006). A lower $\delta^{18}$O value is considered to be fresh and in a positive hydrological
framework whereas a higher $\delta^{18}$O value is considered to reflect either entrance of marine water
and/or a drier climate, preferentially removing the lighter oxygen isotope (i.e., $^{16}$O) from the
water (Deuser 1972). We use the $\delta^{18}$O composition of the dated Sofular Cave stalagmites with
a temporal resolution of $\delta^{18}$O measurements of ~24 years to infer the paleosalinity further back
in time, a measurement previously shown to reflect the composition of the Black Sea water
vapor (Badertscher et al. 2011). The most recent entry of marine water resulted in a rapid
increase in salinity to modern values in both the Black Sea-Lake (Major et al. 2006; Yanchilina
et al. 2017) and in the Marmara Sea-Lake (Sperling et al. 2003). This is not observed on the
$\delta^{18}$O records, suggesting both basins remain fresh.

4.2 MIS 3 Relative Sea Level Index in Marmara Sea-Lake and Black Sea-Lake.
Wave-cut cliffs and their corresponding terraces are observed everywhere in the
subsurface of the outermost Marmara Sea continental shelf at elevations close to the modern



Dardanelles bedrock sill at ~65 mbsl (Gokasan et al. 2008) (Supplementary Material 2-6, Table
1). At the distal edge of each terrace, one observes inclined clinoforms (Çağatay et al. 2009)
indicative of subaqueous prodelta foresets that are truncated by an erosion surface (Vardar et
al. 2014). Where sampled, the youngest foresets of these clinoforms are of early MIS-2 age
(Yaltirak et al. 2002, Ergin et al. 2007, Çağatay et al. 2009, Karakilcik et al. 2014, Vardar et
al. 2014). The mollusk assemblage, composed of exclusively freshwater species (i.e.,
*Dreissena r.*) in core MD04-2745 that sampled the entire succession of incline strata, indicates
there was no observable entry of marine water during MIS 4, 3, and 2. Older MIS 5 deposits
outcrop at elevations up to 7 m above today's sea level on the edge of the Dardanelles Strait
(Supplementary Material 6, Table 1) (Yaltirak et al. 2002).

**Table 1**
________________________________________________________________________

| Figure name | Source | Key observations |
|---|---|---|
| SU 2 | Gökaşan et al. (2010) | MIS 2-3-4 foresets in the Marmara Sea-Lake, seaward of Prince Islands, below 80 mbsl |
| SU 3 | Karakilcik et al. (2014) | MIS 2-3-4 foresets in the Marmara Sea-Lake, Çekmeke shelf break, lower than 90 mbsl |
| SU 4 | Ergin et al. (2007) | MIS 2-3-4 foresets, northwest margin of Marmara Sea, lower than 75 mbsl. The youngest foreset is [14]C dated to MIS 2. |
| SU 5 | Smith et al. (1995) | MIS 2-3-4 foresets, west of Imrali Island in the Marmara Sea, lower than 70 mbsl. |
| SU 6 | Gökaşan et al. (2010) | Paleo-elevation of the Dardanelles strait is inferred to be 85 mbsl. |


In the southern shelf of the Marmara Sea, both the Gemlik and Bandirma lakes are
interpreted to have been separated from the large main Marmara Sea-Lake by the Imrali ridge
during MIS 2, 3, and 4 (Vardar et al. 2014) (Fig. 4). Chirp records show no evidence at all of
MIS 3 and 2 sediments on the shelves except in these perched ponds. The paleo-shorelines of
the Bandirma and Gemlik suspended lakes are observed at ~50 mbsl and ~60 mbsl, respectively
(Vardar et al. 2014). Thin transparent layers of Holocene age are observed on the Imrali Ridge,
lying along an erosional unconformity (C-a) which has been interpreted as the last marine
lacustrine to marine transition. A second deeper unconformity is observed below, on the
intervening shelf region as well beneath the floor of the ponded lakes (Fig. 4). The age of the
deeper unconformity is unconstrained, due to the lack of recovered sediment. It has been
proposed that this unconformity may be 23 kyr or 30 kyr (Hiscott et al. 2002, Vardar et al.
2014). This observation, however, is not compatible with the stratigraphic history of the





deposits.  It is shown that the MIS-2 period corresponds to a large transgressive period
following the MIS-3 low stand (Çağatay et al. 2009). This suggests that the basal surface of
the sediment deposited during MIS-2 should be conformable with sediment below. It is likely
that this erosional surface is a consequence of the brief drying event related to the beginning
of the Bolling/Allerod, immediately before entry of marine water into the Marmara Sea-Lake
(McHugh et al. 2008). The bumps and valleys in Unit 2 are potentially artifacts of gas derived
from below.  The loss of water associated with the brief desiccation is likely to have initiated
this discharge. This phenomenon came to an end after the later loading of marine water.  Core
data would be able to strengthen this interpretation but is at the moment unavailable.  Hence,
no deposition during MIS 3-2 is observed in this lake and lake level had to be below the depth
of the Imrali ridge during this period.
Chirp sub-bottom profiles across the outer shelf of the western Black Sea reveal a
succession of superimposed lacustrine deposits (Fig. 5) belonging to basinward prograding
lowstand deltas (Aksu et al. 2002, Dimitrov 2010).  On top of the youngest clinoforms are sand
dunes and a berm like feature indicative of a paleoshoreface (Lericolais et al. 2009, Yanchilina
2016, Yanchilina et al. 2017).  Where sampled, the youngest set of prograding strata are $^{14}$C
dated to late MIS 3 through MIS 2 and contain lacustrine fauna (Yanchilina et al. 2017).  The
succession of parallel and prograding deposits indicates alternating highstands and lowstands.
If one were to consider that these clinoforms were deposited in a near-shore subaqueous pro-
delta environment, they indicate the surface of the Black Sea-Lake was 80 to 90 mbsl during
MIS 3.
**5. Discussion**
Geochemical and geophysical data suggest that Black Sea and Marmara Sea remained
fresh and the hydrological budget was positive during most of MIS 3, with the exception of a
period between 44 and 55 kyr B.P.  The hydrologic budget must have remained positive in
order for (1) the perched ponds not to have dried out and (2) to account for the similarity in the
elevation of the surface of the Marmara and Black Sea-Lakes and their correspondence  with
elevations of the Dardanelles bedrock, to maintain outflow as suggested by geochemical data
(i.e., identical $CaCO_3$ during MIS 3 and 2, low $\delta^{18}$O of Sofular Cave stalagmites, and similar
$^{87}$Sr/$^{86}$Sr during end of MIS 3/ 2).
The similarity between the present average elevation of the lake's surface and the
elevation of the Dardanelles and Bosporus Sills at ~65-80 mbsl confirms that the straits acted
as outflow channels (i.e., a rivers) expelling Black and Marmara Sea-Lake freshwater to the



external ocean during MIS 3 and MIS 2. This also suggests that the Dardanelles bedrock sill
was at or near (by a few meters) the relative sea level.  It remains possible that RSL is below
by a few meters the Dardanelle paleo-sill, if freshwater flux would have been able to sustain a
certain pressure to deny any marine inflow (Dalziel 1991; Lane-Serff et al. 1997; Lambeck et
al. 2007). The observation of a lower lake surface that indicates that the Bosporus sill also must
have been deeper than today, sitting at its bedrock at 80 mbsl.  If the outlet sill was shallower,
then the elevation of the foresets would likely have been higher. Substantial river entrenchment
may have deepened its modern bedrock depth. For example, MIS 6, 8, 10, and 12 lowstand
clinoforms in the region of Prince Islands (Supplementary Materials 2) are all at elevations as
low or lower than the MIS 2 and 3 clinoforms. Thus the lake's ancient shorelines must have
always had to have fallen to an elevation near to the modern bedrock elevation during each
lacustrine period.
This set of independent geological archives provide RSL index with present elevation
ranging between 70 and 90 mbsl.  The present elevations of indexes are likely different than
their paleo-elevations during MIS 3.  The present elevations of indexes are likely different than
their paleo-elevations during MIS 3.  10-20 m of regional subsidence has likely occurred since
MIS 3, as the results of GIA associated with the overall ice-water budget in between the glacial
MIS 3 time and today.  This may place the RSL during MIS 3 at a maximum value of 80 m.
During MIS 3, RSL was shallower than the ESL here, in response to the GIA associated with
the transition from MIS 5 interglacial stage into MIS 3.  This suggests that the ESL in the
region was likely below 70 mbsl during the overall MIS 3 glacial stage.  This threshold on ESL
is in agreement with the results from U/Th dated coral terraces corrected for GIA (Yokoyama
et al. 2001) that  leads to coral-based global sea level reconstructions about ~58-111 mbsl (~45
to -110 mbsl if error is included) (Fig. 1e) (Yokoyama et al. 2001).  Four of these coral-based
ESL reconstructions during MIS 3 give a range of 58 to 70 mbsl and eight in the range of 70
to 80 mbsl.  Our results on ESL supports the lower estimates obtained from scaling changes
from the LR04 stack of benthic $\delta^{18}O$ records (Bintanja et al. 2005), but does not entirely agree
with changes in ESL from scaling changes from detrended planktonic $\delta^{18}O$ records corrected
for changes in temperature (Shakun et al. 2015).  Our results differ from the ESL estimates
from the U.S. Mid-Atlantic coast and the Yellow Sea deposits show the ocean surface was
shallower than indicated by our observations during MIS 3.  While these estimates serve as
valuable surface of sea level information, they are currently limited by a lack of clear evidence
of submersion by marine water and a reliance on dates from dune and channel sand.  For the



Yellow Sea, since radiocarbon dates on shells from these sediments are at the limit of
reliability, dating of the sand has been accomplished by OSL methods (Liu et al. 2010). The
reflection profiles show that the sampled sand is from a channel fill within a dendritic drainage
system, more likely of fluvial origin rather than sand from a marine transgression. Likewise,
the suggested ESL peak of 40 mbsl during MIS 3 (Pico et al. 2017) in the Albermarle
Embayment on the U.S. Mid-Atlantic coast is based on OSL-dated sands sampled from eolian
dunes with interbedded paleo-sol and resting on peat (Mallison et al. 2008). The diagnostic
evidence of a marine transgression is from mollusk shells under the peat and attributed to MIS
5. Close inspection of analyses from both the Yellow Sea and U.S. Mid-Atlantic coast suggests
assigning a marine transgression to these two areas needs further scrutiny and cannot be
concretely assigned to a eustatic sea level highstand during MIS 3.

**6. Conclusions**

Paleosalinity interpretations from measurements of Ca and $CaCO_3$, porewater $Cl^-$, $\delta^{18}O$

and $^{87}Sr/^{86}Sr$ composition of mollusks from Black Sea and Marmara Seas, $\delta^{18}O$ composition
of the Sofular Cave stalagmite records indicate the two seas were freshwater lakes, connected,
and outflowing to the Mediterranean Sea for the majority of MIS 3, with the possible exception
of a period encompassing 55 to 44 kyr B.P. Lack of marine entry through most of this period
is supported by evidence of ponded/perched lakes in the Marmara Sea that lack any observable
marine deposits. In the future, it is critical to obtain $\delta^{18}O$ and $^{87}Sr/^{86}Sr$ measurements in
carbonates for the Black and Marmara Sea-Lakes during this period to make more robust
conclusions about the paleo-connectivity and outflow of these basins. Low $\delta^{18}O$ of ~-6 ‰ and
$^{87}Sr/^{86}Sr$ of ~0.7088 would show that the water in these lakes was fresh and fed by Black Sea
river water.

Clinoforms, wave cut cliffs and corresponding terraces indicate that the lake level of

the two sea-lakes was at 80 to 90 mbsl during this period, suggesting a positive water budget
and outflow. Both of the sills, the Bosporus that connects the Black Sea with the Marmara Sea
and the Dardanelles that connects the Marmara Sea with the Mediterranean Sea were at the
level of the paleoshoreline. If the RSL on the Mediterranean side of the Dardanelles sill was
higher than the level of the sill, there should be indication of marine entry. As there is not, we
conclude, with taking wave base into consideration, the RSL must have been at or below the
level of the sills, maximum at 65-70 mbsl, for the majority of the period with the exception of
55 to 44 kyr B.P. ESL should have been even lower during this period, as 10-20 m of local



subsidence would have occurred as a consequence of the changes in the distribution of ice
sheets in Eurasia from MIS 3 to present.

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

**Acknowledgments**

The authors would like to acknowledge the crew of the Akademik 2009 and 2011
expeditions, Louise Bolge and Leo Pena for assistance with the geochemical analyses,
Giovanni Aloisi, Helge Arz, Namik Çağatay, Samuel Goldberg for generating GIA corrections
and helpful discussion, Candace Major for the $^{87}Sr/^{86}Sr$ records from the Sea of Marmara,
Guillaume Soulet, and Bill Thompson for sharing datasets employed to reach our conclusions.

**Figures**
Figure 1:

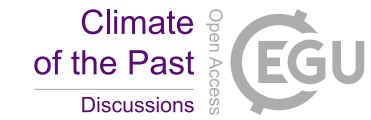

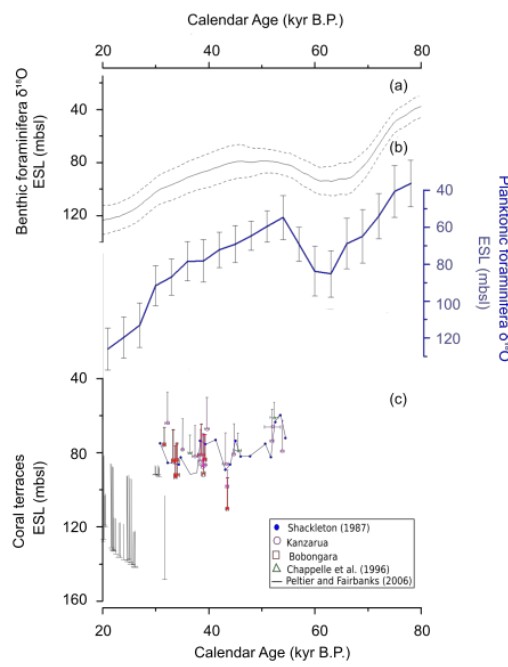

**Figure 1.** Prior sea-level reconstructions for MIS 3. (**a**) ESL reconstruction using inverse
climate ice-sheet modelling from $\delta^{18}$O LR04 record (Bintanja et al. 2005). (**b**) ESL derived
from extraction of ice volume from planktonic $\delta^{18}$O (Shakun et al. 2015). (**c**) ESL
reconstruction corrected for regional uplift and GIA (Yokoyama et al. 2001, Peltier et al. 2006).




Figure 2:





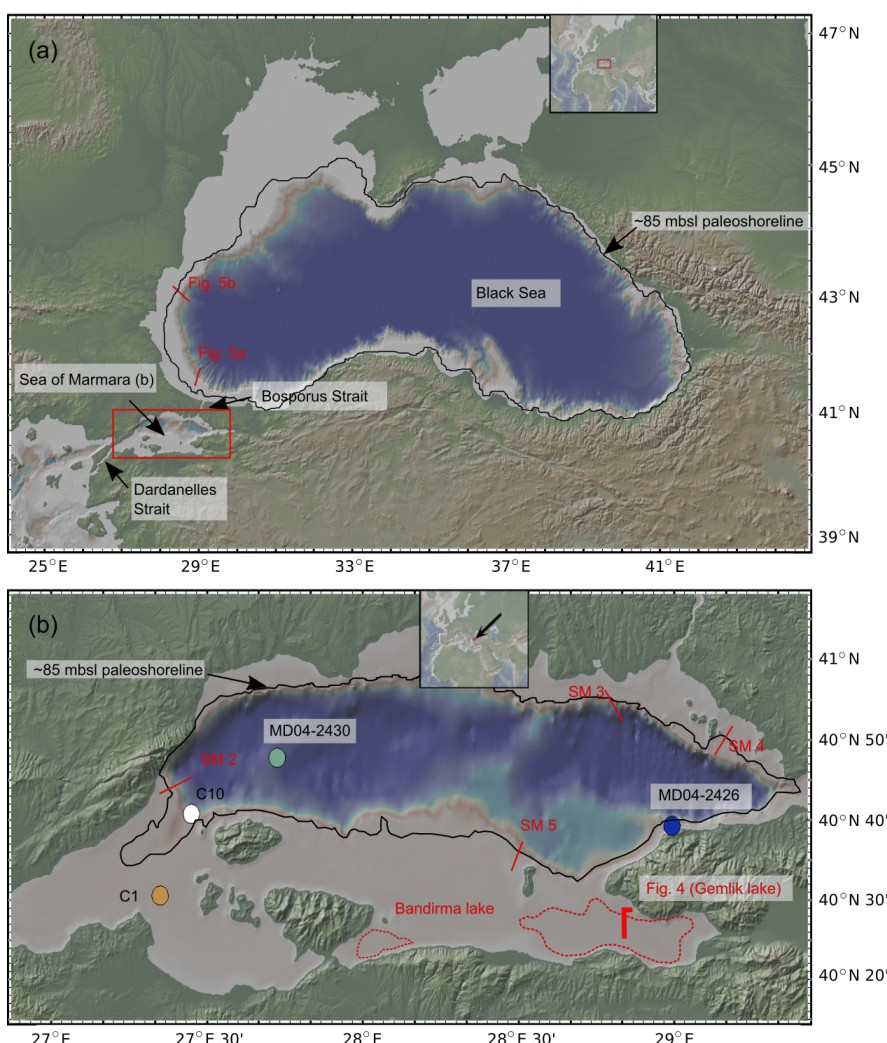

**Figure 2**: Map of the Black Sea (a) and Sea of Marmara (b). Important geographical features are indicated. (a) Location of the Black Sea, Sea of Marmara relative to the Black Seam Bosporus Strait and the Dardanelles Strait. Also are indicated the seismic profiles provided in figure 5a and 5b. The location of the ~85 mbsl paleoshoreline is pointed out. (b) Indicated are the locations of the Gemlik and Bandirma perched lakes in addition to the location of the seismic profile for Gemlik lake presented in Fig. 4. Also are indicated the locations of the cores MD04-2426, MD04-2430, C1, and C10. The locations of the seismic profiles provided in the supplementary materials are also indicated as SM 2, SM 3, SM 4, and SM5.

Figure 3:

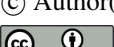


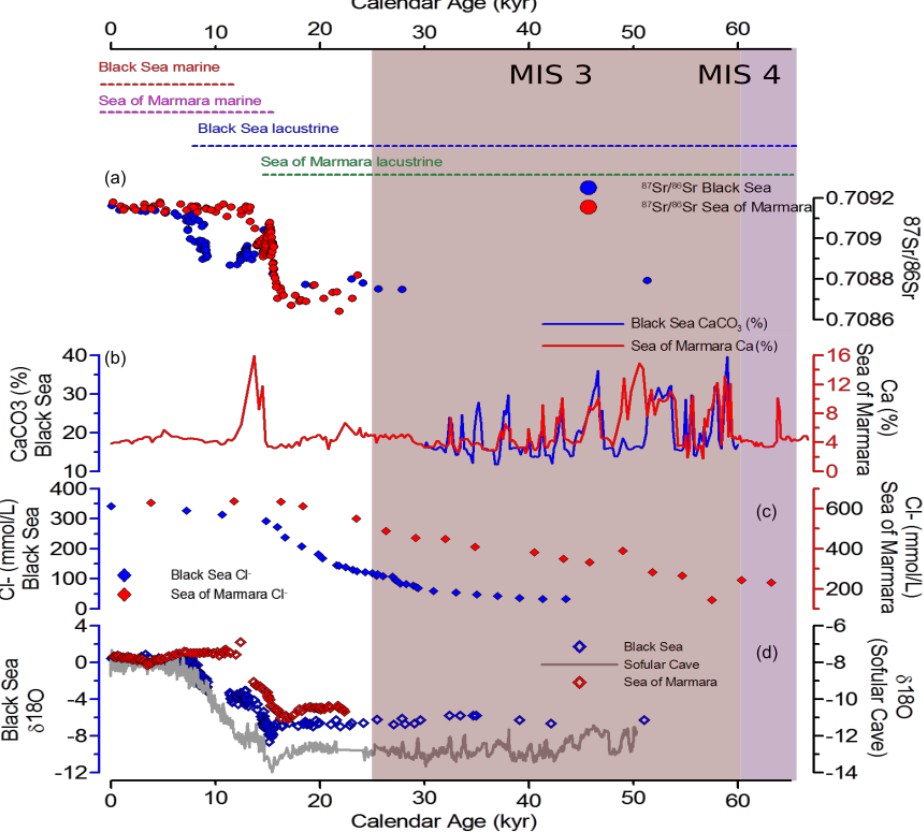

**Figure 3.** Changes in geochemical proxies during MIS 3. (**a**) $^{87}Sr/^{86}Sr$ from the Black and Marmara Sea-Lakes (C.O. Major, National Science Foundation) $^{87}Sr/^{86}Sr$ from the Black Sea is taken from the work of Major et al. (2006), Yanchilina et al. (2017), Yanchilina et al. (2019); the $^{87}Sr/^{86}Sr$ from the Marmara Sea is what is what measured in the present manuscript. (**b**) $CaCO_3$ from the Black Sea-Lake (Nowaczyk et al. 2012) and Ca from the Marmara Sea-Lake (Çağatay et al. 2015). (**c**) Black Sea and Sea of Marmara porewater chlorinity, respectively (Soulet et al. 2010, Aloisi et al. 2015). (**d**) $\delta^{18}O$ from the Black Sea (blue), the Sea of Marmara (red) (Vidal et al. 2010) and Sofular Cave (grey) $\delta^{18}O$ (Badertscher et al. 2011).





Figure 4:



**Figure 4.** Chirp sub-bottom profile across the Gemlik Bay**.** 2 unconformities on the shore of
the Bay, only one in the lake. The shallowest one has been estimated to be associated with the
Last Marine-Lacustrine transition and the deepest.

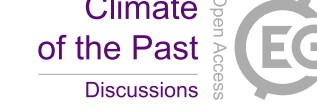



Figure 5:

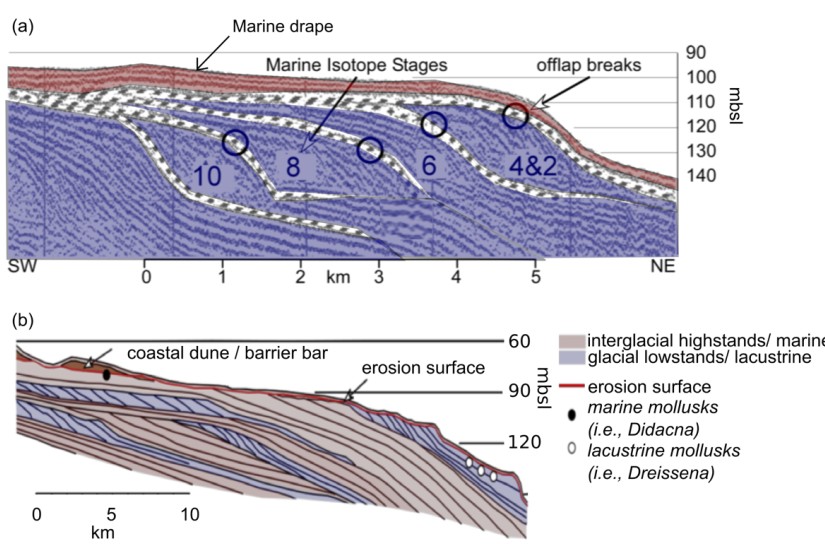


**Figure 5.** Reflection profiles on the SW shelf of the Black Sea. (**a**) Succession of superimposed
prograding clinoforms adopted from a previously published illustration (Fig. 11c) (Aksu et al.
2002) with offlap breaks indicating present and past locations of lowstand deltas at the shelf
edge. Numbers are inferred MIS stages when the Black Sea was a freshwater lake. (**b**) A
similar succession with thinner glacial-age lowstand clinoforms (blue) and thicker interglacial
highstand deposits adopted from an earlier published XIX profile retrieved from R/V
Hydrograph in 1998 (Genov 2015). White circles indicate sampling of lacustrine mollusks of
MIS 2 and 3 age; black circles indicate sampling of marine mollusks presumable of MIS 5 age
(Dimitrov 2010). Coastal dunes place the MIS 3 and 2 shorelines at elevations between 80 and
90 mbsl.