# Peer review of "LACK OF MARINE ENTRY INTO MARMARA AND BLACK SEA-LAKES INDICATE LOW RELATIVE SEA LEVEL DURING MIS 3 IN THE NORTHEASTERN MEDITERRANEAN"

_Climate of the Past, 2019_

## Referee Comment (RC1) · Anonymous Referee #1 · 12 Apr 2019

Yanchilina et al. present a compilation of data that speak to lake levels and paleosalinity in the Sea of Marmara and the Black Sea during MIS 3. The authors argue that these datasets indicate low regional sea level ($\sim$ 80 m below present day) over most or all of MIS 3. As the manuscript stands, it is not entirely clear what data is new, or which interpretations are novel as opposed to drawn from the existing literature. This interpretation of regional sea level near -80 m over much of MIS 3 is based on paleosalinity proxies and seismic profiles of lacustrine deltaic topsets, which are not usually considered proxies for sea level. In my view, the authors do not sufficiently explain how this data can be used to reconstruct regional sea level, and therefore I am not sure these conclusions can be drawn based on the observations presented, as detailed be-

low. Overall, I would recommend that the authors include more discussion of how the data presented can be used to understand sea level, in addition to ample references to literature which might support the proposed interpretation about paleosalinity and lake levels. I would caution the authors to be careful in drawing inferences about sea level from lake levels/salinity, and in doing so, it is necessary to include more references that would support the conclusions.

Specific comments

Salinity proxies are not considered a robust proxy for sea level. A variety of other processes are captured in these measurements including local climate effects such as precipitation patterns and temperature. In the methods section, it is not clear that the authors considered the extent to which these other variables may have dominated the signal measured in these geochemical proxies for salinity. Furthermore, I found that the explanation regarding each method did not include sufficient information (or citations) about what values would be considered significant in indicating saline or freshwater conditions at present day or in the past. I think that the majority of the sea-level community would agree that salinity proxies are not as robust as geological markers such as dated evidence of a shoreline.

Inspection of Figure 3 shows that a number of these salinity proxies do not include measurements during MIS 3 (or in both basins), so it seems difficult to draw a conclusion based on these. it seems plausible that a rise in sea level may not have lasted long enough to substantially change the saltwater content of this region for the duration of time required to be recorded in sedimentary deposits. It would strengthen the paper to include a discussion of what kind of timescales of marine inundation would be required to record an increase in salinity using these geochemical proxies. In fact, previous studies have suggested rapid episodes of sea-level rise of 10-15 m in just 1-2 kyr during MIS 3 (Chappell, QSR, 2002).

Seismic reflection data constitutes the other form of evidence used by the authors to

argue for continuously low sea level near -70 m. I found the figures showing the seismic reflection profiles to be confusing. Importantly, an erosional unconformity exists above the MIS 3 lacustrine ages (which are not reported directly) and this would indicate it is possible that water levels were much higher later during MIS 3 and that any deposition during this time period was subsequently eroded during a major base-level fall leading into MIS 2. Besides the obvious unconformity, which calls into question whether these deltaic deposits represent the highest sea level over the MIS 3 period, I think a more substantial argument is needed for using deltaic topsets as a kind of sea-level record. If the Black & Marmara Seas were not connected to the ocean, as argued in this study, then lake levels will not necessarily reflect sea level. My understanding is that lake levels may be largely controlled by precipitation and evaporation, and would not reflect local regional sea level. A further issue is the treatment of ages used in this study. The ages within MIS 3 deposits (and associated uncertainties) are alluded to, but are actually not reported within the main text. Obviously, this information is crucial to interpreting water depths from this data at a particular instance of time. It would be very helpful to include the ages and location of cores on the figures showing the seismic reflection profiles.

As I have mentioned above, these proxies may not reliably record local relative sea level, however if we were to assume they do, I am still not convinced that these proxies would represent relative sea-level across the entire MIS 3 time interval (60-26 ka). The authors argue based on observations of deltaic deposits and freshwater conditions inferred from geochemical proxies that the sea level in this region was -80 m for most of MIS 3. However, in this manuscript, the authors also note that there may have been a marine incursion from 55-44 ka, although they later dismiss the likelihood of this possibility. This time period, from 50-40 ka, happens to the be the time period during which sea-level high stands are observed globally during MIS 3 (Cann, 2000, Caybioch & Aycliffe, 2001, Hanebuth et al., 2006, Simms et al., 2009, Pico et al., 2016, DaSilva et al., 2017). The authors seem tied to a conclusion about low sea level during MIS 3. I think this manuscript would be improved by presenting the data and

multiple hypotheses, rather than focusing on a single interpretation of this dataset. The important contribution of this study is to bring together a variety of observations about the nature of the connection between the Marmara and Black Sea. In my view, this dataset may or may not shed light on the relative sea level history in this region given the uncertainties associated with the methods used (paleosalinity proxies and deltaic topsets are not considered robust geologic sea level indicators). However, this dataset may be an important contribution to understanding the salinity history of these basins, and I think the authors should focus on this (and presenting possible hypotheses) rather than drawing much wider conclusions.

Line by line comments

Line 17 – Delete "the" in front of MIS 3

Line 17 – what do you mean by persistent?

Line 26 – eustatic is not in caps

Line 38 – replace 'elevations" with "estimates"

Line 39 – delete "existence of"

Line 40 – delete "factors that control"

Line 41 – delete "additionally"

Line 43 – relative sea level not in caps

Line 45 – "The earth is moving" is imprecise language. This sentence should be written
Line 47 – delete "the"

Line 50 - I think you mean methods... while there are some theoretical differences in approaches it is largely the tools that you are referring to

Line 53 – (3) is missing from this list

Line 56 – grammar/wording in sentence "Isolating.." - you are not isolating ice volume

from these records, you are estimating the contribution of global ice volume to changes in seawater delta 18O

Line 65 - This is not exactly what the study showed - it used estimates from the Pico et al., 2016 paper to infer the source of ice loading in North America

Line 69 – This may be misleading because the authors do not run a GIA model to look at this question specifically. Rather authors should say they use previous studies of GIA in this region to estimate the effect of GIA-related SL changes

Line 91 – Confusing sentence

Line 97 – need citation for volume of Marmara and river inflow

Line 100 - confusing wording. Do you mean the lakes must have been alkaline in order to explain these accumulations? And why would you get this kind of accumulation during warming? Could you explain this concept more clearly?

Line 108 – Can you cite references for what values would be considered freshwater or saline? What is the range of values you might find globally? What values are considered significant?

Line 130 - Is this new data in this study? Authors should make that very clear, here, and in introduction. What kind of new data are the authors presenting?

Line 134 – need reference!

Line 137 - Can you cite something? And what is the change you would expect per volume % change of marine water?

Line 140- How would you expect this value to change through time?

Line 165 – Is this standard practice? Can you cite a study that has used a similar method?

Line 195-196 – Confusing sentence

Line 196 – "contest" means to disagree, I think you mean we present data to support the hypothesis that the two lakes were freshwater.

Line 204- How exactly is this signal changing over this time period? Can you reference a figure that would show this?

Line 207 - There are no references for this. How do you know that every time there is marine incursion there will be a sapropel layer? How long would you expect that a marine incursion should last in order to develop this layer? Also probably should briefly define what a sapropel is for readership

Line 207 – delete "evidence of", delete "here"

Line 211 - How do you know this is the reason? You need a little more explanation connecting the observations with the mechanism for producing these.

Line 215 – Confusing discussion

Line 218 – Replace "with the light" with "in light of"

Line 236 – Delete "Index"

Line 246 – Replace "indicates..." with "suggesting a freshwater environment"

Line 284 – However there is a large SL fall after MIS 3, so you might expect a lot of erosion during this time. No deposition during MIS 3/2 is simply absence of evidence rather than evidence in itself.

Line 310 – grammar in sentence "It remains possible..."

Line 321 – RSL index is not a conventional term, you might mean RSL indicator. Essentially you want to say that this set of data suggests RSL ranged from X-X mbsl. But, do you mean the seismic data suggest this? These are not really ocean shorelines; the elevation of lake shorelines cannot be used to represent a sea level shoreline. I would be very skeptical of interpreting sea level from lake levels because there are a

number of other factors that control lake elevation including precipitation, for example. If you are inferring this from the geochemical data can you explain exactly where those numbers (70-90 m) come from (depth of sill?). This needs to be much clearer, as this is a main conclusion of the paper. However, as the paper stands I am not sure what data this sea level is inferred from, and cannot determine how robust this conclusion is.

Line 374 – This is uncited and it is not clear how the authors estimated this.

Line 583 – Figure 3 Based on this figure I feel confused about the conclusion. Which proxies are supposed to be the same in both water bodies if there is freshwater? Also, there is not a record during most of MIS 3 for all proxies in both water bodies, so it is not clear that it makes sense to compare these proxies (for example Sea of Marmara does not have Cl measurements during MIS 3 and near does Sr for either water bodies).

Line 605 – Figure 4. What is the point of this figure? Is this an area that matters or are we interpreting a paleo shoreline? I am not sure I understand the significance of a shoreline in Gemlik Bay as this is far from the connection between the Marmara and Black Sea.

Line 641 – Are there cores taken in the profile in Fig 5a? What are the dates? It would be great to include the dates next to the location of these. Are these figures from another study? I think that this could be said at the beginning of the figure caption. It is confusing that these might be from two separate papers so the terms are not the same. For example, an erosional surface is shown in Fig 5b, but these aren't highlighted in Fig 5a. It would be great if the terminology in the captions were consistent across both figures.

References:

Chappell, J. (2002). Sea level changes forced ice breakouts in the Last Glacial cycle: New results from coral terraces. Quaternary Science Reviews, 21(10), 1229–1240.

http://doi.org/10.1016/S0277-3791(01)00141-X

Cann, J. H. (2000). Late Quaternary Paleosealevels and Paleoenvironments Inferred From Foraminifera, Northern Spencer Gulf, South Australia. The Journal of Foraminiferal Research, 30(1), 29–53. http://doi.org/10.2113/0300029

Cabioch, G., & Ayliffe, L. K. (2001). Raised Coral Terraces at Malakula, Vanuatu, Southwest Pacific, Indicate High Sea Level During Marine Isotope Stage 3. Quaternary Research, 56, 357–365. http://doi.org/10.1006

da Silva Salvaterra, A., Cesar, R., Figueira, L., & Mahiques, M. M. De. (2017). Evidence of an Marine Isotope Stage 3 transgression at the Baixada Santista , south-eastern Brazilian coast. Brazilian Journal of Geology, 47(December), 693–702. http://doi.org/10.1590/2317-4889201720170057

Simms, A. R., DeWitt, R., Rodriguez, A. B., Lambeck, K., & Anderson, J. B. (2009). Revisiting marine isotope stage 3 and 5a (MIS3-5a) sea levels within the northwestern Gulf of Mexico. Global and Planetary Change, 66(1–2), 100–111. http://doi.org/10.1016/j.gloplacha.2008.03.014

Hanebuth, T. J. J., Saito, Y., Tanabe, S., Vu, Q. L., & Ngo, Q. T. (2006). Sea levels during late marine isotope stage 3 (or older?) reported from the Red River delta (northern Vietnam) and adjacent regions. Quaternary International, 145–146, 119–134. http://doi.org/10.1016/j.quaint.2005.07.008

---

## Short Comment (SC1) · 19 Apr 2019

The publication "Lack of marine entry into Marmara and Black Sea-lakes indicate low relative sea level during MIS 3 in the northeastern Mediterranean" was written in high professional level and confirmed all data of investigations in the Bulgarian Sector of the Black Sea during for the last 40 years.

---

## Referee Comment (RC2) · Anonymous Referee #2 · 1 May 2019

Yanchilina et al., "Lack of marine entry into Marmara and Black Sea-lakes indicate low relative sea level during MIS 3 in the Northeastern Mediterranean".

Summary: This manuscript presents new 87Sr/86Sr data from the Sea of Marmara (the Black Sea Sr/Sr data was published in Yanchilina et al., 2017) that provides constraints on the timing of the reconnection of the Sea of Marmara and the Black Sea with the Mediterranean. The authors also present seismic/reflection profiles from the region, although only one is new (Chrip profile of Gemlik Bay, figure 2).

I have significant concerns about the work presented here (for example; the age model, interpretation of the proxies within the wider context of the region, mechanisms and

assumptions. . .) and feel the submission of this manuscript is premature. The work could make a good contribution to the field, however, it requires more thought, a clearer focus and greater attention to detail to do so. As such, I would not recommend the acceptance of the manuscript for publication in Climate of the Past in its current format.

GENERAL COMMENTS

The main issues that must be addressed are:

(1) Greater integration of the literature: Currently, the framing of this work within the wider context of the literature on the Black Sea and Sea of Marmara is poor (for example, there is no mention of the work of Aksu et al., 2016, Yaltirak et al 2002 etc.). Additionally, the selection and representation of sea-level data for the interval is inadequate (see section 3.2 below for examples).

(2) Focus of the manuscript (and title): the data here is not a sea-level record per se, rather a record of marine incursion(s) into the Sea of Marmara and the Black Sea with rising sea level during the deglaciation. A key outstanding question that this paper helps to address, is when these transitions occurred (although your age model needs considerable work, see below). What you have here are very valuable independent constraints for the region on rising deglacial sea level (87Sr/86Sr data). I would suggest trimming (or removing?) most of the introductory sea level discussion (which is far too general), and focus on the new 87Sr/86Sr data and seismic/reflection/Chirp profiles (most of which have been relegated to the supplement). If you wish to make this more of a sea level story, you will need greater consideration the wider sea-level data available in MIS 3 (see below). The focus on MIS 3 is also a little odd, given the data you present. In figure 3, your Cl data suggests increasing salinity but you do not really attempt to untangle the mechanisms driving this (see section 3.3). Similarly, you do not comment on the fluctuations in the Ca (Sea of Marmara) and CaCO3 (Black Sea) records. Is there a connection to the Dansgaard-Oeschger events that are characteristic of the time period?

(3) General lack of rigour: The three major aspects that need addressing are;

3.1 There is insufficient information in your methods (in particular, your age model is not described in sufficient detail, see below);

(3.1.1) Age model. I am unconvinced that your age model is reasonable (or robust) given that methodology and rationale/mechanistic relationships are currently poorly explained and justified. Either more detail is needed (including reporting of the 14C data, see comment below, see *), clearly stating how your age model was constructed (a supplementary figure would be useful), or I suggest adopting a more direct approach using fact that the Black Sea is the dominant source water source for Sofular Cave (you do mention this in lines 167 to 169). In the latter, you could use the well dated (i.e., precise, radiometric dates) of the speleothem to constrain your age model (which you can then check and/or refine with your 14C determinations, especially given your records extends beyond the limits of radiocarbon). Although you do not have a high resolution, continuous $\delta 18O$ record for either the Black Sea or Sea of Marmara to tune to the speleothem record, you do have very nicely resolved Ca (Sea of Marmara) and CaCO3 (Black Sea) records that have good signal correspondence to the Sofular Cave $\delta 18O$ record (and the Dansgaard-Oeschger (D/O) events more generally, e.g., Rasmussen et al., 2014) – warmer – more blooms and higher Ca/CaCO3 production etc. I would also suggest increasing the vertical exaggeration of the Sofular $\delta 18O$ record in figure 3 to help the reader.

Eyeballing your record of Ca (Sea of Marmara) and CaCO3 (Black Sea), I would place the transition from low to higher values (and more square-wave signal) that you currently have at 55 ka at about 48 ka. This would shift most of your records to younger ages (this has the upshot of making your data more consistent with other sea level records – see below)

I would suggest an additional step of incorporating the stratigraphic relationships into you age modelling, and assessment of the age uncertainties of the final age model

[Figure]

(e.g., using Bacon, or the deposition model in OxCal). This is not vital but it would allow you to provide some age uncertainty estimates for the marine incursion(s) into the Sea of Marmara and the Black Sea.

\* It must be an oversight that you do not fulfil the minimum reporting requirements for the 14C dates (e.g., Stuvier and Polach, 1977; Mook and van der Plicht, 1999; Millard, 2014). I know these are not new dates but I would expect as a minimum for you to list the 14C dates you use, the source for any $\Delta$R correction you use, nor the calibration curve/programme you use. A supplementary figure with the age-depth relationship would be a good addition.

(3.1.2) Chirp profiles: There is insufficient information here (lines 175 to 182). What was the vertical resolution? What processing did you undertake (and using what software) etc.?

3.2 As mentioned in (1) above, there is poor integration with other available data and literature. For example, there are some cursory attempts to couch this work within the literature but most are related to the proxies and seismic/reflection profiles presented.

Other issues that should be considered include:

(3.2.1) What is the impact of glaciation (e.g., the potential outflow of glacially dammed rivers and lakes) and especially the deglacial, e.g., the melting of Northern hemisphere ice sheets filling the Black Sea, e.g., Chepalyga, 2007, Thom, 2010, Vidal et al., 2010, which in turn led to the outflow of brackish water to the Mediterranean via the Marmara Sea? Also, how do your palaeo shorelines compare to the lowstand terrace in Sea of Marmara at ∼-85 m (Çaħatay et al., 2009, Asku et al., 1999)? These authors suggest that post 15 ka in Sea of Marmara, evaporation exceeded riverine and Black Sea inputs – how does this compare to your work?

(3.2.2) There are other estimates for the depth of the sills (e.g., Major et al., 2002 – a co-author? - gives the elevation of the Dardanelles sill as -85 $\pm$ 5 m, which is consistent

with the clinoforms). In addition, you do not discuss (or model) the GIA processes and how these might affect the connection between the various basins.

(3.2.3) The discussion of MIS 3 sea levels is incomplete and misses some key references. There also seems to be some confusion/conflation of relative- (RSL) and eustatic sea level (ESL), and ice volume equivalent throughout the manuscript. You explicitly state the difference between RSL and ESL (in lines 44 to 45) and yet the discussion of the various means of determining sea level (and what, RSL, ESL or ice-volume equivalent) in the introduction is muddled (lines 51 to 68) and omits several well-constrained lines of evidence (as does figure 1). The most obvious are the high resolution, continuous relative sea level records from the Red Sea (e.g., Grant et al., 2012) and the Mediterranean (Rohling et al., 2014) – both of which are publically available.

In more detail, in lines 51 to 57, you also have a list of 1 to 4 methods for deriving past changes in sea level, but 3 is missing. These are subsequently returned to in the discussion but only in a very superficial manner. My comments are:

Isotopic methods and deconvolution of the $\delta$18O signal: The oxygen isotope ratio of marine sediments can be used to infer past sea levels (as a first order approximation) using the relationship between the $\delta$18O of the mineral precipitated (e.g., foraminiferal calcite) and the processes governing the hydrological cycle (and thus sea level). The relative contribution of global ice volumes and temperature to foraminiferal oxygen isotopes is complex and subject to substantial uncertainties and several attempts to unravel this are available – e.g., through various assumptions and/or modelling (e.g., Bintanja et al.,2005, Shakun et al., 2015, as mentioned by the authors but see also de Boer et al., 2014, Waelbroeck et al., 2002, Elderfield et al., 2012). However, it should be noted that in all these reconstructions, the global ice volume component is comprised of both a terrestrial component AND any floating ice. Changes in the former would contribute to both $\delta$18O AND sea level, whereas changes in any floating ice would ONLY change the $\delta$18O record and not sea level. In other words, reconstructed

changes in global ice volumes may not be equivalent to changes in sea level (e.g., Rohling et al., 2017). In figure 1, this could easily be fixed by changing the axis labels of (a) and (b) to ice volume. The authors might also consider adding the Elderfield et al. (2012) and/or the de Boer et al. (2014) reconstructions.

Coral terraces: these are RSL records, unless they have been corrected for glacio-isostatic (GIA) processes, in which case they do provide ESL constraints. In figure 3, the data is incorrectly referenced and there are no details on your(?) GIA corrections to the data. Please clarify. In figure 3, you plot (some) of the coral Barbados (for other Barbados sea-level data within the time period e.g., Bard et al., 1990, Fairbanks et al., 2005), and (some) Huon Peninsula (Yokoyama et al., 2001 but see also Cutler et al., 2003) data along with the $\delta$18O record of Shackleton (1987). The latter is an ice volume equivalent sea level not ESL. There are other coral sea-level records available that span some of your time window – e.g., Chappell, 2002, Cutler et al., 2003; or the very recent Yokoyama et al 2018 paper from the Great Barrier Reef. The authors might consider a wider selection of coral data. . .?

Lithofacies, salt marshes etc. where former sea levels are reconstructed using a modern analogue for the relationship between the indicator and sea level at the time of formation (note, the Pico et al., 2017 study is a GIA modelling studies of previously published sea-level reconstructions, assuming different ice models – i.e., variations in the volume and the spatial extent of the former ice sheets – as well as Earth rheologies). The mention of these in the introduction is a little odd, given that none of this data is plotted nor referred to in the text. The Pico et al. (2016, 2017) studies are returned to in lines 339 to 340 but with very little analysis.

Given the above, the introductory section does not sit well with the data you present, and the discussion of MIS 3 sea-level data is poor, in particular how this fits with your data. I would significantly trim this unfocused sea level portion of the introduction and discussion and refocus your manuscript on the timing of the (re)connection of the Sea of Marmara and Black Sea to the Mediterranean.

3.3 Insufficient/simplistic consideration of mechanisms of change and what is influencing your proxies – there was no real consideration of the:

(3.3.1) hydrological balance of the palaeo-lakes (evaporation, precipitation, lake area and riverine inputs; the potential impact of the glacial re-routing of riverine inputs etc.) and how this might impact your proxies. In addition, there seemed to be some confusion of the systematics of $\delta$18O in marine, lake and speleothems environments.

(3.3.2) impact of the former proximal ice sheets on the glacio-isostatic response of the region;

(3.3.3) tectonic setting of the region and the influence of active faults (e.g., fault segments that developed during the Late Pleistocene, for example, see Vardar et al., 2014 and references therein).

3.4 Writing: some careless mistakes in the manuscript – for example, poor/incomplete referencing (e.g., line 34) and repetition (line 323 to 324 is immediately repeated as the next sentence).

TECHNICAL CORRECTIONS

References: Greater care with referencing needed. Please check manuscript. For example, line 34: "Members 2006" should be "EPICA Community Members"

Figures: Figure 1: incorrect axis labelling, poor selection of available data, inaccurate/incomplete referencing of data in the caption.

REFERENCES CITED: • Aksu, A.E. et al., 2016. Early Holocene age and provenance of a mid-shelf delta lobe south of the Strait of Bosporus, Turkey, and its links to vigorous Black Sea outflow. Mar. Geol., 380, 113-137 • Aksu, A.E., et al., 1999. Oscillating Quaternary water levels of the Marmara Sea and vigorous outflow into the Aegean Sea from the Marmara Sea-Black Sea drainage corridor. Mar Geol, 153, 275–302. • Bard, E., et al., 1990. Calibration of the 14C timescale over the past 30,000 years using mass spectrometric U-Th ages from Barbados corals. Nature, 345, 405-

410. • Bintanja, R., et al., 2005. Modelled atmospheric temperatures and global sea levels over the past million years. Nature 437, 125–128. • Çaħatay et al., 2002. Late Glacial–Holocene paleoceanography of the Sea of Marmara: timing connections with the Mediterranean and the Black Seas. Mar Geol., 167, 191–206. • Chappell, J. 2002. Sea level changes forced ice breakouts in the Last Glacial Cycle: New results from coral terraces, Quat. Sci. Rev., 21, 1229-1240 • Cutler, K.B., et al., 2003. Rapid sea-level fall and deep-ocean temperature change since the last interglacial period. Earth and Planetary Science Letters, 206, 253-271. • de Boer et al., 2014. Persistent 400,000-year variability of Antarctic ice volume and the carbon cycle is revealed throughout the Plio-Pleistocene. Nature Comms. doi: 10.1038/ncomms3999 • Elderfield, H., et al. 2012. Evolution of ocean temperature and ice volume through the mid-Pleistocene climate transition. Science 337, 704–709. • Fairbanks, R. G. et al. 2005. Radiocarbon calibration curve spanning 0 to 50,000 years BP based on paired Th-230/U-234/U-238 and C-14 dates on pristine corals. Quat. Sci. Rev. 24, 1781-1796. • Grant, K.M., et al., 2012. Rapid coupling between ice volume and polar temperature over the past 150,000 years. Nature, doi:10.1038/nature11593 • Major, C., et al., 2002. Constraints on Black Sea outflow to the Sea of Marmara during the last glacial- interglacial transition. Mar. Geol. 190:19–34. • Millard, A. 2014. Conventions for reporting radiocarbon determinations. Radiocarbon 56, 555–559. • Mook, W. G. & van der Plicht, J. 1999. Reporting 14C activities and concentrations. Radiocarbon 41, 227–239. • Rasmussen, S.O., et al., 2014. A stratigraphic framework for abrupt climatic changes during the Last Glacial period based on three synchronized Greenland ice-core records: refining and extending the INTIMATE event stratigraphy. Quat. Sci. Rev., 106, 14-28. • Rohling, J.R., et al., 2014. Sea-level and deep-sea-temperature variability over the past 5.3 million years. Nature, 508, doi:10.1038/nature13230. • Potter E-K., et al., 2004. Suborbital-period sea-level oscillations during marine isotope substages 5a and 5c. Earth and Planetary Science Letters 225: 191-204. • Shakun, J.D., et al., 2015. An 800-kyr record of global surface ocean $\delta$18O and implications for ice volume-temperature coupling. Earth and

Planetary Science Letters, 426, 58-68. • Stuiver, M. & Polach, H. A. 1977. Reporting of 14C data. Radiocarbon 19, 355–363. • Vardar, D., et al., 2014. Late Pleistocene–Holocene evolution of the southern Marmara shelf and sub-basins: middle strand of the North Anatolian fault, southern Marmara Sea, Turkey. Mar Geophys Res, 35, 69–85. • Waelbroeck, C., et al., 2002. Sea-level and deep water temperature changes derived from benthic foraminifera isotopic records. Quat. Sci. Rev., 21, 295-305. • YaltÄśrak, C. M et al., 2002. Global sea-level variations and raised coastal deposits along the southwestern Marmara Sea during the last 224,000 years Mar. Geol., 190, 283-305. • Yanchilina, A.G., et al., 2017. Compilation of geophysical, geochronological, and geochemical evidence indicates a rapid Mediterranean-derived submergence of the Black Sea's shelf and subsequent substantial salinification in the early Holocene. Mar. Geol. 383, 14-34. • Yokoyama, Y. et al., 2001. Coupled climate and sea-level changes deduced from Huon Peninsula coral terraces of the last ice age. Earth and Planetary Science Letters 193, 579-587. • Yokoyama, Y. et al 2018. Rapid glaciation and a two-step sea level plunge into the Last Glacial Maximum. Nature, 559, doi.org/10.1038/s41586-018-0335-4.

---

## Short Comment (SC2) · 2 May 2019

The paper "Lack of marine entry into Marmara and Black Sea-lakes indicate low relative sea level during MIS 3 in the northeastern Mediterranean" presents new and valuable data on a subject that still raises a lot of questions. In my opinion the information that you present in this study are of great importance to our understanding of paleoclimate and paleoceanographic conditions in the restricted Black Sea and Marmara Sea. Keep up the good work and I wait the next paper, detailing the paleo-connectivity and outflow of Black and Marmara Seas.

---

## Editor Comment (EC1) · Erin McClymont (Editor) · 6 May 2019

Dear Dimitar Petkov Dimitrov, Thank you for reading the discussion paper and for posting a comment. However, the discussion phase is the opportunity for the scientific community to propose and discuss additional scientific comments on the paper, for the authors to consider and submit a reply to. As you have not provided points for discussion here I will not expect the authors to make any reply, but nor will I include your comments as part of the evaluation of the manuscript. Best wishes, Erin McClymont (Editor)

---

## Editor Comment (EC2) · Erin McClymont (Editor) · 6 May 2019

Dear Andrei Briceag, Thank you for reading the discussion paper and for posting a comment. However, the discussion phase is the opportunity for the scientific community to propose and discuss additional scientific comments to the paper for the authors to consider and submit a reply to. As you have not provided points for discussion here I will not expect the authors to make any reply, but nor will I include your comments as part of the evaluation of the manuscript. Best wishes, Erin McClymont (Editor)
* * *

---

## Author Comment (AC1) · 14 May 2019

Response to first peer review:

Response to (1) paleosalinity and seismic profiles not usually considered regional sea level proxies. First of all, just because the methods and conclusions drawn from non-traditional methods are not considered traditional and usual is not a justified reason for not considering the conclusions drawn. We see that the concept presented is very simple.

Before we continue with our response, however, we would like to note that saline intrusion in and out of basins, that are marginally connected to the global ocean, have been used to give an idea for regional and eustatic sea level in the past. We think that including more of this discussion will help explain how our method will additionally contribute to reconstruction of paleo sea level.

The first example is from the work of Van Daele et al. (2011). Here, the authors look at infills in the Gulf of Cariaco, a marginal basin, that is connected to the Caribbean Sea via a shallow 58-m-deep sill. The authors use a similar idea to ours, changes in regional sea level, to infer when the regional sea level was higher than the sill versus lower. When the regional sea level is higher, saline water intrudes into the Gulf of Cariaco, creating sedimentary infills. When the regional sea level is lower, saline water does not intrude and sedimentary infills related to saline water intrusions do not occur.

The second example is from the work of Pico et al. (2016). The authors use sedimentary core analyses from a Yellow River Delta in the Bohai Sea of China to make inferences about a migrating paleoshoreline. Pico et al. (2016) use information from cores taken from the delta, to observe changes in inundation of the delta associated with first changes in regional sea level and second, make further to make conclusions about eustatic sea level after applying glacio-isostatic corrections.

The third example is using changes in the $\delta$18O record of the eastern Mediterranean Sea (Grant et al. 2012) and the Red Sea (Siddall et al. 2003) using a basin isolation concept to infer changes in regional sea level. The basin isolation concept features reduction of intrusion of saline water into a basin that is marginally connected to the global ocean; in the Red Sea from the description of Siddall et al. (2003), "Reduction of the strait profile by sea-level lowering decreases the exchange transport of water masses through the strait. This results in increased residence times of the water within the Red Sea, enhancing the effect of the high rate of evaporation (2.06 m yr-1) on properties in the Red Sea. The basin though amplifies the signal of sea level change, which are recorded in d18O values of foraminifera in Red Sea sediment cores."

These above all give examples of non traditional methods that do not include uplifted coral terraces that have been used to reconstruct past regional and global sea levels.

For our manuscript, we want to make conclusions from a simple concept. The Black and Marmara Seas are connected to the Mediterranean by two shallow sills, both approximately shown to have been 80 meters below sea level during MIS 2 and 3 (the sill level connecting Marmara and Black Seas, Bosporus, is now 30 mbsl but during MIS 2 and 3 it was lower, at 80 mbsl). Now, marine water enters through these sills and these seas are saline because the regional sea level (from the Mediterranean side) is higher than the sills, hence, saline water has no option than to flow into both of these basins and make them seas as opposed to freshwater lakes. A good visual schematic is filling water in a bathtub. The water flows and if its higher than the level of the bathtub, it will overflow over the edges of the bath tub onto the floor. Same concept almost for the water inflowing into the Marmara and Black Seas, when the level of the Mediterranean is lower, no salt water flows in and when the level of the Mediterranean is higher, salt water flows in. This is what we are trying to conclude from the data presented. The paper is not about making conclusions about lake/sea level in the Black Sea and the Marmara Seas, although it alludes to them, the paper is about the regional sea on the Mediterranean Side of the Marmara and Black Seas.

We agree that it would be great to have detailed regional and global sea level data from uplifted coral terraces but such information is largely absent and is mostly available for the glacial and postglacial and the Eemian. One of the reasons we are making and drawing these conclusions is because of the lack of regional and global sea level data that currently exists during MIS 3. We believe that knowing a constraint for regional sea level and an overall understanding that it was low, is a very important contribution for the paleo sea level community.

If there was any marine intrusion into the Marmara and Black Seas, there would have been a rise in salinity, during Marine Isotope Stage 3. There is not recorded a change in paleosalinity from all proxies available to date, most strongly the lack of change in

the $\delta$18O of the Sofular Cave in the Black Sea, shown to reflect $\delta$18O of surface Black Sea water. Porewater Cl- from both Marmara and Black Seas adds to this conclusion. Second (2), new results. We have a section that indicates all new results, we do not understand how this not clear to the reviewers. We have sections 3.1 and 3.2. In section 3.1, we detail the new data that we are presenting and adding from the geochemistry side, which are 87Sr/86Sr from the Sea of Marmara that go into MIS 3 and in section 3.2, we add data for a chirp profile of a perched lake in the Sea of Marmara that does not is there to present no entry of water from external sources, such as the Mediterranean Sea/ global ocean, during MIS 3, supporting what is seen from geochemistry. We agree that discussion of the time scales necessary for salinities of the Black and Marmara Seas to have responded to marine incursions would make this a stronger paper and hope to make these changes in the revised manuscript, including the observation of changes in eustatic sea level on time scales of 1-2 years by 10-15 m as shown by work of Chappell (2002).

Regarding the seismic profiles, while it is true there is a seismic unconformity related to sea level fall during MIS 2, and prior sediment, if it was present, would have been removed, our point is that no such prior sediment existed. This is largely seen from the Figure 5 and Supplementary Material Figures 1-5. The main clear point we are trying to make with these figures can be seen From Supplementary Figure 4. Core C9 dates the cut off sedimentary layers at the modern depth of 80-90 mbsl to 24.9 14C thousand years. This indicates, that those layers that were outcropping towards Southeast (SE), (in this figure to the right of the 14C date), must be younger than 24.9 14C years. In calendar years, 24.9 14C thousand years is the end of MIS 3 / beginning of MIS 2 so all of the sediments belonging to those cut off to the right, must all be MIS 3, by logic of sediment accumulation, indicating the local lake level in the Sea of Marmara was low and lower than 80 mbsl, regionally. If the regional sea level on the Mediterranean side, was higher, it would have overfilled the Marmara Sea and the deposits that would have formed, would be like those indicated in the red above the blue (that date to the post connection of the Marmara Sea with the Mediterranean). It doesn't matter that there is

an unconformity during MIS 2, the sediments that were deposited during MIS 3 are still there. This is the same pattern observed all over the Sea of Marmara and the Black Sea (Figure 5, Supplementary Figures 1-3, 5).

References:

Grant, K. M., E. J. Rohling, M. Bar-Matthews, A. Aaylon, M. Medina-Elizalde, C. B. Ramsey, C. Satow and A. P. Roberts: Rapid coupling between ice volume and polar temperature over the past 150,000 years, Nature, 491, 744-47, 10.1038/nature11593, 2012. Pico, T., J. X. Mitrovica, K. L. Ferrier and J. Braun: Global ice volume during MIS 3 inferred from a sea-level analysis of sedimentary core reciords in the Yellow River Delta, Quaternary Science Reviews, 152, 72-79, 2016. Siddall, M., E. J. Rohling, A. Almogi-Labin, C. Hemleben, D. Meischner, I. Schmelzer and D. A. Smeed: Sea-level fluctuations during the last glacial cycle, Nature, 423, 853-58, 10.1038/nature01690, 2003. Van Daele, M., A. van Welden, J. Moernaut, C. Beck, F. Audermard, J. Sanchez, F. Jouanne, E. Carrillo, G. Malavé, A. Lemus and M. De Batist: Reconstruction of Late-Quaternary sea- and lake-level changes in a tectonically active marginal basin using seismic statigraphy: The Gulf of Cariaco, NE Venezuela, Marine Geology, 279, 37-51, 2011.

---

## Author Comment (AC2) · 14 May 2019

Response to second peer review:

Reviewer: The reviewer comments in 3.2.3 that the introductory section does not sit well with the data that we present and the discussion of MIS 3 sea-level data is poor and in particular how this fits with our data. It is suggested to significantly trim this unfocused sea level portion of the introduction and discussion and refocus your manuscript on the timing of the reconnection of the Black Sea with the Mediterranean.

Response: We respond that we want this to be specifically a regional sea level data,

not connections of the Black and Marmara Seas with the eastern Mediterranean. Our contribution to writing about the last reconnection of the Black Sea with the Mediterranean is previously published in Yanchilina et al. (2017) and (Yanchilina et al. 2019).

Reviewer: The reviewer comments about the age model that either more detail is needed, including reporting of the 14C data, clearly stating how the age model was constructed, and/or a more direct approach using the fact that the Black Sea is the dominant source for water vapor feeding the formation of the Sofular Cave stalagmites. The reviewer goes to further suggest that we could use the well dated speleothem $\delta$18O and $\delta$13C records to constrain the age model which we can check or refine with the 14C determinations, especially for the time period that extends beyond the limits of the radiocarbon dating.

Response: We agree and we should have described the age model for each of the geochemical set of measuremennts we present in Figure 3. The only new data in the figure is our 87Sr/86Sr data from the Sea of Marmara which we describe in lines 163-164. We should have mentioned that we follow the age model for the $\delta$18O and $\delta$13C of the mollusks in the Black Sea previously published in Yanchilina et al. (2017) and Yanchilina et al. (2019), which we did exactly with the method advised here by the reviewer, matching the $\delta$18O and $\delta$13C of the mollusks to the $\delta$18O and $\delta$13C composition of the U/Th dated Sofular Cave stalagmites.

Reviewer: Age model - It must be an oversight that the minimum reporting requirements for the 14C dates are not confirmed. One expects to list the 14C dates that are used, any source for the 14C reservoir correction that you use, and the calibration curve / programme. A supplementary figure with the age-depth relationship would be a good addition.

Response: We are more than happy to include all of the measurements we present in Figure 3 along with the 14C measured dates. If the manuscript gets accepted as a paper in Climate in the Past, we will submit the measurements as a supplementary

spreadsheet.

Reviewer: Interpretation of the proxies within the wider context of the region - D18O. Confusing sea level with ice volume.

Response: Not exactly. Bintanja et al. (2005) make separate figures for ice volume (Figure 3a) of their paper and global mean sea level (Figure 3b). They specifically state, "the main strength of our method is that it yields long and mutually consistent records of surface air temperature, ice volume, and global sea level by separating the ice-sheet and deep-water parts of the marine $\delta$18O signal."

Similarly, Shakun et al. (2015) also separately discuss sea level and ice volume contributions to the $\delta$18O records. Their methodology is described in section 3.3 of their paper. Hence, our figures 1a and 1b are correct in referencing ESL and not ice volume.

Having said this, we agree that we should more thoroughly discuss the difference between ice volume and sea level in our introduction sections.

Reviewer: - Coral terraces: these are RSL records. The data is incorrectly referenced and there are no details on GIA corrections. Also see Cutler et al. (2003), Chappell (20020, Yokoyama et al. (2018) from the Great Barrier Reef, Shackleton d18O record (1987) with the latter being an ice volume equivalent, not ESL

Response: These are not RSL records.

With the exception of the Barbados data, these are ice-equivalent sea-level, and in this case, we should actually have plotted this correctly (Figure 5) of Yokoyama et al. (2001). Yokoyama et al. (2001) corrected the dated coral reefs for GIA in their paper (Section 2.2 Glacio-hydro-isostatic modelling).

Barbados data was shown to be in the region of the world that is minimally affected by GIA effects, unless we understood correctly, and hence can be interpreted as representing ESL. It's a good idea for us to include this discussion in the text.

[Figure]

Reviewer: Insufficient/simplistic consideration of mechanisms of change and what is influencing your proxies—there was no real consideration of the (1) the hydrological balance of the palaeo-lakes (evaporation, precipitation, lake area and riverine inputs; the potential impact of the glacial re-routing of riverine inputs, etc. and how this may impact your proxies; (2) impact of the former proximal ice sheets on the glacio-isostatic response of the region and (3) tectonic setting of the region and the influence of active faults. The latter from the work of Vadar et al. (2014) on fault segments that developed during the Late Pleistocene.

Response: We disagree; we discussed this in lines 223-232 of the present manuscript. The $\delta$18O value of the Black Sea-Lake and Marmara Sea-Lake carbonate is shown to reflect the composite hydrological balance of the basin through the integration of inputs in the form of river and rain water and outputs in the form of evaporative processes. This is discussed before in Major et al. (2006), a manuscript that we referenced. The fact that the $\delta$18O of the Sofular Cave stays at -12 ‰ through the entirety of MIS 3 shows that the hydrological balance of the glacial period existed also through the entirety of MIS 3, cold with decreased evaporation but wet from continuous riverine input, leading to a positive hydrological framework. Perhaps we should discuss the hydrological framework more.

We also discussed the impact of the former proximal ice sheets on the glacio-isostatic response of the region in lines 328-329. We are happy to expand our discussion of the GIA effects on the region. Regardless, with or without considering GIA, RSL, would still be at the level we indicate is suggested by both the geochemical evidence and by the chirp profiles.

Discussion of the potential rerouting of riverine inputs is also a great idea but we believe this is irrelevant, because regardless of how the rivers rerouted or did not reroute, the $\delta$18O of the Sofular Cave reflects and shows that the hydrological balance, that was largely controlled by riverine inputs, stayed the same from beginning of MIS 3 to the glacial period in the Black Sea.

It's a great idea to include Vardar et al. (2014)'s work on the influence of active faults and we hope to include this discussion in the revised manuscript.

Reviewer: Mechanisms and assumptions - Chirp profiles: there is insufficient information here; what was the vertical resolution and what processing did you undertake and using what software, etc?

Response: This is great comment and we will include this information in the revised manuscript.

Reviewer: General integration of the literature. - Currently, the framing of this work within the wider context of the literature on the Black and Marmara Seas is poor. - Does not include Aksu et al. (2016) nor Yaltirak et al. (2002).

Response: Yes we agree and we will include much more of the prior work that was done on the Black Sea, including the works of Aksu et al. (2016) Aksu et al. (2016) and Yaltirak et al. (2002) in the revised manuscript.

Reviewer: Focus of the manuscript. If you wish to make this more of a sea level story, you will need greater consideration of the wider sea-level data available in MIS 3.

Response: We did our best, and we wanted to focus on the available ESL and ice volume equivalent data, not RSL from different regions of the world. If there are any additional ESL records available, we will gladly include them in Figure 1. We do think it may be a good idea to include a threshold line of 40 m from Pico et al. (2016) as another record; we did not originally as we wanted to plot actual physical records from the $\delta$18O and uplifted coral terrace data.

Reviewer: What is the focus of the glaciation, referring to the potential outflow of glacially dammed rivers and lakes, and especially the deglacial, e.g. the melting of the Northern hemisphere ice sheets filling the Black Sea, e.g., Chepalyga, 2007; Thom, 2010, Vidal, 2010, which in turn led to the outflow of brackish water to the Mediterranean via the Marmara Sea?

Response: We didn't include the effects of the glaciation as we didn't think it was necessary, given that we show that the hydrological balance must have stayed the same through MIS 3, both the lake levels in the Marmara and Black Seas which we know from the chirp profiles and also from the $\delta$18O data of the Sofular Cave stalagmites. We are happy to include this discussion in the text in the revised form of the manuscript.

Reviewer: How do your paleoshorelines compare to the lowstand terrace in Sea of Marmara at -85 m (Cagatay et al., 2009 and Aksu et al., 1999). These authors suggest that post 15 ka in Sea of Marmara, evaporation exceeded riverine and Black Sea inputs, how does this compare with your work?

Response: This is not relevant to our work as our work is about the MIS 2 and MIS 3 periods of the Black and Marmara Seas. We are happy to include this discussion as it would be relevant for the hydrology of the region.

Reviewer: There are other estimates for the depth of the sills (e.g. Major et al. 2002), who gives the elevation of the Dardanelles sill as 85 mbsl, which is consistent with the clinoforms.

Response: Exactly, we agree.

Reviewer: Do not discuss the GIA processes and how these might affect the connection between the various basins.

Response: Without a robust GIA model, we cant really discuss this in detail. We did the best we could in lines 328-329. We can give a more detailed and thought out discussion given indications of the location and extent of the Eurasian ice sheet from the work of Chepalyga (2007).

Reviewer: Discussion of the MIS 3 sea levels is incomplete and misses key references. The introduction is muddled (lines 51 to 68) and omits several well-constrained lines of evidence. The most obvious are the high resolution, continuous relative sea level records from the Red Sea from the work of Grant et al. (2012) and the Mediterranean

(Rohling et al., 2014), both of which are publically available.

Response: We did not originally include these data because they give RSL not ESL and comparing one RSL with another RSL does not make much sense. If the reviewer does think that we should include these data on RSL from the Red Sea, we are happy to include this in the introduction of the revised manuscript.

Reviewer: - Discussion of the lithofacies, marshes (Pico et al., 2017 and 2016 studies). The mention of these in the introduction is a little odd, given that none of this data is plotted or referred to in the text. Response: We discuss Pico et al. (2016) in the introduction and in the discussion. Perhaps it is a good idea to plot the threshold they indicate in our figure 1 as its an additional independent ESL record and also shows how much disagreement there is between all of the ESL and ice volume equivalent data.

References:

Aksu, A. E., R. N. Hiscott and C. Yaltirak: Early Holocene age and provenance of a mid-shelf delta lobe south of the Strait of Bosporus, Turkey, and its links to vigorous Black Sea outflow, Marine Geology, 380, 113-37, 2016. Bintanja, R., R. S. W. van De Wal and O. Johannes: Modelled atmospheric temperatures and global sea levels over the past million years, Nature, 437, 125-28, 2005. Chepalyga, A., The late glacial great flood in the ponto-caspian basin. The Black Sea flood Question: Changes in Coastline, Climate, and Human Settlement, V. Yanko-Hombach, A. Gilbert, N. Panin and P. Dolukhanov, Dordrecht, The Netherlands, Springer, 119-48, 2007. Major, C., S. Goldstein, W. Ryan, G. Lericolais, A. M. Piotrowski and I. Hajdas: The co-evolution of Black Sea level and composition through the last deglaciation and its paleoclimatic significance, Quatern. Sci. Rev., 25, 2031-47, doi:10.1016/j.quascirev.2006.01.032, 2006. Pico, T., J. X. Mitrovica, K. L. Ferrier and J. Braun: Global ice volume during MIS 3 inferred from a sea-level analysis of sedimentary core reciords in the Yellow River Delta, Quaternary Science Reviews, 152, 72-79, 2016. Shakun, J. D., D. W.

[Figure]

Lea, L. E. Lisiecki and M. E. Raymo: An 800-kyr record of global surface ocean $\delta$18O and implications for ice volume-temperature coupling, Earth and Planetary Science Letters, 426, 58-68, 2015. Vardar, D., K. Öztürk, C. Yaltirak, B. Alpar and H. Tur: Late Pleistocene-Holocene evolution of the southern Marmara shelf and sub-basins: middle strand of the North Anatolian fault, southern Marmara Sea, Turkey, Marine Geophysical Research, 35, 69-85, 2014. Yaltirak, C., M. Sakinc, A. E. Aksu, R. N. Hiscott, B. Galleb and U. B. Ulgen: Late Pleistocene uplift history along the southwestern Marmara Sea determined from raised coastal deposits and global sea-level variations, Marine Geology, 192, 283-305, 10.1016/S0025-3227(02)00351-1, 2002. Yanchilina, A. G., W. B. F. Ryan, J. McManus, P. Dimitrov, D. Dimitrov, K. Slavova and M. Filipova-Marinova: Reply to comment on, "Compilation of geophysical, geochronological, and geochemical evidence indicates a rapid Mediterranean-derived submergence of the Black Sea's shelf and subsequent substantial salinification in the early Holocene, Marine Geology, 407, 354-61, 2019. Yanchilina, A. G., W. B. F. Ryan, J. F. McManus, P. Dimitrov, D. Dimitrov, K. Slavova and M. Filipova-Marinova: Compilation of geophysical, geochronological, and geochemical evidence indicates a rapid Mediterranean-derived submergence of the Black Sea's shelf and subsequent substantial salinification in the early Holocene, Marine Geology, 383, 14-34, 10.1016/j.margo.2016.11.001, 2017. Yokoyama, Y., P. De Deckker, K. Lambeck, P. Johnston and K. Fifield: Sea-level at the Last Glacial Maximum: evidence from northwestern Australia to constrain ice volumes for oxygen isotope stage 2, Paleoceanography, Palaeoclimatology, Palaeoecology, 165, 281-97, 10.1016/S0031-0182(00)00164-4, 2001.

---

## Referee Comment (RC3) · Anonymous Referee #3 · 15 May 2019

General comments. Yanchilina et al. present a compilation of geochemical data that is used to infer periods of connection/disconnection within the Black Sea – Marmara Sea – Mediterranean Sea system during MIS 3. The authors also present seismic profiles in order to determine the level of the Marmara Sea Lake during MIS 3 and hence provide a RSL maximum boundary of 65-70 mbsl during MIS 3, with a possible intrusion of marine waters around 50 kyrs BP.

I have major concerns regarding the study (see below). In its present form (poor use of the existing literature, poor description of the methods, misuse of some set of previously published data, lack of confident chronological constraints for interpretation of

the seismic profiles. . .), I would not recommend publication of the study in Climate of the Past. With some more work, I think that such a contribution could be useful to the community.

1. Existing literature:

1.1. It is hard to understand which data are new, and which interpretations are novel compared to the existing literature (for example compared to Çagatay et al. (2009) regarding the level of the Marmara Sea Lake during MIS 1 – MIS 2 – MIS 3, or compared to Aloisi et al (2015) regarding the freshening of the Marmara Sea Lake from MIS 4 to MIS 3 and its link to a potential connection to the Black Sea lake during this time).

1.2. In general, the paper could benefit from a better use of the existing literature. For example, high-resolution qualitative paleosalinity records exist for the Black Sea during MIS 3 (Shumilovskikh et al., 2014; Wegwerth et al., 2016). They actually show changes (freshening and salinity rise) and could be integrated to the study to better document salinity changes in the Marmara Sea and connectivity between both basins during MIS 3. The authors may also integrate the new 87Sr/86Sr data from the Black Sea published in Ankindinova et al. (2019). Also, literature on the Gemlik Lake exists and some cores were dated and may be useful to the study (Gasperini et al., 2011; Taviani et al., 2014; Filikci et al., 2017). A review paper about the connectivity between basins in the Caspian Sea – Black Sea – Marmara Sea through time has been recently published and could be integrated to the study (Krijgsman et al., 2018).

2. Methods are poorly described. In some aspects, a better transparency regarding the methods, which dataset is new and which ones are from other sources would benefit to the paper:

2.1. The authors write they use a GIA model to provide highstand thresholds on RSL in the Black Sea and Marmara Sea (Lines 69-71). In the discussion it seems like they refer to their GIA results (lines 325-327). However, the authors do not provide any kind of information in the method section detailing the GIA model they used, the parameters,

the time period, etc. There is not even a figure showing these results for the Black Sea – Marmara Sea area during MIS 3. The authors simply acknowledge a researcher for generating GIA corrections (line 544). Did the author actually run a GIA model and present these results in the paper?

2.2. In this paper, the only new geochemical dataset (although it is an impressive one) is the 87Sr/86Sr record from ostracods/shells from the Marmara Sea. However, the authors do not mention that few of these data were already previously published in Vidal et al. (2010). Also, the ostracods and mollusks used for strontium analyses come from the sediments of 4 different cores (ITU-C1, ITU-C10, MD01-2430, MD01-2426). However the authors mention the age model for only one of these cores (core MD04-2430). What about ITU-C1, ITU-C10 and MD01-2426 age models?

2.3. MD01-2430 age model is poorly detailed. So it is difficult to evaluate its quality. The authors (lines 164-166) write they composed their own age model from 14C measurements made from pieces of mollusks from MD01-2430 (Vidal et al. 2010) and it seems that they tuned MD01-2430 ostracod $\delta$18O data (from Vidal et al, 2010) to Black Sea mollusk ones (Yanchilina et al., 2017) and Sofular Cave speleothems (Fleitmann et al., 2009; Badertscher et al., 2011). Isn't it a bit circular, as Black Sea data have been already tuned to Sofular Cave speleothem (Yanchilina et al., 2017)? Furthermore, similar patterns between $\delta$18O and 87Sr/86Sr changes in the Marmara Sea and in the Black Sea are used to infer periods of connection between basins later in the paper. If they are tuned to each other, then the inferred periods of connection between basins are not a result per se anymore, it is premise. To get around this problem, the data series shall be kept on independent age models (Blaauw, 2012). Please provide a detailed and transparent description of the age-depth models: 14C dates used (they seem to have been already published in Vidal et al. (2010) and Çagatay et al. (2015)), reservoir correction applied, software used, tuning, uncertainties...

2.4. Also, the age model of core MD01-2430 has been already published in Vidal et al. (2010) based on mollusk 14C dates, and later revised/refined in Çagatay et al. (2015)

using tephra markers and tuning to Greenland ice cores. Why did the authors need to redo this reliable age model? Also, as the MD01-2430 Ca% data are from Çagatay et al. (2015), are the MD01-2430 $^{87}Sr/^{86}Sr$ data and Ca% data shown in figure 3 with different age-depth models whereas they come from the exact same core?

3. Misuse and over interpretation of chloride data

3.1. The authors mention pore water chloride data from the Black Sea (Soulet et al., 2010) and Marmara Sea (Aloisi et al., 2015) to say that the Black Sea and Marmara Sea were fresh during MIS 3 and MIS 4 (Lines 112-114). This is not exactly what the original papers state. According to Soulet et al. (2010) the Black Sea was fresh ($\sim$1 psu) during the LGM. There is no mention for earlier periods. According to Aloisi et al. (2015) the Marmara Sea was not fresh but brackish (4psu) during MIS 3 and freshened at the end of MIS-4. The authors should stick to the conclusions of the original papers if they don't provide further material to modify/refine the original conclusions.

3.2. In figure 3, Black Sea and Marmara Sea pore water chloride data are shown as a function of the sediment age which is nonsensical. Diffusion and advection processes continuously alter the geochemical pore water composition, and hence there is no obvious relationship between the age of the sediment and its pore water geochemical composition (for example Manheim and Chan, 1974; Adkins and Schrag, 2003; Soulet et al., 2010; Aloisi et al., 2015). Basically one cannot interpret pore water geochemical data directly as a function of the age of the sediment as unfortunately the authors do at lines 214-219. These data should be dropped from figure 3.

4. The paper is mainly focused on MIS-3 and on the Marmara Sea. However, if one removes the pore water chloride data from figure 3 because of the reasons detailed just above, then only the MD01-2430 Ca% (Çagatay et al., 2015) extends back to MIS-3. The remaining Marmara Sea geochemical data ($\delta$18O [Vidal et al., 2010] and $^{87}Sr/^{86}Sr$ [Vidal et al., 2010; this study]) only extend back to MIS-2. Can these data be used to support the conclusions for MIS-3?

5. The similarity between the Black Sea CaCO3% (Nowaczyk et al., 2012) and Marmara Sea Ca% (Cagatay et al., 2015) during MIS-3 is interpreted here without clear support as reflecting the connection between the Black Sea and Marmara Sea. The authors do not show on figure 3 that a Black Sea CaCO3% peak (for example Major et al., 2002; Bahr et al., 2005) correlates to the Marmara Sea Ca% during Bølling oscillation (see figure 6 in Çagatay et al., 2015) just before the Marmara Sea reconnection to the Mediterranean. Also Yanchilina et al. (2017) suggest that the Black Sea was isolated from the Marmara Sea during the Bølling-Allerød. So, why would the Black Sea CaCO3% and Marmara Sea Ca% positive correlation suggest the connection of the two basins during MIS-3, but not during the Bølling oscillation? Can this correlation be solely explained by a common climatic conditions as it has been shown that the Black Sea carbonate peaks are driven by surface productivity during warmer oscillations (Major et al., 2002; Bahr et al., 2005; Shumilovskikh et al., 2014; Wegwerth et al., 2016). Conclusive evidence would come from the geochemical measurements of the Black Sea and Marmara Sea carbonates deposited during these events as the authors suggest (L362-364). In the absence of such data, I would suggest the authors to be more balanced in their interpretations or to provide additional and stronger/conclusive support for a Black Sea – Marmara Sea connection during MIS-3.

6. Chronological interpretations of the seismic profiles are not supported by direct dating, and thus if I acknowledge that a sedimentary structure that is below another one should be older, in my view it is very difficult to be conclusive regarding their exact age.

Specific comments:

L22: "connections and disconnections (partial or total)". It sounds odd, maybe remove "(partial or total)".

L24: "persistent freshwater lakes". The Marmara Sea salinity was reconstructed to be brackish (4psu; Aloisi et al., 2015)

L36: Remove "bi-polar", DO oscillations are North Hemispheric climatic features. The bipolar seesaw concept does not fit the context of the paragraph.

L70: "using a GIA model": Please describe it in the methods along with the parameters you used. The information is currently missing.

L77: "We show that the two lakes were freshwater". I am afraid that the authors do not provide original data showing that both lakes were fresh. Instead they are building up on many previous works. Please amend this sentence.

L87: These studies do not provide "accumulation". They only provide contents.

L91: Vidal et al. (2010) published $\delta$18O from ostracods not from "mollusks".

L92-103: The whole paragraph lacks support from the literature. Please cite for example Leng and Marshall (2004) for the mechanism and for example Major et al. (2002), Bahr et al. (2005), Shumilovskikh et al. (2014), Wegwerth et al. (2016) for its observation in the Black Sea.

L97: The Black Sea outflowing into the "small" Marmara Sea as an evidence for explaining the temporal synchronicity is unsupported material. Alternatively the regional climatic context could explain the temporal synchronicity.

L104-106: The way it is written is misleading. In 2010, Soulet et al. tested the age of the reconnection between 8500 and 9500 based on the available literature (Major et al., 2006) at this time. So the Yanchilina et al (2017) reference for the age of the reconnection should be dropped in this context. Similarly, Aloisi et al set the Marmara Sea reconnection to 14.7 ka based on Vidal et al. (2010) results. Furthermore the authors speak about pore water measurements in the Black Sea (Soulet et al., 2010) and in the Marmara Sea (Aloisi et al., 2015) without citing the original works.

L107: This statement does not reflect the reality. 1) Pore water chloride content of the sediment does not reflect a paleo-salinity, as advection-diffusion processes alter both the original concentrations and the age-depth relationship with the sediment. The

pore water chloride content profile is modern, not fossil. 2) The advection/diffusion model is actually a quantitative (within the limitations of the model and scenaris tested) reconstruction of the paleo-salinity. The authors state, however the opposite.

L109: "ppt": Part per trillion? "psu" instead?

L112-114: This is not exactly what the original publications says. Soulet et al. (2010) only tested salinity models not older than 20ka. So it cannot be inferred from this study that the Black Sea was fresh at times older than 20 ka. Instead other studies can be cited. Aloisi et al. (2015) tested salinity models spanning 130 ka, with a salinity decrease at the end of MIS-4, and the salinity value inferred for the Marmara Sea lake during MIS-3 is 4psu (brackish, not fresh).

L128: You may add Shumilovskikh et al. (2014).

L140: You may add Ankindinova et al. (2019).

L155-156: There is something odd. Are you sure you leached Sr to retrieve the Sr fraction?

L163-164: "Although the original 14C age measurements have been misplaced": Please clarify.

L169-171: Isn't it the purpose of the paper to infer when the Black Sea was overflowing into the Marmara Sea? From this sentence it seems it was a premise of your work.

L171: Supplementary data 1 is missing so one cannot evaluate the quality of the age model and data.

L175: Literature exits for Gemlik lake: Gasperini et al., 2011; Taviani et al., 2014 ; Filikci et al., 2017

L195: Same conclusions has been reached by Vidal et al. (2010).

L211-218: Naive or even nonsensical interpretations of pore water chloride profiles, as

there is no obvious relationship between the age of the sediment and its pore water composition.

L219: Marmara Sea $\delta18O$ and Sr records do not extend back to MIS 3 and the Black Sea record is very scarce for this period so conclusions of this sentence are unsupported for the Marmara Sea and weakly supported for the Black Sea.

L226-227: Direct proxies have been published for this period in the Black Sea (Shumilovskikh et al., 2014; Wegwerth et al., 2016).

L266-267: Chronological data are needed to infer such statement and Vardar et al. (2014) do not provide direct dating of these strata.

L276-277: "It is shown that the MIS-2 period corresponds to a large transgressive period following the MIS-3 low stand (ÇaÄ§atay et al. 2009)." Yet, Çagatay et al. (2009) suggest the opposite: a forced-regression during MIS-2 to -85mbsl that followed a MIS-3 Marmara Sea level at 70mbsl.

L306: These data do not extend back to MIS-3.

L284-286: I don't follow the authors. They write that core data are unavailable to strengthen the interpretation but "hence. . .".

L326-327: Where are these GIA results? Which results are the author referring to? No method has been described.

References: Adkins, J. F., & Schrag, D. P. (2003). Reconstructing Last Glacial Maximum bottom water salinities from deep-sea sediment pore fluid profiles. Earth and Planetary Science Letters, 216(1-2), 109-123.

Aloisi, G., Soulet, G., Henry, P., Wallmann, K., Sauvestre, R., Vallet-Coulomb, C., ... & Bard, E. (2015). Freshening of the Marmara Sea prior to its post-glacial reconnection to the Mediterranean Sea. Earth and Planetary Science Letters, 413, 176-185.

Ankindinova, O., Hiscott, R. N., Aksu, A. E., & Grimes, V. (2019). High-resolution

Sr-isotopic evolution of Black Sea water during the Holocene: Implications for reconnection with the global ocean. Marine Geology, 407, 213-228.

Badertscher, S., Fleitmann, D., Cheng, H., Edwards, R. L., Göktürk, O. M., Zumbühl, A., ... & Tüysüz, O. (2011). Pleistocene water intrusions from the Mediterranean and Caspian seas into the Black Sea. Nature Geoscience, 4(4), 236.

Bahr, A., Lamy, F., Arz, H., Kuhlmann, H., & Wefer, G. (2005). Late glacial to Holocene climate and sedimentation history in the NW Black Sea. Marine Geology, 214(4), 309-322.

Blaauw, M. (2012). Out of tune: the dangers of aligning proxy archives. Quaternary Science Reviews, 36, 38-49.

Çağatay, M. N., Eriş, K., Ryan, W. B. F., Sancar, Ü., Polonia, A., Akçer, S., ... & Bard, E. (2009). Late Pleistocene–Holocene evolution of the northern shelf of the Sea of Marmara. Marine Geology, 265(3-4), 87-100.

Çağatay, M. N., Wulf, S., Sancar, Ü., Özmaral, A., Vidal, L., Henry, P., ... & Gasperini, L. (2015). The tephra record from the Sea of Marmara for the last ca. 70 ka and its palaeoceanographic implications. Marine Geology, 361, 96-110.

Filikci, B., Eriş, K. K., Çağatay, N., Sabuncu, A., & Polonia, A. (2017). Late glacial to Holocene water level and climate changes in the Gulf of Gemlik, Sea of Marmara: evidence from multi-proxy data. Geo-Marine Letters, 37(5), 501-513.

Fleitmann, D., Cheng, H., Badertscher, S., Edwards, R. L., Mudelsee, M., Göktürk, O. M., ... & Kramers, J. (2009). Timing and climatic impact of Greenland interstadials recorded in stalagmites from northern Turkey. Geophysical Research Letters, 36(19).

Gasperini, L., Polonia, A., Çağatay, M. N., Bortoluzzi, G., & Ferrante, V. (2011). Geological slip rates along the North Anatolian Fault in the Marmara region. Tectonics, 30(6). Krijgsman, W., Tesakov, A., Yanina, T., Lazarev, S., Danukalova, G., Van Baak, C. G., ... & Bruch, A. (2018). Quaternary time scales for the Pontocaspian domain:

Interbasinal connectivity and faunal evolution. Earth-science reviews.

Leng, M. J., & Marshall, J. D. (2004). Palaeoclimate interpretation of stable isotope data from lake sediment archives. Quaternary Science Reviews, 23(7-8), 811-831.

Major, C. O., Goldstein, S. L., Ryan, W. B., Lericolais, G., Piotrowski, A. M., & Hajdas, I. (2006). The co-evolution of Black Sea level and composition through the last deglaciation and its paleoclimatic significance. Quaternary Science Reviews, 25(17-18), 2031-2047.

Major, C., Ryan, W., Lericolais, G., & Hajdas, I. (2002). Constraints on Black Sea outflow to the Sea of Marmara during the last glacial–interglacial transition. Marine Geology, 190(1-2), 19-34.

Manheim, F. T., & Chan, K. M. (1974). Interstitial Waters of Black Sea Sediments: New Data and Review: Water.

Nowaczyk, N. R., Arz, H. W., Frank, U., Kind, J., & Plessen, B. (2012). Dynamics of the Laschamp geomagnetic excursion from Black Sea sediments. Earth and Planetary Science Letters, 351, 54-69.

Schumilovskikh, L. S., Fleitmann, D., Nowaczyk, N. R., Behling, H., Marret, F., Wegwerth, A., & Arz, H. W. (2014). Orbital and millenial-scale environmental changes between 64 and 25 ka BP recorded in Black Sea sediments. Climate of the Past, 10, 939-945.

Soulet, G., Delaygue, G., Vallet-Coulomb, C., Böttcher, M. E., Sonzogni, C., Lericolais, G., & Bard, E. (2010). Glacial hydrologic conditions in the Black Sea reconstructed using geochemical pore water profiles. Earth and Planetary Science Letters, 296(1-2), 57-66.

Taviani, M., Angeletti, L., Çağatay, M. N., Gasperini, L., Polonia, A., & Wesselingh, F. P. (2014). Sedimentary and faunal signatures of the post-glacial marine drowning of the Pontocaspian Gemlik "lake"(Sea of Marmara). Quaternary international, 345,

11-17.

Vardar, D., Öztürk, K., YaltÄśrak, C., Alpar, B., & Tur, H. (2014). Late Pleistocene–Holocene evolution of the southern Marmara shelf and sub-basins: middle strand of the North Anatolian fault, southern Marmara Sea, Turkey. Marine Geophysical Research, 35(1), 69-85.

Vidal, L., Menot, G., Joly, C., Bruneton, H., Rostek, F., ÇaÄśatay, M. N., ... & Bard, E. (2010). Hydrology in the Sea of Marmara during the last 23 ka: Implications for timing of Black Sea connections and sapropel deposition. Paleoceanography, 25(1).

Wegwerth, A., Kaiser, J., Dellwig, O., Shumilovskikh, L. S., Nowaczyk, N. R., & Arz, H. W. (2016). Northern hemisphere climate control on the environmental dynamics in the glacial Black Sea "Lake". Quaternary Science Reviews, 135, 41-53.

Yanchilina, A. G., Ryan, W. B., McManus, J. F., Dimitrov, P., Dimitrov, D., Slavova, K., & Filipova-Marinova, M. (2017). Compilation of geophysical, geochronological, and geochemical evidence indicates a rapid Mediterranean-derived submergence of the Black Sea's shelf and subsequent substantial salinification in the early Holocene. Marine Geology, 383, 14-34.

---

## Author Comment (AC3) · 14 Jun 2019

Yanchilina et al. present a compilation of data that speak to lake levels and paleosalinity in the Sea of Marmara and the Black Sea during MIS 3. The authors argue that these datasets indicate low regional sea level ( 80 m below present day) over most or all of MIS 3. As the manuscript stands, it is not entirely clear what data is new, or which interpretations are novel as opposed to drawn from the existing literature. This

interpretation of regional sea level near -80 m over much of MIS 3 is based on pale-osalinity proxies and seismic profiles of lacustrine deltaic topsets, which are not usually considered proxies for sea level. In my view, the authors do not sufficiently explain how this data can be used to reconstruct regional sea level, and therefore I am not sure these conclusions can be drawn based on the observations presented, as detailed be low. Overall, I would recommend that the authors include more discussion of how the data presented can be used to understand sea level, in addition to ample references to literature which might support the proposed interpretation about paleosalinity and lake levels. I would caution the authors to be careful in drawing inferences about sea level from lake levels/salinity, and in doing so, it is necessary to include more references that would support the conclusions.

Specific comments

Reviewer: Salinity proxies are not considered a robust proxy for sea level. A variety of other processes are captured in these measurements including local climate effects such as precipitation patterns and temperature. In the methods section, it is not clear that the authors considered the extent to which these other variables may have domi-nated the signal measured in these geochemical proxies for salinity.

Response to reviewer:

Changes in salinity of marginal basins have been used to give an idea for regional and eustatic sea level in the past. We think that including more of this discussion will help explain how our method will additionally contribute to reconstruction of paleo sea level.

The first example is from the work of Van Daele et al. (2011). Here, the authors look at infills in the Gulf of Cariaco, a marginal basin, that is connected to the Caribbean Sea via a shallow 58-m-deep sill. The authors use a similar idea to ours, changes in regional sea level, to infer when the regional sea level was higher than the sill versus lower. When the regional sea level is higher, saline water intrudes into the Gulf of Cariaco, creating sedimentary infills. When the regional sea level is lower, saline water

does not intrude and sedimentary infills related to saline water intrusions do not occur.

The second example is from the work of Pico et al. (2016). The authors use sedimentary core analyses from a Yellow River Delta in the Bohai Sea of China to make inferences about a migrating paleoshoreline. Pico et al. (2016) use information from cores taken from the delta, to observe changes in inundation of the delta associated with first changes in regional sea level and second, make further to make conclusions about eustatic sea level after applying glacio-isostatic corrections.

The third example is using changes in the $\delta$18O record of the eastern Mediterranean Sea (Grant et al. 2012) and the Red Sea (Siddall et al. 2003) using a basin isolation concept to infer changes in regional sea level. The basin isolation concept features reduction of intrusion of saline water into a basin that is marginally connected to the global ocean; in the Red Sea from the description of Siddall et al. (2003), "Reduction of the strait profile by sea-level lowering decreases the exchange transport of water masses through the strait. This results in increased residence times of the water within the Red Sea, enhancing the effect of the high rate of evaporation (2.06 m yr-1) on properties in the Red Sea. The basin though amplifies the signal of sea level change, which are recorded in $\delta$18O values of foraminifera in Red Sea sediment cores."

For our manuscript, we want to make conclusions from a simple concept. The Black and Marmara Seas are connected to the Mediterranean by two shallow sills, both approximately shown to have been 80 meters below sea level during MIS 2 and 3 (the sill level connecting Marmara and Black Seas, Bosporus, is now 30 mbsl but during MIS 2 and 3 it was lower, at 80 mbsl). Now, marine water enters through these sills and these seas are saline because the regional sea level (from the Mediterranean side) is higher than the sills, hence, saline water has no option than to flow into both of these basins and make them seas as opposed to freshwater lakes. A good visual schematic is filling water in a bathtub. The water flows and if its higher than the level of the bathtub, it will overflow over the edges of the bath tub onto the floor. Same concept almost for the water inflowing into the Marmara and Black Seas, when the level of the Mediterranean

is lower, no salt water flows in and when the level of the Mediterranean is higher, salt water flows in. This is what we are trying to conclude from the data presented. The paper is not about making conclusions about lake/sea level in the Black Sea and the Marmara Seas, although it alludes to them, the paper is about the regional sea on the Mediterranean Side of the Marmara and Black Seas.

We agree that it would be great to have detailed regional and global sea level data from uplifted coral terraces but such information is largely absent and is mostly available for the glacial and postglacial and the Eemian. One of the reasons we are making and drawing these conclusions is because of the lack of regional and global sea level data that currently exists during MIS 3. We believe that knowing a constraint for regional sea level and an overall understanding that it was low, is a very important contribution for the paleo sea level community.

If there was any marine intrusion into the Marmara and Black Seas, there would have been a rise in salinity, during Marine Isotope Stage 3. There is not recorded a change in paleosalinity from all proxies available to date, most strongly the lack of change in the $\delta 18O$ of the Sofular Cave in the Black Sea, shown to reflect $\delta 18O$ of surface Black Sea water.

Reviewer: Furthermore, I found that the explanation regarding each method did not include sufficient information (or citations) about what values would be considered significant in indicating saline or freshwater conditions at present day or in the past. I think that the majority of the sea-level community would agree that salinity proxies are not as robust as geological markers such as dated evidence of a shoreline.

Response to reviewer:

Please see above. We would love to have an opportunity to more thoroughly discuss how proxies such as $\delta 18O$ and porewater Cl-, among others, indicate past changes in salinity.

One of the points we make in the manuscript is that there is limited information from uplifted coral terraces from the time periods of MIS 3 and hence, we offer an independent method towards providing what regional sea level must have been in the eastern Mediterranean. If current disagreement is giving indicators of eustatic sea level being as high as 40 mbsl to as low as 80 mbsl, there is sufficient disagreement regarding what sea level must have been. 40 m of global ocean is a very large piece of missing water volume. We argue and give sufficient evidence that it was on the lower side in the eastern Mediterranean.

Reviewer: Inspection of Figure 3 shows that a number of these salinity proxies do not include measurements during MIS 3 (or in both basins), so it seems difficult to draw a conclusion based on these. it seems plausible that a rise in sea level may not have lasted long enough to substantially change the saltwater content of this region for the duration of time required to be recorded in sedimentary deposits. It would strengthen the paper to include a discussion of what kind of timescales of marine inundation would be required to record an increase in salinity using these geochemical proxies. In fact, previous studies have suggested rapid episodes of sea level rise of 10-15 m in just 1-2 kyr during MIS 3 (Chappell, QSR, 2002).

Response to reviewer:

This is a great idea and we would be happy to include this discussion in the revised manuscript. It is a correct observation that not all of the proxies go into MIS 3. The strongest piece of evidence comes from the $\delta$18O of the Sofular Cave stalagmites with temporal resolution of 24 years (line 119 in the present manuscript). Hence, if salt water did enter, the saline intrusions would have had to be less than 24 years, which geologically speaking on the time scale of MIS 3, are practically insignificant.

Reviewer: Seismic reflection data constitutes the other form of evidence used by the authors to argue for continuously low sea level near -70 m. I found the figures showing the seismic reflection profiles to be confusing. Importantly, an erosional unconformity

exists above the MIS 3 lacustrine ages (which are not reported directly) and this would indicate it is possible that water levels were much higher later during MIS 3 and that any deposition during this time period was subsequently eroded during a major base-level fall leading into MIS 2. Besides the obvious unconformity, which calls into question whether these deltaic deposits represent the highest sea level over the MIS 3 period, I think a more substantial argument is needed for using deltaic topsets as a kind of sea-level record.

Response to reviewer:

While it is true there is a seismic unconformity related to sea level fall during MIS 2, and prior sediment, if it was present, would have been removed. Our point is that no such prior sediment existed. This is largely seen from the Figure 5 and Supplementary Material Figures 1-5. The chirp profiles show that MIS 3 and MIS 4 are comformable.

Reviewer: If the Black & Marmara Seas were not connected to the ocean, as argued in this study, then lake levels will not necessarily reflect sea level. My understanding is that lake levels may be largely controlled by precipitation and evaporation, and would not reflect local regional sea level.

Response to reviewer:

Lake levels are controlled by evaporation and precipitation unless they are seas which is exactly the case for the modern day Marmara Sea and Black Sea. The two basins are not lakes but are seas because of the saltwater intrusion into the two basins as a consequence of the two sills that connect them being shallower than the regional sea level from the eastern Mediterranean. Because the level of the sills is shallower, saltwater flows in. In the period before the two bodies of water connected with the Mediterranean Sea at 12500 and 9000 years ago (Sea of Marmara at 12500 and Black Sea at 9000), the two bodies of water were lakes. We use this observation to conclude that the level on the eastern Mediterranean side must have been shallower than the level of the sills, hence leading the two lakes to be controlled by precipitation and

evaporation.

Reviewer: A further issue is the treatment of ages used in this study. The ages within MIS 3 deposits (and associated uncertainties) are alluded to, but are actually not reported within the main text. Obviously, this information is crucial to interpreting water depths from this data at a particular instance of time. It would be very helpful to include the ages and location of cores on the figures showing the seismic reflection profiles.

Response to reviewer:

This is a great suggestion and we are happy to supply the 14C dates as a supplementary file.

Reviewer: As I have mentioned above, these proxies may not reliably record local relative sea level, however if we were to assume they do, I am still not convinced that these proxies would represent relative sea-level across the entire MIS 3 time interval (60-26 ka). The authors argue based on observations of deltaic deposits and freshwater conditions inferred from geochemical proxies that the sea level in this region was -80 m for most of MIS 3. However, in this manuscript, the authors also note that there may have been a marine incursion from 55-44 ka, although they later dismiss the likelihood of this possibility. This time period, from 50-40 ka, happens to the be the time period during which sea-level high stands are observed globally during MIS 3 (Cann, 2000, Caybioch & Aycliffe, 2001, Hanebuth et al., 2006, Simms et al., 2009, Pico et al., 2016, DaSilva et al., 2017).

Response to reviewer:

This is an intriguing suggestion and documentation. The authors strongly believe that the relative sea level must have remained below 80 mbsl during the period 55 to 44 kyr B.P. If the regional sea level on the side of the eastern Mediterranean was sufficiently high such as to overflow into the Sea of Marmara, there would be both indications in the seismic profiles, as there is of the documentation of the last intrusion by a uniform

layer and indications of a sapropel layer, that also formed with the most recent marine incursion into the Marmara Sea 12,500 years ago. There is indeed a disagreement of the two CaCO3 curves between the Black Sea and the Marmara Seas which most likely instead documents a disconnection of the two basins with the inflow from the Black Sea decreasing / coming to a stop and with the Sea of Marmara behaving independently for a few thousand years.

Reviewer: The authors seem tied to a conclusion about low sea level during MIS 3. I think this manuscript would be improved by presenting the data and multiple hypotheses, rather than focusing on a single interpretation of this dataset.

Response to reviewer:

That is a great idea but there aren't any other reasonable conclusions that the authors can reach given the data that we have. We are open to suggestions of what lack of saline entry into the Marmara and Black Seas could mean for regional sea level on the eastern Mediterranean side but really strongly see that all the data that we have points to a low regional sea level. We believe it is good to focus on one main message in our study as opposed to making several unfocused conclusions.

Reviewer: The important contribution of this study is to bring together a variety of observations about the nature of the connection between the Marmara and Black Sea. In my view, this dataset may or may not shed light on the relative sea level history in this region given the uncertainties associated with the methods used (paleosalinity proxies and deltaic topsets are not considered robust geologic sea level indicators). However, this dataset may be an important contribution to understanding the salinity history of these basins, and I think the authors should focus on this (and presenting possible hypotheses) rather than drawing much wider conclusions.

Response to reviewer:

This manuscript is not about the salinity history of the basins but is about regional sea

level on the eastern Mediterranean during MIS 3. See discussion above. The methods we use are not nontraditional as several other studies also use saline entry in and out of marginal basins to make conclusions about regional (and some go further) towards sea level through this time period. One of the main reasons to use, as the reviewer characterizes, "nontraditional" methods is because alternative methods using uplifted coral terraces do not give robust sea level indications through this time period. We use our data to argue for a low sea level during this period which we think is very important, both for development of GIA models and to understand fluctuations of past climate change.

Line by line comments

Reviewer: Line 17 – Delete "the" in front of MIS 3

Response to reviewer:

This is a great suggestion and we would remove "the" in the future version of the revised manuscript.

Reviewer: Line 17 – what do you mean by persistent?

Response to reviewer:

We meant that the climate was variable; we agree that this word is confusing and is better removed and/or changed to variable climate change.

Reviewer: Line 26 – eustatic is not in caps

Response to reviewer:

We agree and in the revised manuscript we would have eustatic with a lower case.

Reviewer: Line 38 – replace 'elevations" with "estimates"

Response to reviewer:

We agree with this suggestion and in the revised manuscript would replace 'elevations'

with estimates.

Reviewer: Line 39 – delete "existence of"

Response to reviewer:

We agree with this suggestion and in the revised manuscript would delete "existence of,:

Reviewer: Line 40 – delete "factors that control"

Response to reviewer:

We disagree with this and think that it makes sense to leave "factors that control" as we discuss what controls the changes in ice sheet growth and collapse.

Reviewer: Line 41 – delete "additionally"

Response to reviewer:

We disagree with deleting "additionally" as this is the second reason why its important to understand changes in ice volume.

Reviewer: Line 43 – relative sea level not in caps

Response to reviewer:

We agree and in the revised manuscript will make this change.

Reviewer: Line 45 – "The earth is moving" is imprecise language. This sentence should be written

Response to reviewer:

We agree that this sentence is not well written and agree to rewrite it to, "The relative sea level changes because of . . ."

Reviewer: Line 47 – delete "the"

Response to reviewer:

We agree with this suggestion and will change this in the future revised manuscript.

Reviewer: Line 50 - I think you mean methods... while there are some theoretical differences in approaches it is largely the tools that you are referring to

Response to reviewer:

Yes, we agree to change this to methods in the revised manuscript.

Reviewer: Line 53 – (3) is missing from this list

Response to reviewer:

Thank you for this note, we agree that the (4) should be (3).

Reviewer: Line 56 – grammar/wording in sentence "Isolating.." - you are not isolating ice volume from these records, you are estimating the contribution of global ice volume to changes in seawater delta $\delta$18O.

Response to reviewer:

We agree with this suggestion and in the revised manuscript will change from "isolating" to "estimating the contribution of global ice volume."

Reviewer: Line 65 - This is not exactly what the study showed - it used estimates from the Pico et al., 2016 paper to infer the source of ice loading in North America

Response to reviewer:

The Pico et al. (2016) study reading directly from their introduction reevaluates previous geological records from the Albemarle Embayment using a new set of GIA calculations to show that the peak equivalent global mean sea level (GMSL) during MIS 3 reached a peak of -40 m which is what we wrote.

Reviewer: Line 69 – This may be misleading because the authors do not run a GIA

model to look at this question specifically. Rather authors should say they use previous studies of GIA in this region to estimate the effect of GIA-related SL changes

Response to reviewer:

This is true and we will remove the discussion of using a GIA model from the introduction. We previously did run a GIA model but decided not to include the discussion because that made the paper more complicated.

Reviewer: Line 91 – Confusing sentence

Response to reviewer:

Yes, we agree and will change this sentence to, "CaCO3 is a proxy that can be used to infer connectivity between water bodies." This sentence is more clear and direct.

Reviewer: Line 97 – need citation for volume of Marmara and river inflow

Response to reviewer:

We agree with these additions. The Sea of Marmara is 3378 km3 (Ünlüata et al. 1990) in volume and the Black Sea is 544,000 km3 in volume (Lane-Serff et al. 1997). The modern river flow into the Black Sea is 350 km3 yr-1 (Ünlüata et al. 1990) and the net modern flow of water from the Black Sea into the Sea of Marmara is 300 km3 yr-1 (Latif et al. 1991, Oszoy et al. 1995, Polat and Tugrul 1996). The modern river inflow into the Sea of Marmara is 5.80 km3yr-1 (EIE 1993). We will include these citations

Reviewer: Line 100 - confusing wording. Do you mean the lakes must have been alkaline in order to explain these accumulations? And why would you get this kind of accumulation during warming? Could you explain this concept more clearly?

Response to reviewer:

We agree and this should be better clarified. CaCO3 maxima during Bolling/Allerod and Preboreal periods in the Black Sea have been previously interpreted to be mainly

composed of authigenically precipitated calcite, which occurs within lakes as a consequence of photosynthetic utilization of $CO_2$ and resultant calcium carbonate supersaturation in the water column during growing season (Major et al. 2002, Leng and Marshall 2004, Bahr et al. 2005, Soulet et al. 2011a).

Reviewer: Line 108 – Can you cite references for what values would be considered freshwater or saline? What is the range of values you might find globally? What values are considered significant?

Response to reviewer:

The amount of the chloride ion in the water directly reflects salinity through the equation: salinity (ppt) = 0.0018066 * Cl- with Cl- concentration being in mg/L (Foundriest Environmental, 2014). NOAA defines freshwater salinity is near 0 ppt (parts per thousand) while those that are considered brackish, mixture of fresh and marine, will range between 0.5 and 35 ppt (Foundriest Environmental, Inc. 2014, National Oceanic and Atmospheric Administration 2017). Based on this range, fresh and marine waters are identified. For a lake to be fresh, its salinity must then be near 0. Given this information, it makes sense to instead refer to our lakes as brackish not freshwater and in the revised manuscript we will make this alteration.

Reviewer: Line 130 - Is this new data in this study? Authors should make that very clear, here, and in introduction. What kind of new data are the authors presenting?

Response to reviewer:

Great idea and in the revised text we would make it clear what 87Sr/86Sr data we are presenting is new and which was published before.

Reviewer: Line 134 – need reference!

Response to reviewer:

We agree with this suggestion. 87Sr/86Sr ratio of seawater has evolved over geologic

time, but value has remained constant during the period that is focused on in this study (Henderson et al. 1994). Sr isotope ratios of continental water bodies are known to reflect the geographical origin as different geological formations will have their own unique strontium isotope composition (e.g. Stein et al. 1997, Krom et al. 1999). This is used to differentiate sources of water into basins. Biological and inorganic precipitates that form in the basin of interest record the changes, which result from input and mixing of different water sources, provided the different water sources have sufficiently different isotope ratios. Seawater has currently 87Sr/86Sr value of 0.709155 for the global ocean (Henderson et al. 1994) and 0.709157 for the Aegean Sea and 0.709150 for the Sea of Marmara (Major et al. 2006). The average 87Sr/86Sr of the water feeding the Black Sea is 0.7088. The marine water also has a concentration of 30 times higher of dissolved Sr (Broecker and Peng 1982, Palmer and Edmond 1989). The modern salinity of the Black Sea surface water is 18 ppt and it represents roughly 1 to 1 mixture between freshwater and marine water. We will include the above explanation into the revised manuscript.

Reviewer: Line 137 - Can you cite something? And what is the change you would expect per volume % change of marine water?

Response to reviewer:

The citations we will plan to include are in the above description.

Reviewer: Line 140- How would you expect this value to change through time?

Response to reviewer:

Great suggestion; we would include how this proxy was previously used by both Yanchilina et al. (2017) and by Major et al. (2006) to indicate entry of marine water into the Black Sea in the early Holocene transforming it from a freshwater lake to a sea.

Reviewer: Line 165 – Is this standard practice? Can you cite a study that has used a similar method?

Response to reviewer:

Tuning geochemical records of one series of measurements to another well dated series of measurements is indeed standard practice and has been used by many. A good review of tuning paleoclimate proxies is given by Blaauw (2012). This method is not perfect but is the best known alternative for constructing age models for records that do not have their own perfectly known independent age model, which given the uncertainties associated with 14C dating and changing reservoir ages in lake basins, is used for the work of this manuscript.

Reviewer: Line 195-196 – Confusing sentence

Response to reviewer:

We agree. In the revised manuscript we would change it to, "The Sea of Marmara, given its much smaller volume relative to the Black Sea, must have outflowed to the Mediterranean Sea."

Reviewer: Line 196 – "contest" means to disagree, I think you mean we present data to support the hypothesis that the two lakes were freshwater.

Response to reviewer:

Yes, that is correct. Instead of contest, we would change it to "show."

Reviewer: Line 204- How exactly is this signal changing over this time period? Can you reference a figure that would show this?

Response to reviewer:

Great point. We need to add a sentence to show the difference and refer to the respective figure. The figure is figure 3. The Ca concentration of the Sea of Marmara is high and the CaCO3 concentration of the Black Sea is low. The two data series bifurcate here for this time period.

Reviewer: Line 207 - There are no references for this. How do you know that every time there is marine incursion there will be a sapropel layer? How long would you expect that a marine incursion should last in order to develop this layer? Also probably should briefly define what a sapropel is for readership

Response to reviewer:

Great point. We do need to make references that note the formation of the sapropel layer in the Sea of Marmara and the Black Sea. The sapropel in the Black Sea is recorded to have formed both during the Eemian (Shumilovskikh et al. 2013), when the global eustatic sea level was close to that of the modern global eustatic sea level and during the early Holocene (Ross and Degens 1974, Lamy et al. 2006). The sapropel in the Marmara Sea is recorded to have formed during the Bolling Allerod period (Vidal et al. 2010), because the global sea level was rising at this period as the planet was coming out of its last glacial period.

Sapropel is sediment with high organic content and is understood to have formed in the Black and Marmara Seas as a consequence of changes in surface nutrients and anoxia (Emerson and Hedges 1988).

We also would welcome to add a discussion section specifying what would be the expected temporal resolution for there to be a formation of the sapropel layer. I would argue that the marine intrusion would have to be for 50-100 years, this being resolution of accumulated sediments, and of significant enough marine volume for a sapropel to have developed.

Reviewer: Line 207 – delete "evidence of", delete "here"

Response to reviewer:

This is a great suggestion and we would make this alteration in the revised manuscript.

Reviewer: Line 211 - How do you know this is the reason? You need a little more explanation connecting the observations with the mechanism for producing these.

Response to reviewer:

I think we here explain the mechanism succinctly and sufficiently. "...as a result of the earlier connection of the Marmara Sea with the global ocean and higher post-connection salinity, leading to an earlier diffusion of marine water into the previously lacustrine sediments." Basically because the marine intrusion occurred earlier for the Marmara Sea, the saline water had more time to diffuse into the sediments relative to the Black Sea. We could add however that because the salinity in the Sea of Marmara is higher than in the Black Sea, the diffused trend also has higher porewater Cl- for the Sea of Marmara relative to the Black Sea.

Reviewer: Line 215 – Confusing discussion

Response to reviewer:

We are not sure exactly what is confusing here as what we mean to say is that if there was marine entry into Marmara and Black Seas during MIS 3, this should be reflected in the porewater Cl- measurements of this age. The porewater Cl- values would not be trending towards freshwater for both of the seas.

Reviewer: Line 218 – Replace "with the light" with "in light of"

Response to reviewer:

We think we should delete this phrase and finish the sentence after "...and we discuss this in this paper."

Reviewer: Line 236 – Delete "Index"

Response to reviewer:

We will think about this suggestion and take it into consideration upon revising the manuscript.

Reviewer: Line 246 – Replace "indicates: : :" with "suggesting a freshwater environment"

Response to reviewer:

We agree with this alteration and will incorporate it into the revised manuscript.

Reviewer: Line 284 – However there is a large SL fall after MIS 3, so you might expect a lot of erosion during this time. No deposition during MIS 3/2 is simply absence of evidence rather than evidence in itself.

Response to reviewer: We agree with this point and need to update our discussion to clarify our interpretation of the figure.

Reviewer: Line 310 – grammar in sentence "It remains possible: : :"

Response to reviewer:

Yes, we agree and we would change the wording to, "It remains possible that strong freshwater flux out of the Sea of Marmara through the Dardanelles Strait would keep out marine water by a few meters. In such a scenario, RSL would still have to be low, maybe a few meters above the paleodepth of the Dardanelles Strait at most."

Reviewer: Line 321 – RSL index is not a conventional term, you might mean RSL indicator. Essentially you want to say that this set of data suggests RSL ranged from X-X mbsl. But, do you mean the seismic data suggest this? These are not really ocean shorelines; the elevation of lake shorelines cannot be used to represent a sea level shoreline. I would be very skeptical of interpreting sea level from lake levels because there are a number of other factors that control lake elevation including precipitation, for example. If you are inferring this from the geochemical data can you explain exactly where those numbers (70-90 m) come from (depth of sill?). This needs to be much clearer, as this is a main conclusion of the paper. However, as the paper stands I am not sure what data this sea level is inferred from, and cannot determine how robust this conclusion is.

Response to reviewer:

We are using three indicators to draw our main conclusion with the illustration below: (1) geochemical data; (2) seismic lake shorelines; (4) depth of the paleosills; (3) seismic imaging of Gemlik lake. The geochemical data suggests no marine entry occurred. The seismic shorelines suggest that the lake was consistently at 80 mbsl with a positive hydrological balance leading to outflow from the Black Sea and Sea of Marmara to the Mediterranean Sea. If the Mediterranean Sea was connected, then it would have flowed into the Sea of Marmara and the Black Sea and the level of the paleoshorelines would have been higher and the water chemistry would also have changed. None of this is observed suggesting saltwater did not enter into the Sea of Marmara and into the Black Sea. With this we draw the conclusion that the RSL on the eastern Mediterranean side of the Dardanelles strait, had to remain below 80 mbsl. Illustration below:

Also, sill depths are approximate here, just to indicate our argument and our point.

Reviewer: Line 374 – This is uncited and it is not clear how the authors estimated this.

Response to reviewer:

This is part of the conclusions, perhaps we should leave the 10-20 m estimations out of this discussion and just say lower. We ran a GIA model before and this was the result given by the model.

Reviewer: Line 583 – Figure 3 Based on this figure I feel confused about the conclusion. Which proxies are supposed to be the same in both water bodies if there is freshwater? Also, there is not a record during most of MIS 3 for all proxies in both water bodies, so it is not clear that it makes sense to compare these proxies (for example Sea of Marmara does not have Cl measurements during MIS 3 and near does Sr for either water bodies).

Response to reviewer:

In the revised manuscript we will clarify which of the values on the figure indicate freshness and which marine.

While not all proxies go into MIS 3, the porewater Cl- (b), left side certainly does and goes into MIS 4 as well. It is the red diamonds in the figure.

Reviewer: Line 605 – Figure 4. What is the point of this figure? Is this an area that matters or are we interpreting a paleo shoreline? I am not sure I understand the significance of a shoreline in Gemlik Bay as this is far from the connection between the Marmara and Black Sea.

Response to reviewer:

This is a perched lake. It is a lake that is located in the Sea of Marmara (Figure 4). The point of this figure is to show that there was no accumulation of marine sediments into this lake during MIS 4 and 3 and this lake was disconnected from the Sea of Marmara. This lake is separated from the deeper parts of the Sea of Marmara by a ridge. This is just an additional piece of evidence indicating the level of the lake remained at its paleoshoreline of 50-60 m while the local lake level in the Sea of Marmara was 80 mbsl. The lake was fed by river water and discharged into the Sea of Marmara before it connected with the Sea of Marmara at 12,000 years (Vardar et al. 2014). After that, sediments that accumulated were marine and were also dated by a series of cores (Gasperini et al. 2011, Taviani et al. 2014, Filicki et al. 2017). We need to include the latter to strengthen our argument.

Reviewer: Line 641 – Are there cores taken in the profile in Fig 5a? What are the dates? It would be great to include the dates next to the location of these. Are these figures from another study? I think that this could be said at the beginning of the figure caption. It is confusing that these might be from two separate papers so the terms are not the same. For example, an erosional surface is shown in Fig 5b, but these aren't highlighted in Fig 5a. It would be great if the terminology in the captions were consistent across both figures.

Response to reviewer:

There were no cores taken from profile 5a. Both 5a and 5b are from another study and this is said in the figure caption, "(a) Succession of superimposed prograding clinoforms adopted from a previously published illustration (Fig. 11c) (Aksu et al. 2002)..." and "adopted from an earlier published XIX profile retrieved from R/V Hydrograph in 1998 (Genov 2015)."

We will update the terms in the revised paper to make the two figures more easy to follow and complement each other.

Review references:

Chappell, J. (2002). Sea level changes forced ice breakouts in the Last Glacial cycle: New results from coral terraces. Quaternary Science Reviews, 21(10), 1229–1240.

Cann, J. H. (2000). Late Quaternary Paleosealevels and Paleoenvironments Inferred From Foraminifera, Northern Spencer Gulf, South Australia. The Journal of Foraminiferal Research, 30(1), 29–53. http://doi.org/10.2113/0300029

Cabioch, G., & Ayliffe, L. K. (2001). Raised Coral Terraces at Malakula, Vanuatu, Southwest Pacific, Indicate High Sea Level During Marine Isotope Stage 3. Quaternary Research, 56, 357–365. http://doi.org/10.1006

da Silva Salvaterra, A., Cesar, R., Figueira, L., & Mahiques, M. M. De. (2017). Evidence of an Marine Isotope Stage 3 transgression at the Baixada Santista , south-eastern Brazilian coast. Brazilian Journal of Geology, 47(December), 693–702. http://doi.org/10.1590/2317-4889201720170057 Simms, A. R., DeWitt, R., Rodriguez, A. B., Lambeck, K., & Anderson, J. B.

(2009). Revisiting marine isotope stage 3 and 5a (MIS3-5a) sea levels within the northwestern Gulf of Mexico. Global and Planetary Change, 66(1–2), 100–111. http://doi.org/10.1016/j.gloplacha.2008.03.014
Hanebuth, T. J. J., Saito, Y., Tanabe, S., Vu, Q. L., & Ngo, Q. T. (2006). Sea levels during late marine isotope stage 3 (or older?) reported from the Red River delta (northern Vietnam) and adjacent regions. Quaternary International, 145–146, 119–134. http://doi.org/10.1016/j.quaint.2005.07.008

Response references:

Aksu, A. E., R. N. Hiscott, D. Yasar, F. I. Isler and S. Marsh: Seismic stratigraphy of late Quaternary deposits from the southwestern Black Sea shelf: evidence for non-catastrophic variations in sea-level during the last ∼10,000 yr, Marine Geology, 190, 61-94, 2002. Bahr, A., F. Lamy, H. Arz, H. Kuhlmann and G. Wefer: Late glacial to Holocene climate and sedimentation history in the NW Black Sea, Marine Geology, 214, 309-22, 2005. Blaauw, M.: Out of tune: the dangers of aligning proxy archives, Quaternary Science Reviews, 36, 38-49, 2012. Broecker, W. and T.-H. Peng, Tracers in the Sea, Palisades, NY, Lamont-Doherty Geological Observatory, 1982. EIE: Turkeyi akarsularinda sediment tasinim miktarlari, Elektrik Isleri Erud Idaresy Genel Mudurlugu, 93-59, 1993. Emerson, S. and J. I. Hedges: Processes controlling the organic carbon content of open ocean sediments, Paleoceanography, 3, 621-34, 1988. Filicki, B., K. Kadir Eris, N. Cahatay, A. ASabuncu and A. Polonia: Late glacial to Holocene water level and climate changes in the Gulf of Gemlik, Sea of Marmara: evidence from multi-proxy data, Geo-Mar Letters, 37, 501-13, 2017. Foundriest Environmental, I. (2014). "Conductivity, Salinity and Total Dissolved Solids, Fundamentals of Environmental Measurements." from http://www.fondriest.com/environmental-measurements/parameters/waterquality/conductivty-salinity-tds. Gasperini, L., A. Polonia, M. N. Cagatau, G. Boruluzzi and V. Ferranted: Geological slip rates along the North Anatolian Fault in the Marmara region, Tectonics, 30, 2011. Genov, I.: Seismostratigraphy and the last Black Sea level changes, Geologie, 68, 1419-24, 2015. Grant, K. M., E. J. Rohling, M. Bar-Matthews, A. Aaylon, M. Medina-Elizalde, C. B. Ramsey, C. Satow and A. P. Roberts: Rapid coupling between ice volume and polar temperature over the past 150,000 years, Nature, 491, 744-47, 10.1038/nature11593,

2012. Henderson, G. M., D. J. Martel, R. K. O'Nions and N. J. Shackleton: Evolution of seawater Sr-87/Sr-86 over the last 400-ka - the absence of glacial interglacial cycles, Earth and Planetary Science Letters, 128, 1994. Krom, M. D., A. Michard, R. A. Cliff and K. Strohle: Sources of sediment to the Ionian Sea and western Levantine Basin of the Eastern Mediterranean during S-1 sapropel times, Marine Geology, 160, 45-61, 1999. Lamy, F., H. W. Arz, G. C. Bond, A. Bahr and J. Pätzold: Multicentennial-scale hydrological changes in the Black Sea and northern Red Sea during the Holocene and the Arctic/North Atlantic Oscillation, Paleoceanography, 21, 10.1029/2005PA001184, 2006. Lane-Serff, G. F., E. J. Rohling, H. L. Bryden and H. Charnock: Post-glacial connection of the Black Sea to the Mediterranean and its relation to the timing of sapropel formation, Paleoceanography, 12, 169-74, 10.1029/96PA03934, 1997. Latif, M. A., E. Ozsoy, T. Oguz and Ü. Ünlüata: Observations of the Mediterranean inflow into the Black Sea, Deep-Sea Research, 38, S711-S23, 1991. Leng, M. J. and J. D. Marshall: Palaeo- climate interpretation of stable isotope data from lake sediment archives, Quatern. Sci. Rev., 23, 811-31, doi:10.1016/j.quascirev.2003.06.012, 2004. Major, C., S. Goldstein, W. Ryan, G. Lericolais, A. M. Piotrowski and I. Hajdas: The co-evolution of Black Sea level and composition through the last deglaciation and its paleoclimatic significance, Quatern. Sci. Rev., 25, 2031-47, doi:10.1016/j.quascirev.2006.01.032, 2006. Major, C. O., W. B. F. Ryan, G. Lericolais and I. Hajdas: Constraints on Black Sea outflow to the Sea of Marmara during the last glacial-interglacial transition, Marine Geology, 190, 19-34, 2002. National Oceanic and Atmospheric Administration: Salinity and Total Dissolved Salts, 2017. Oszoy, E., M. A. Latif, S. Tugrul and U. Unluata, Exchanges with the Mediterranean, fluxes and boundary mixing processes in the Black Sea, CIESM Sci. Ser. 1, 1995. Palmer, M. R. and J. M. Edmond: The strontium budget of the modern ocean, Earth and Planetary Science Letters, 92, 11-26, 1989. Pico, T., J. X. Mitrovica, K. L. Ferrier and J. Braun: Global ice volume during MIS 3 inferred from a sea-level analysis of sedimentary core reciords in the Yellow River Delta, Quaternary Science Reviews, 152, 72-79, 2016. Polat, C. and S. Tugrul, Chemical exchange between the Mediterranean and

Black Sea via the Turkish Straits, CIESME Sci. Ser. 2, 1996. Ross, D. A. and E. T. Degens, Recent sediments of the Black Sea. The Black Sea - Geology, Chemistry and Biology, E. T. Degens and D. A. Ross, Tulsa, Amer. Assoc. Petrol. Geol. Mem., 20, 183-99, 1974. Shumilovskikh, L. S., F. Marret, D. Fleitmann, H. W. Arz, N. R. Nowaczyk and H. Behling: Eemian and Holocene sea-surface conditions in the southern Black Sea: Organic-walled dinoflagellate cyst record from core 22-GC3, Marine Micropaleontology, 101, 146-60, 2013. Siddall, M., E. J. Rohling, A. Almogi-Labin, C. Hemleben, D. Meischner, I. Schmelzer and D. A. Smeed: Sea-level fluctuations during the last glacial cycle, Nature, 423, 853-58, 10.1038/nature01690, 2003. Soulet, G., G. MeÌ₋not, V. Garreta, F. Rostek, S. Zaragosi, G. Lericolais and E. Bard: Black Sea "Lake" reservoir age evolution since the Last Glacial — Hydrologic and climatic implications, Earth Planet. Sc. Lett., 308, 245-58, doi:10.1016/j.epsl.2011.06.002, 2011a. Stein, M., A. Starinsky, A. Katz, S. L. Goldstein, M. Machlus and A. Schramm: Strontium isotopic, chemical, and sedimentologi- cal evidence for the evolution of Lake Lisan and the Dead Sea, Geochim. Cosmochim. Acta, 61, 3975-92, 1997. Taviani, M., L. Angeletti, M. N. Cagatay, L. Gasperini, A. Polonia and F. P. Wesselingh: Sedimentary and faunal signatures of the post-glacial marine drowning of the Pontocaspian Gemlik "lake" (Sea of Marmara), Quaternary International, 345, 11-17, 2014. Ünlüata, Ü., T. Oguz, M. A. Latif and E. Özsoy, On the physical oceanography of the Turkish Straits. The Physical Oceanography of Sea Straits, L. J. Pratt, Deventer, The Netherlands, Kluwer, 25-60, 1990. Van Daele, M., A. van Welden, J. Moernaut, C. Beck, F. Audermard, J. Sanchez, F. Jouanne, E. Carrillo, G. Malavé, A. Lemus and M. De Batist: Reconstruction of Late-Quaternary sea- and lake-level changes in a tectonically active marginal basin using seismic statigraphy: The Gulf of Cariaco, NE Venezuela, Marine Geology, 279, 37-51, 2011. Vardar, D., K. Öztürk, C. Yaltirak, B. Alpar and H. Tur: Late Pleistocene-Holocene evolution of the southern Marmara shelf and sub-basins: middle strand of the North Anatolian fault, southern Marmara Sea, Turkey, Marine Geophysical Research, 35, 69-85, 2014. Vidal, L., G. Ménot, C. Joly, H. Bruneton, F. Rostek, M. N. ÇaÄ�§atay, C. Major and E. Bard: Hydrology

in the Sea of Marmara during the last 23 ka: Implications for timing of Black Sea connections and sapropel deposition, Paleoceanography, 25, 10.1029/2009PA001735, 2010. Yanchilina, A. G., W. B. F. Ryan, J. F. McManus, P. Dimitrov, D. Dimitrov, K. Slavova and M. Filipova-Marinova: Compilation of geophysical, geochronological, and geochemical evidence indicates a rapid Mediterranean-derived submergence of the Black Sea's shelf and subsequent substantial salinification in the early Holocene, Marine Geology, 383, 14-34, 10.1016/j.margo.2016.11.001, 2017.

Please also note the supplement to this comment:
https://www.clim-past-discuss.net/cp-2019-30/cp-2019-30-AC3-supplement.pdf

[Figure]

**Fig. 1.**

---

## Author Comment (AC4) · 14 Jun 2019

Yanchilina et al., "Lack of marine entry into Marmara and Black Sea-lakes indicate low relative sea level during MIS 3 in the Northeastern Mediterranean". Summary: This manuscript presents new 87Sr/86Sr data from the Sea of Marmara (the Black Sea Sr/Sr data was published in Yanchilina et al., 2017) that provides constraints on the timing of the reconnection of the Sea of Marmara and the Black Sea with the

Mediterranean. The authors also present seismic/reflection profiles from the region, although only one is new (Chrip profile of Gemlik Bay, figure 2). I have significant concerns about the work presented here (for example; the age model, interpretation of the proxies within the wider context of the region, mechanisms and assumptions: : :) and feel the submission of this manuscript is premature. The work could make a good contribution to the field, however, it requires more thought, a clearer focus and greater attention to detail to do so. As such, I would not recommend the acceptance of the manuscript for publication in Climate of the Past in its current format.

GENERAL COMMENTS

Reviewer: The main issues that must be addressed are: (1) Greater integration of the literature: Currently, the framing of this work within the wider context of the literature on the Black Sea and Sea of Marmara is poor (for example, there is no mention of the work of Aksu et al., 2016, Yaltirak et al 2002 etc.). Additionally, the selection and representation of sea-level data for the interval is inadequate (see section 3.2 below for examples).

Response to reviewer:

In the revised form of the manuscript we will make a bigger effort to integrate more of the prior work and will make sure to add very important references currently missing such as that of Aksu et al. (2016) and Yaltirak et al. (2002).

We thank for the note in regarding to develop a more thorough selection and representation of sea-level data for the interval and in the revised manuscript, will also make every attempt to make it better.

Reviewer: (2) Focus of the manuscript (and title): the data here is not a sea-level record per se, rather a record of marine incursion(s) into the Sea of Marmara and the Black Sea with rising sea level during the deglaciation. A key outstanding question that this paper helps to address, is when these transitions occurred (although your age model

needs considerable work, see below). What you have here are very valuable independent constraints for the region on rising deglacial sea level (87Sr/86Sr data). I would suggest trimming (or removing?) most of the introductory sea level discussion (which is far too general), and focus on the new 87Sr/86Sr data and seismic/reflection/Chirp profiles (most of which have been relegated to the supplement). If you wish to make this more of a sea level story, you will need greater consideration the wider sea-level data available in MIS 3 (see below). The focus on MIS 3 is also a little odd, given the data you present. In figure 3, your Cl data suggests increasing salinity but you do not really attempt to untangle the mechanisms driving this (see section 3.3). Similarly, you do not comment on the fluctuations in the Ca (Sea of Marmara) and CaCO3 (Black Sea) records. Is there a connection to the Dansgaard-Oeschger events that are characteristic of the time period?

Response to reviewer:

We want this to be a sea level paper not an incursion of marine water paper. In the revised manuscript and we will answer further below, we will make a better effort to consider the wider sea-level data available during MIS 3.

We do discuss the changes in porewater Cl- and attribute this, as from previously published work, to diffusion of Cl- in the sediments after the marine incursion first into the Sea of Marmara at 12,500 and then into the Black Sea at 9,300 years B.P. The concept is pretty straightforward but in the revised manuscript we will add more discussion on diffusion of porewater Cl- and how this reflects paleo-salinity.

We agree that we need to enhance the discussion on the fluctuations in Ca and CaCO3. The accumulation of CaCO3, reflected in the XRF-measured Ca as well, is a climatic response of lake chemistry. CaCO3 maxima during Bolling/Allerod and Preboreal periods in the Black Sea have been previously interpreted to be mainly composed of authigenically precipitated calcite, which occurs within lakes as a consequence of photosynthetic utilization of CO2 and resultant calcium carbonate supersaturation in the water column during growing season (Major et al. 2002; Leng and Marshall 2004; Bahr et al. 2005; Soulet et al. 2011a). This happened during warm periods of Bolling/Allerod and the Preboreal. Dansgaard/Oeschger events in the Northern Hemisphere are rapid warming episodes that occurred typically in the manner of decades. Hence, since Dansgaard/Oeschger events are also warming events, the increases in CaCO3 accumulation must indeed been related to their occurrence.

Reviewer: (3) General lack of rigour: The three major aspects that need addressing are; 3.1 There is insufficient information in your methods (in particular, your age model is not described in sufficient detail, see below);

Response to reviewer:

Thank you for the comment, we hope our response below will address this comment.

Reviewer: (3.1.1) Age model. I am unconvinced that your age model is reasonable (or robust) given that methodology and rationale/mechanistic relationships are currently poorly explained and justified. Either more detail is needed (including reporting of the 14C data, see comment below, see *), clearly stating how your age model was constructed (a supplementary figure would be useful), or I suggest adopting a more direct approach using fact that the Black Sea is the dominant source water source for Sofular Cave (you do mention this in lines 167 to 169). In the latter, you could use the well dated (i.e., precise, radiometric dates) of the speleothem to constrain your age model (which you can then check and/or refine with your 14C determinations, especially given your records extends beyond the limits of radiocarbon). Although you do not have a high resolution, continuous 18O record for either the Black Sea or Sea of Marmara to tune to the speleothem record, you do have very nicely resolved Ca (Sea of Marmara) and CaCO3 (Black Sea) records that have good signal correspondence to the Sofular Cave 18O record (and the Dansgaard-Oeschger (D/O) events more generally, e.g., Rasmussen et al., 2014) – warmer – more blooms and higher Ca/CaCO3 production etc. I would also suggest increasing the vertical exaggeration of the Sofular 18O record

in figure 3 to help the reader.

Response to reviewer:

We agree that we need to describe the age models for all of the records presented in Figure 3. The $\delta$18O and $\delta$13C record of the Black Sea mollusks was tuned to the Sofular Cave stalagmites and the ages for the 87Sr/86Sr of the Black Sea were interpolated from the calculated calendar ages of the mollusks given that the measurements were done on the same set of cores. The age model for the $\delta$18O of Sea of Marmara was adopted from Vidal et al. (2010). The age model for our measurements for 87Sr/86Sr of the Sea of Marmara is described in lines 163-164. The age model for the CaCO3 was adopted from (Nowaczyk et al. 2012) and the age model for Ca was adopted from (Çaħatay et al. 2015). Perhaps as the reviewer suggests, one way to put all the models into one would be to tune all the records to the Sofular Cave. We agree with the vertical exaggeration and would make this change in the revised figure. We should also, having looked at figure 3, should add the CaCO3 record for the Black Sea that covers MIS 2 and the Holocene.

Reviewer: Eyeballing your record of Ca (Sea of Marmara) and CaCO3 (Black Sea), I would place the transition from low to higher values (and more square-wave signal) that you currently have at 55 ka at about 48 ka. This would shift most of your records to younger ages (this has the upshot of making your data more consistent with other sea level records – see below)

Response to reviewer:

The authors are confused about what transition and what low and high values the reviewer is referring to.

Reviewer: I would suggest an additional step of incorporating the stratigraphic relationships into you age modelling, and assessment of the age uncertainties of the final age model (e.g., using Bacon, or the deposition model in OxCal). This is not vital but it

would allow you to provide some age uncertainty estimates for the marine incursion(s) into the Sea of Marmara and the Black Sea.

Response to reviewer:

Thank you for the suggestion, this would have been great for our earlier paper that exactly focuses on the marine incursions into the Black and Marmara Seas. Here, the marine incursion is not relevant for the MIS 3 period but perhaps this would be a great idea to still add more certainty to the age model and we will take this into consideration.

Reviewer: It must be an oversight that you do not fulfil the minimum reporting requirements for the 14C dates (e.g., Stuvier and Polach, 1977; Mook and van der Plicht, 1999; Millard, 2014). I know these are not new dates but I would expect as a minimum for you to list the 14C dates you use, the source for any R correction you use, nor the calibration curve/programme you use. A supplementary figure with the age-depth relationship would be a good addition.

Response to reviewer:

We are more than happy to include all of the measurements we present in Figure 3 along with the 14C measured dates. If the manuscript gets accepted as a paper in Climate in the Past, we will submit the measurements as a supplementary spreadsheet.

Reviewer: (3.1.2) Chirp profiles: There is insufficient information here (lines 175 to 182). What was the vertical resolution? What processing did you undertake (and using what software) etc.?

Response to reviewer:

This is great comment and we will include this information in the revised manuscript.

Reviewer: 3.2 As mentioned in (1) above, there is poor integration with other available data and literature. For example, there are some cursory attempts to couch this work within the literature but most are related to the proxies and seismic/reflection profiles

presented.

Response to reviewer:

In the revised manuscript we will make a stronger effort to integrate more of the available data and prior work, both for global sea level and that relating to the Black Sea and Sea of Marmara research.

Reviewer: Other issues that should be considered include: (3.2.1) What is the impact of glaciation (e.g., the potential outflow of glacially dammed rivers and lakes) and especially the deglacial, e.g., the melting of Northern hemisphere ice sheets filling the Black Sea, e.g., Chepalyga, 2007, Thom, 2010, Vidal et al., 2010, which in turn led to the outflow of brackish water to the Mediterranean via the Marmara Sea? Also, how do your palaeo shorelines compare to the lowstand terrace in Sea of Marmara at -85 m (ÇaÄet al., 2009, Asku et al., 1999)? These authors suggest that post 15 ka in Sea of Marmara, evaporation exceeded riverine and Black Sea inputs – how does this compare to your work?

Response to reviewer:

We didn't include the effects of the glaciation as we didn't think it was necessary, given that we show that the hydrological balance must have stayed the same through MIS 3, both the lake levels in the Marmara and Black Seas which we know from the chirp profiles and also from the $\delta18O$ data of the Sofular Cave stalagmites. We are happy to include this discussion in the text in the revised form of the manuscript.

Our paleoshorelines are at the same level of the lowstand terrace in Sea of Marmara.

Prior to the Sea of Marmara being connected with the Mediterranean at 12500, there was indeed an arid period during which the lake level decreased. This is not exactly super relevant for our work as we make an effort to focus on MIS 3 not deglaciation during MIS 2. The only point of relevance is the increase in CaCO3 accumulation in the Sea of Marmara, what occurs during warm periods in Marmara and Black Sea lakes.

Reviewer: (3.2.2) There are other estimates for the depth of the sills (e.g., Major et al., 2002 – a co-author? - gives the elevation of the Dardanelles sill as -85 5 m, which is consistent with the clinoforms). In addition, you do not discuss (or model) the GIA processes and how these might affect the connection between the various basins.

Response to reviewer:

So the first point is that the elevation of the Dardanelles sill is consistent with the clinoforms so yes, this is the case and thank you for the agreement with us. Second, we did do GIA modeling in previous versions of the paper but we decided not to add them here as we want to focus on reporting RSL and not ESL.

Perhaps it is good to give GIA more consideration as we do mention some conclusions about ESL and more discussion of the location and distribution of Eurasian ice sheets during MIS 3.

Reviewer: (3.2.3) The discussion of MIS 3 sea levels is incomplete and misses some key references. There also seems to be some confusion/conflation of relative- (RSL) and eustatic sea level (ESL), and ice volume equivalent throughout the manuscript. You explicitly state the difference between RSL and ESL (in lines 44 to 45) and yet the discussion of the various means of determining sea level (and what, RSL, ESL or icevolume equivalent) in the introduction is muddled (lines 51 to 68) and omits several well-constrained lines of evidence (as does figure 1). The most obvious are the high resolution, continuous relative sea level records from the Red Sea (e.g., Grant et al., 2012) and the Mediterranean (Rohling et al., 2014) – both of which are publically available. In more detail, in lines 51 to 57, you also have a list of 1 to 4 methods for deriving past changes in sea level, but 3 is missing. These are subsequently returned to in the discussion but only in a very superficial manner. My comments are: Isotopic methods and deconvolution of the 18O signal: The oxygen isotope ratio of marine sediments can be used to infer past sea levels (as a first order approximation) using the relationship between the 18O of the mineral precipitated (e.g., foraminiferal calcite) and the

processes governing the hydrological cycle (and thus sea level). The relative contribution of global ice volumes and temperature to foraminiferal oxygen isotopes is complex and subject to substantial uncertainties and several attempts to unravel this are available – e.g., through various assumptions and/or modelling (e.g., Bintanja et al.,2005, Shakun et al., 2015, as mentioned by the authors but see also de Boer et al., 2014, Waelbroeck et al., 2002, Elderfield et al., 2012). However, it should be noted that in all these reconstructions, the global ice volume component is comprised of both a terrestrial component AND any floating ice. Changes in the former would contribute to both 18O AND sea level, whereas changes in any floating ice would ONLY change the 18O record and not sea level. In other words, reconstructed changes in global ice volumes may not be equivalent to changes in sea level (e.g., Rohling et al., 2017). In figure 1, this could easily be fixed by changing the axis labels of (a) and (b) to ice volume. The authors might also consider adding the Elderfield et al. (2012) and/or the de Boer et al. (2014) reconstructions. Coral terraces: these are RSL records, unless they have been corrected for glacioisostatic (GIA) processes, in which case they do provide ESL constraints. In figure 3, the data is incorrectly referenced and there are no details on your(?) GIA corrections to the data. Please clarify. In figure 3, you plot (some) of the coral Barbados (for other Barbados sea-level data within the time period e.g., Bard et al., 1990, Fairbanks et al., 2005), and (some) Huon Peninsula (Yokoyama et al., 2001 but see also Cutler et al., 2003) data along with the 18O record of Shackleton (1987). The latter is an ice volume equivalent sea level not ESL. There are other coral sea-level records available that span some of your time window – e.g., Chappell, 2002, Cutler et al., 2003; or the very recent Yokoyama et al 2018 paper from the Great Barrier Reef. The authors might consider a wider selection of coral data: : :?

Lithofacies, salt marshes etc. where former sea levels are reconstructed using a modern analogue for the relationship between the indicator and sea level at the time of formation (note, the Pico et al., 2017 study is a GIA modelling studies of previously published sea-level reconstructions, assuming different ice models – i.e., variations in the volume and the spatial extent of the former ice sheets – as well as Earth rheologies). The mention of these in the introduction is a little odd, given that none of this data is plotted nor referred to in the text. The Pico et al. (2016, 2017) studies are returned to in lines 339 to 340 but with very little analysis. Given the above, the introductory section does not sit well with the data you present, and the discussion of MIS 3 sea-level data is poor, in particular how this fits with your data. I would significantly trim this unfocused sea level portion of the introduction and discussion and refocus your manuscript on the timing of the (re)connection of the Sea of Marmara and Black Sea to the Mediterranean.

Response to reviewer:

(1) Response to not including sea level records from Grant et al. (2012) and Rohling et al. (2014). These are both RSL not ESL and we wanted to focus our discussion on ESL.

(2) Listing 4 as opposed to 3 and only discussing three is a typo on our part, it is three methods that we wanted to list.

(3) Going back to discussing prior ESL records in the discussion was our best attempt but we hope to do better in the revised manuscript.

(4) Reply to comment about $\delta$18O signal: Not exactly. Bintanja et al. (2005) make separate figures for ice volume (Figure 3a) of their paper and global mean sea level (Figure 3b). They specifically state, "the main strength of our method is that it yields long and mutually consistent records of surface air temperature, ice volume, and global sea level by separating the ice-sheet and deep-water parts of the marine $\delta$18O signal."

Similarly, Shakun et al. (2015) also separately discuss sea level and ice volume contributions to the $\delta$18O records. Their methodology is described in section 3.3 of their paper. Hence, our figures 1a and 1b are correct in referencing ESL and not ice volume.

Having said this, we agree that we should more thoroughly discuss the difference between ice volume and sea level in our introduction sections.

(5) Reply about coral terrace records: These are not RSL records.

With the exception of the Barbados data, these are ice-equivalent sea-level, and in this case, we should actually have plotted this correctly (Figure 5) of Yokoyama et al. (2001). Yokoyama et al. (2001) corrected the dated coral reefs for GIA in their paper (Section 2.2 Glacio-hydro-isostatic modelling). We are happy to include this discussion in the revised text.

Barbados data was shown to be in the region of the world that is minimally affected by GIA effects, unless we understood correctly, and hence can be interpreted as representing ESL. It's a good idea for us to include this discussion in the text.

We wanted to focus on coral terraces that have been corrected for GIA and hence did not include other data.

(6) Reply to comment about Lithofacies. We discuss Pico et al. (2016) in the introduction and in the discussion. Perhaps it is a good idea to plot the threshold they indicate in our figure 1 as its an additional independent ESL record and also shows how much disagreement there is between all of the ESL and ice volume equivalent data.

Reviewer: 3.3 Insufficient/simplistic consideration of mechanisms of change and what is influencing your proxies – there was no real consideration of the: (3.3.1) hydrological balance of the palaeo-lakes (evaporation, precipitation, lake area and riverine inputs; the potential impact of the glacial re-routing of riverine inputs etc.) and how this might impact your proxies. In addition, there seemed to be some confusion of the systematics of 18O in marine, lake and speleothems environments. (3.3.2) impact of the former proximal ice sheets on the glacio-isostatic response of the region; (3.3.3) tectonic setting of the region and the influence of active faults (e.g., fault segments that developed during the Late Pleistocene, for example, see Vardar et al., 2014 and references therein).

Response to reviewer:

We disagree; we discussed this in lines 223-232 of the present manuscript. The $\delta$18O value of the Black Sea-Lake and Marmara Sea-Lake carbonate is shown to reflect the composite hydrological balance of the basin through the integration of inputs in the form of river and rain water and outputs in the form of evaporative processes. This is discussed before in Major et al. (2006), a manuscript that we referenced. The fact that the $\delta$18O of the Sofular Cave stays at -12 ‰ through the entirety of MIS 3 shows that the hydrological balance of the glacial period existed also through the entirety of MIS 3, cold with decreased evaporation but wet from continuous riverine input, leading to a positive hydrological framework. Perhaps we should discuss the hydrological framework more.

We also discussed the impact of the former proximal ice sheets on the glacio-isostatic response of the region in lines 328-329. We are happy to expand our discussion of the GIA effects on the region. Regardless, with or without considering GIA, RSL, would still be at the level we indicate is suggested by both the geochemical evidence and by the chirp profiles.

Discussion of the potential rerouting of riverine inputs is also a great idea but we believe this is irrelevant, because regardless of how the rivers rerouted or did not reroute, the $\delta$18O of the Sofular Cave reflects and shows that the hydrological balance, that was largely controlled by riverine inputs, stayed the same from beginning of MIS 3 to the glacial period in the Black Sea.

It's a great idea to include Vardar et al. (2014)'s work on the influence of active faults and we hope to include this discussion in the revised manuscript.

Reviewer: 3.4 Writing: some careless mistakes in the manuscript – for example, poor/incomplete referencing (e.g., line 34) and repetition (line 323 to 324 is immediately repeated as the next sentence).

Response to reviewer:

These are great observations and we will make the requested changes in the revised for of the manuscript.

TECHNICAL CORRECTIONS

Reviewer: References: Greater care with referencing needed. Please check manuscript. For example, line 34: "Members 2006" should be "EPICA Community Members"

Response to reviewer:

We agree with this suggestion, this was a problem with our endnote referencing and we will fic this.

Reviewer: Figures: Figure 1: incorrect axis labelling, poor selection of available data, inaccurate/ incomplete referencing of data in the caption.

Response to reviewer:

We disagree, our axis labeling is correct and we decided to select the data only with ESL records and not all the available RSL as well as its harder to directly compare RSL from our region to all the other RSLs as they are RSL for a reason, impacted by GIA.

Reviewer References:

Aksu, A.E. et al., 2016. Early Holocene age and provenance of a mid-shelf delta lobe south of the Strait of Bosporus, Turkey, and its links to vigorous Black Sea outflow. Mar. Geol., 380, 113-137.

Aksu, A.E., et al., 1999. Oscillating Quaternary water levels of the Marmara Sea and vigorous outflow into the Aegean Sea from the Marmara Sea-Black Sea drainage corridor. Mar Geol, 153, 275– 302.

Bard, E., et al., 1990. Calibration of the 14C timescale over the past 30,000 years using mass spectrometric U-Th ages from Barbados corals. Nature, 345, 405- 410.

Bintanja, R., et al., 2005. Modelled atmospheric temperatures and global sea levels over the past million years. Nature 437, 125–128.

ÇaÄet al., 2002. Late Glacial–Holocene paleoceanography of the Sea of Marmara: timing connections with the Mediterranean and the Black Seas. Mar Geol., 167, 191–206.

Chappell, J. 2002. Sea level changes forced ice breakouts in the Last Glacial Cycle: New results from coral terraces, Quat. Sci. Rev., 21, 1229-1240.

Cutler, K.B., et al., 2003. Rapid sea-level fall and deep-ocean temperature change since the last interglacial period. Earth and Planetary Science Letters, 206, 253-271.

Boer et al., 2014. Persistent 400,000-year variability of Antarctic ice volume and the carbon cycle is revealed throughout the Plio-Pleistocene. Nature Comms. doi: 10.1038/ncomms3999.

Elderfield, H., et al. 2012. Evolution of ocean temperature and ice volume through the mid-Pleistocene climate transition. Science 337, 704–709.

Fairbanks, R. G. et al. 2005. Radiocarbon calibration curve spanning 0 to 50,000 years BP based on paired Th-230/U-234/U-238 and C-14 dates on pristine corals. Quat. Sci. Rev. 24, 1781-1796.

Grant, K.M., et al., 2012. Rapid coupling between ice volume and polar temperature over the past 150,000 years. Nature, doi:10.1038/nature11593

Major, C., et al., 2002. Constraints on Black Sea outflow to the Sea of Marmara during the last glacial- interglacial transition. Mar. Geol. 190:19–34.

Millard, A. 2014. Conventions for reporting radiocarbon determinations. Radiocarbon 56, 555– 559.

Mook, W. G. & van der Plicht, J. 1999. Reporting 14C activities and concentrations. Radiocarbon 41, 227–239.

Rasmussen, S.O., et al., 2014. A stratigraphic framework for abrupt climatic changes during the Last Glacial period based on three synchronized Greenland ice-core records: refining and extending the INTIMATE event stratigraphy. Quat. Sci. Rev., 106, 14-28.

Rohling, J.R., et al., 2014. Sealevel and deep-sea-temperature variability over the past 5.3 million years. Nature, 508, doi:10.1038/nature13230.

Potter E-K., et al., 2004. Suborbital-period sea-level oscillations during marine isotope substages 5a and 5c. Earth and Planetary Science Letters 225: 191-204.

Shakun, J.D., et al., 2015. An 800-kyr record of global surface ocean 18O and implications for ice volume-temperature coupling. Earth and Planetary Science Letters, 426, 58-68.

Stuiver, M. & Polach, H. A. 1977. Reporting of 14C data. Radiocarbon 19, 355–363.

Vardar, D., et al., 2014. Late Pleistocene–Holocene evolution of the southern Marmara shelf and sub-basins: middle strand of the North Anatolian fault, southern Marmara Sea, Turkey. Mar Geophys Res, 35, 69–85.

Waelbroeck, C., et al., 2002. Sea-level and deep water temperature changes derived from benthic foraminifera isotopic records. Quat. Sci. Rev., 21, 295-305.

YaltÄ'srak, C. Met al., 2002. Global sea-level variations and raised coastal deposits along the southwestern Marmara Sea during the last 224,000 years Mar. Geol., 190, 283-305.

Yanchilina, A.G., et al., 2017. Compilation of geophysical, geochronological, and geochemical evidence indicates a rapid Mediterranean-derived submergence of the Black Sea's shelf and subsequent substantial salinification in the early Holocene. Mar. Geol. 383, 14-34.

Yokoyama, Y. et al., 2001. Coupled climate and sea-level changes deduced from Huon

Peninsula coral terraces of the last ice age. Earth and Planetary Science Letters 193, 579-587.

Yokoyama, Y. et al 2018. Rapid glaciation and a two-step sea level plunge into the Last Glacial Maximum. Nature, 559, doi.org/10.1038/s41586-018-0335-4.

Response references:

Aksu, A. E., R. N. Hiscott and C. Yaltirak: Early Holocene age and provenance of a mid-shelf delta lobe south of the Strait of Bosporus, Turkey, and its links to vigorous Black Sea outflow, Marine Geology, 380, 113-37, 2016. Bahr, A., F. Lamy, H. Arz, H. Kuhlmann and G. Wefer: Late glacial to Holocene climate and sedimentation history in the NW Black Sea, Marine Geology, 214, 309-22, 2005. Bintanja, R., R. S. W. van De Wal and O. Johannes: Modelled atmospheric temperatures and global sea levels over the past million years, Nature, 437, 125-28, 2005. Çaħatay, M. N., S. Wulf, Ü. Sancar, A. Özmaral, L. Vidal, P. Henry, O. Appelt and L. Gasperini: The tephra record from the Sea of Marmara for the last ca. 70 ka and its paleoceanographic implications, Marine Geology, 361, 96-100, 10.1016/j.margeo.2015.01.005, 2015. Leng, M. J. and J. D. Marshall: Palaeo- climate interpretation of stable isotope data from lake sediment archives, Quatern. Sci. Rev., 23, 811-31, doi:10.1016/j.quascirev.2003.06.012, 2004. Major, C., S. Goldstein, W. Ryan, G. Lericolais, A. M. Piotrowski and I. Hajdas: The co-evolution of Black Sea level and composition through the last deglaciation and its paleoclimatic significance, Quatern. Sci. Rev., 25, 2031-47, doi:10.1016/j.quascirev.2006.01.032, 2006. Major, C. O., W. B. F. Ryan, G. Lericolais and I. Hajdas: Constraints on Black Sea outflow to the Sea of Marmara during the last glacial-interglacial transition, Marine Geology, 190, 19-34, 2002. Nowaczyk, N. R., H. W. Arz, U. Frank, J. Kind and B. Plessen: Dynamics of the Laschamp geomagnetic excursion from Black Sea sediments, Earth and Planetary Science Letters, 351-352, 54-69, 10.1016/j.epsl.2012.06.050, 2012. Pico, T., J. X. Mitrovica, K. L. Ferrier and J. Braun: Global ice volume during MIS 3 inferred from a sea-level analysis of sedimentary core reciords in the Yellow River Delta, Quaternary Science Reviews, 152, 72-79,

2016. Shakun, J. D., D. W. Lea, L. E. Lisiecki and M. E. Raymo: An 800-kyr record of global surface ocean $\delta$18O and implications for ice volume-temperature coupling, Earth and Planetary Science Letters, 426, 58-68, 2015. Soulet, G., G. MeÌ₳not, V. Garreta, F. Rostek, S. Zaragosi, G. Lericolais and E. Bard: Black Sea "Lake" reservoir age evolution since the Last Glacial — Hydrologic and climatic implications, Earth Planet. Sc. Lett., 308, 245-58, doi:10.1016/j.epsl.2011.06.002, 2011a. Vardar, D., K. Öztürk, C. Yaltirak, B. Alpar and H. Tur: Late Pleistocene-Holocene evolution of the southern Marmara shelf and sub-basins: middle strand of the North Anatolian fault, southern Marmara Sea, Turkey, Marine Geophysical Research, 35, 69-85, 2014. Vidal, L., G. Ménot, C. Joly, H. Bruneton, F. Rostek, M. N. Çaħatay, C. Major and E. Bard: Hydrology in the Sea of Marmara during the last 23 ka: Implications for timing of Black Sea connections and sapropel deposition, Paleoceanography, 25, 10.1029/2009PA001735, 2010. Yaltirak, C., M. Sakinc, A. E. Aksu, R. N. Hiscott, B. Galleb and U. B. Ulgen: Late Pleistocene uplift history along the southwestern Marmara Sea determined from raised coastal deposits and global sea-level variations, Marine Geology, 192, 283-305, 10.1016/S0025-3227(02)00351-1, 2002. Yokoyama, Y., P. De Deckker, K. Lambeck, P. Johnston and K. Fifield: Sea-level at the Last Glacial Maximum: evidence from northwestern Australia to constrain ice volumes for oxygen isotope stage 2, Paleoceanography, Palaeoclimatology, Palaeoecology, 165, 281-97, 10.1016/S0031-0182(00)00164-4, 2001.

Please also note the supplement to this comment:
https://www.clim-past-discuss.net/cp-2019-30/cp-2019-30-AC4-supplement.pdf

---

## Author Comment (AC5) · 14 Jun 2019

General comments. Yanchilina et al. present a compilation of geochemical data that is used to infer periods of connection/disconnection within the Black Sea – Marmara Sea - Mediterranean Sea system during MIS 3. The authors also present seismic profiles in order to determine the level of the Marmara Sea Lake during MIS 3 and hence provide a RSL maximum boundary of 65-70 mbsl during MIS 3, with a possible intrusion

of marine waters around 50 kyrs BP. I have major concerns regarding the study (see below). In its present form (poor use of the existing literature, poor description of the methods, misuse of some set of previously published data, lack of confident chronological constraints for interpretation of C1, I would not recommend publication of the study in Climate of the Past. With some more work, I think that such a contribution could be useful to the community.

1. Existing literature:

Reviewer: 1.1. It is hard to understand which data are new, and which interpretations are novel compared to the existing literature (for example compared to Çagatay et al. (2009) regarding the level of the Marmara Sea Lake during MIS 1 – MIS 2 – MIS 3, or compared to Aloisi et al (2015) regarding the freshening of the Marmara Sea Lake from MIS 4 to MIS 3 and its link to a potential connection to the Black Sea lake during this time).

Response to reviewer:

The new data that we do our best to clearly show is our 87Sr/86Sr measurements and the seismic profiles from Gemlik lake. What is specifically novel in our study is our interpretation of both our results and re-interpretation of what was previously published. Neither Çaħatay et al. (2009) nor Aloisi et al. (2015) make the conclusion about RSL, they only focus on the interpretation of the lake data. We strongly believe this work should be interesting and relevant to the paleo sea-level community.

Reviewer: 1.2. In general, the paper could benefit from a better use of the existing literature. For example, high-resolution qualitative paleosalinity records exist for the Black Sea during MIS 3 (Shumilovskikh et al., 2014; Wegwerth et al., 2016). They actually show changes (freshening and salinity rise) and could be integrated to the study to better document salinity changes in the Marmara Sea and connectivity between both basins during MIS 3. The authors may also integrate the new 87Sr/86Sr data from the Black Sea published in Ankindinova et al. (2019).

Also, literature on the Gemlik Lake exists and some cores were dated and may be useful to the study (Gasperini et al., 2011; Taviani et al., 2014; Filikci et al., 2017). A review paper about the connectivity between basins in the Caspian Sea – Black Sea – Marmara Sea through time has been recently published and could be integrated to the study (Krijgsman et al., 2018).

Response to reviewer:

We agree, many of these studies should be better integrated into ours. Given that we initially wrote the manuscript in 2016, were focused on revising the 2016 version, when some of these studies were not yet published and hence, we did not know of their existence.

We think the work of Shumilovskikh et al. (2014) should be integrated into our study. Their paper goes back to 63 kyr B.P. and they show a freshening of the Black Sea lake from 63 kyr B.P. to 19 kyr B.P., which agrees with our work.

The work of Wegwerth et al. (2016) also should be integrated and we thank the reviewer greatly for this suggestion. The authors also show and confirm that the Black Sea only freshened from MIS 4 through MIS 3 and into MIS 2 supporting our work.

The work of Ankindinova et al. (2019) is fascinating for the Black and Marmara Sea community but it is not very relevant for our work as we focus on the Black and Marmara connection to the global ocean during MIS 3 not the early Holocene connection of the Black Sea with the Sea of Marmara.

The authors were not aware that there was existing core data for Lake Gemlik, this is very interesting. The MARM05-124 core from the study of Gasperini et al. (2011), cores GE123, GE124 and the corecatcher GE126 from the study of Taviani et al. (2014), and core M13-08 from the work of Filicki et al. (2017) were recovered from approximately 70 mbsf of water depth. Cores MARM05-124, GE123, GE124, and GE126 indicate that below the second erosion surface, there was brown lacustrine mud of

dates 13,950 uncalibrated 14C years and older (Gaspirini et al., 2011 and Taviani et al., 2014). The erosion that occurred after that is indicated by the sandy gray in these cores that followed the deposition of lake sediments fed by rivers. All the sediments that are observed to have been below those belonging to the brown mud must then be of age older, belonging to MIS 3 and 2. Hence, the authors would like to thank the reviewer for providing us this information that also we think supports our work and interpretation.

In Vadar et al. (2014), the authors indicate that Gemlik and Bandirma lakes were fed by rivers during MIS 2 and 3 and had lake levels of -50.3 and – 60.5 mbsl (and -53.3 and -63.3 mbsl with the GIA correction). At this period, the lake level in the Sea of Marmara was down to -90 m. Both of the lakes are shown to have discharged into the Marmara during MIS 3 and 2. Gemlik lake is shown to have discharged into the Imraeli Basin, located east of the Imrali Island. The lacustrine unit is indicated to have deposited on the acoustic basement of the lakes, this is supported by the cores from the work of Gasperini et al. (2011), Taviani et al. (2014) and Filicki et al. (2017).

Reviewer: 2. Methods are poorly described. In some aspects, a better transparency regarding the methods, which dataset is new and which ones are from other sources would benefit to the paper:

Response to reviewer:

We tried our best to make the clarification but will do even better for the revised manuscript. We will elaborate specifically on the seismic profile of the Lake Gemlik.

Reviewer: 2.1. The authors write they use a GIA model to provide highstand thresholds on RSL in the Black Sea and Marmara Sea (Lines 69-71). In the paper the time period, etc. There is not even a figure showing these results for the Black Sea – Marmara Sea area during MIS 3. The authors simply acknowledge a researcher for generating GIA corrections (line 544). Did the author actually run a GIA model and present these results in the paper?

Response to reviewer:

So this was a mistake on our part. We did use a GIA model for earlier version of the paper, the results of which were heavily criticized so tried to refocus our paper on the RSL results. This part was not altered / removed.

Reviewer: 2.2. In this paper, the only new geochemical dataset (although it is an impressive one) is the 87Sr/86Sr record from ostracods/shells from the Marmara Sea. However, the authors do not mention that few of these data were already previously published in Vidal et al. (2010). Also, the ostracods and mollusks used for strontium analyses come from the sediments of 4 different cores (ITU-C1, ITU-C10, MD01-2430, MD01- 2426). However the authors mention the age model for only one of these cores (core MD04-2430). What about ITU-C1, ITU-C10 and MD01-2426 age models?

Response to reviewer:

The age models for the other cores we tuned to the 87Sr/86Sr record of the core that was 14C dated with results from Vidal et al. (2010). We acknowledge this is an imperfect method but the best we could do under the given circumstances as the original 14C dates were misplaced.

Reviewer: 2.3. MD01-2430 age model is poorly detailed. So it is difficult to evaluate its quality. The authors (lines 164-166) write they composed their own age model from 14C measurements made from pieces of mollusks from MD01-2430 (Vidal et al. 2010) and it seems that they tuned MD01-2430 ostracod $\delta$18O data (from Vidal et al, 2010) to Black Sea mollusk ones (Yanchilina et al., 2017) and Sofular Cave speleothems (Fleitmann et al., 2009; Badertscher et al., 2011). Isn't it a bit circular, as Black Sea data have been already tuned to Sofular Cave speleothem (Yanchilina et al., 2017)? Furthermore, similar patterns between $\delta$18O and 87Sr/86Sr changes in the Marmara Sea and in the Black Sea are used to infer periods of connection between basins later in the paper. If they are tuned to each other, then the inferred periods of connection between basins are not a result per se anymore, it is premise.

To get around this problem, the data series shall be kept on independent age models (Blaauw, 2012). Please provide a detailed and transparent description of the age-depth models: 14C dates used (they seem to have been already published in Vidal et al. (2010) and Çagatay et al. (2015)), reservoir correction applied, software used, tuning, uncertainties. . .

Response to reviewer:

We decided to tune the age model of MDO01-2430 to Sofular Cave U/Th record for consistency with the Black Sea $\delta$18O. In Vidal et al. (2010), the authors correct their 14C dates with a water reservoir age of 400 years and then convert to calendar ages using marine04 data set with Calib 5.01 software. Vidal et al. (2010) go on to say that they used the 400 year correction as an assumption. Çaħatay et al. (2015) re-calibrate Vidal et al. (2010)'s 14C dates with INTCAL13 software. We believe our method by tuning to a well dated Sofular Cave is a better alternative to calibration to calendar ages. The assumption of 400 years is just an assumption, especially given that its been clearly shown that the 14C reservoir age changes significantly for lakes. Our method isn't perfect but using a constant 400 14C reservoir age with 400 years as an assumption isn't either. Our age model also does not affect our interpretation of MIS 3 lake levels.

Reviewer: 2.4. Also, the age model of core MD01-2430 has been already published in Vidal et al. (2010) based on mollusk 14C dates, and later revised/refined in Çagatay et al. (2015) using tephra markers and tuning to Greenland ice cores. Why did the authors need to redo this reliable age model? Also, as the MD01-2430 Ca% data are from Çagatay et al. (2015), are the MD01-2430 87Sr/86Sr data and Ca% data shown in figure 3 with different age-depth models whereas they come from the exact same core?

Response to reviewer:

The authors see the point and perhaps it is better to leave the MD01-2430 87Sr/86Sr

data with the original Vidal et al. (2010) age. We think our method by independently tuning to another reliable record was an upgrade on the previously published age model, that is independent of a variable 14C reservoir age as its been shown that the 14C reservoir age in the Black Sea, for instance, has varied up to 2000 14C years and is not a constant 400 14C years (Soulet et al., 2010).

Reviewer: 3. Misuse and over interpretation of chloride data 3.1. The authors mention pore water chloride data from the Black Sea (Soulet et al., 2010) and Marmara Sea (Aloisi et al., 2015) to say that the Black Sea and Marmara Sea were fresh during MIS 3 and MIS 4 (Lines 112-114). This is not exactly what the original papers state. According to Soulet et al. (2010) the Black Sea was fresh (âĹij1 psu) during the LGM. There is no mention for earlier periods. According to Aloisi et al. (2015) the Marmara Sea was not fresh but brackish (4psu) during MIS 3 and freshened at the end of MIS-4. The authors should stick to the conclusions of the original papers if they don't provide further material to modify/refine the original conclusions.

Response to reviewer:

The authors respectfully disagree. The data is published and it is for other researchers to use in order to build upon earlier work. This is exactly what we are doing. The authors don't have to stick to the original conclusions if we are building up on the original data by compiling other data sets that show, in our interpretation, that both lakes were fresh / brackish with lower salinity.

Aloisi et al. note in 6.1 that, "Since surface waters cannot be significantly saltier than bottom waters our results are consistent with those of Çaħatay et al. (2000), Çaħatay et al. (2009) who provide the lowest salinity estimate (S of about 3 to 7 ‰ for Marmara Sea surface waters during the last glacial from 75 cal kyr BP to the post-glacial reconnection with the Aegean Sea at 14.7 cal B.P."

Aloisi et al. also note in 6.3, contrary to what is suggested by the reviewer, that, "An earlier freshening, such as that modelled in the standard run, can allow for the return

to marine conditions for a few thousand years after the initial freshening at 75 cal kyr BP (data not shown). Nevertheless, even in this scenario, lacustrine conditions have to be persistent over at least 50 kyr before the 14.7 cal kyr BP marine ingression in order to model to reproduce the pore water Cl- profile." Hence, 50 kyr is given as the latest age the freshening could have started not the earliest and is consistent with our interpretation of the Marmara Sea porewater Cl-.

Reviewer: 3.2. In figure 3, Black Sea and Marmara Sea pore water chloride data are shown as a function of the sediment age which is nonsensical. Diffusion and advection processes continuously alter the geochemical pore water composition, and hence there is no obvious relationship between the age of the sediment and its pore water geochemical composition (for example Manheim and Chan, 1974; Adkins and Schrag, 2003; Soulet et al., 2010; Aloisi et al., 2015). Basically one cannot interpret pore water geochemical data directly as a function of the age of the sediment as unfortunately the authors do at lines 214-219. These data should be dropped from figure 3.

Response to reviewer:

The authors respectfully disagree. While yes, porewater Cl- profile will change over time but this is something that is understood to happen in the future as porewater Cl- is subject to diffusion and advection. But, at this moment, the porewater Cl- at a specific depth that is given a calendar age is porewater Cl- at that age of the current sediment (it would not have changed much between when the porewater Cl- was measured a few years ago to now). We are not making direct interpretations but building up on the work of Aloisi et al. (2015) and Soulet et al. (2010). Perhaps we need to make our interpretations more clear and that porewater Cl- at this age of the sediment is not what was porewater Cl- at the time the sediments accumulated. Instead it reflects the porewater Cl- at this depth and age of the sediment after the processes of advection and diffusion have altered the original porewater Cl-.

Reviewer: 4. The paper is mainly focused on MIS-3 and on the Marmara Sea. However, if one removes the pore water chloride data from figure 3 because of the reasons detailed just above, then only the MD01-2430 Ca% (Çagatay et al., 2015) extends back to MIS-3.

The remaining Marmara Sea geochemical data ($\delta$18O [Vidal et al., 2010] and 87Sr/86Sr [Vidal et al., 2010; this study]) only extend back to MIS-2. Can these data be used to support the conclusions for MIS-3? C4 CPD

The similarity between the Black Sea CaCO3% (Nowaczyk et al., 2012) and Marmara Sea Ca% (Cagatay et al., 2015) during MIS-3 is interpreted here without clear support as reflecting the connection between the Black Sea and Marmara Sea. The authors do not show on figure 3 that a Black Sea CaCO3% peak (for example Major et al., 2002; Bahr et al., 2005) correlates to the Marmara Sea Ca% during Bølling oscillation (see figure 6 in Çagatay et al., 2015) just before the Marmara Sea reconnection to the Mediterranean.

Also Yanchilina et al. (2017) suggest that the Black Sea was isolated from the Marmara Sea during the Bølling-Allerød. So, why would the Black Sea CaCO3% and Marmara Sea Ca% positive correlation suggest the connection of the two basins during MIS-3, but not during the Bølling oscillation?

Can this correlation be solely explained by a common climatic conditions as it has been shown that the Black Sea carbonate peaks are driven by surface productivity during warmer oscillations (Major et al., 2002; Bahr et al., 2005; Shumilovskikh et al., 2014; Wegwerth et al., 2016). Conclusive evidence would come from the geochemical measurements of the Black Sea and Marmara Sea carbonates deposited during these events as the authors suggest (L362-364). In the absence of such data, I would suggest the authors to be more balanced in their interpretations or to provide additional and stronger/conclusive support for a Black Sea – Marmara Sea connection during MIS-3.

Response to reviewer:

If one removes the porewater chloride data, which should not be done because of the reasons above, then in addition to the Marmara Ca% there is also the Black Sea CaCO3 from Nowaczyk et al. (2012) and also the Sofular Cave $\delta$18O from Badertscher et al. (2011) and Fleitmann et al. (2009).

We do think we should include the rest of the Black Sea CaCO3 record from other published data for completeness.

The reviewer correctly point out that Yanchilina et al. (2017) argue for a disconnection between the Black Sea and the Sea of Marmara during the Bolling Allerod. This actually strengthens our point. While the Marmara Sea Ca record is constant after its connection with the Mediterranean, the Black Sea Ca record (or interchangeably CaCO3) has a peak during early Preboreal. The two peaks behave differently after the Sea of Marmara has connected with the Mediterranean. We do agree that the reviewer does point out something we do agree with, as CaCO3 accumulation does correspond to fluctuations in climate, and both the Sea of Marmara and the Black Sea are in the same geographical region, both of the CaCO3 could be responses to climate and not connectivity. We should add more discussion to this point.

Reviewer: 6. Chronological interpretations of the seismic profiles are not supported by direct dating, and thus if I acknowledge that a sedimentary structure that is below another one should be older, in my view it is very difficult to be conclusive regarding their exact age.

Response to reviewer:

This is not exactly the case. In Supplementary Figure 4, there is a measured 14C date of the lacustrine sediments of 24.9 kyr 14C years. All the sediments that are outcropping to the left in the figure and below the erosion surface and the marine sediments (in red) are younger than 24.9 14C years. So the reviewer comment does not give accurate criticism of our work.

Reviewer: Specific comments:

L22: "connections and disconnections (partial or total)". It sounds odd, maybe remove "(partial or total)".

Response to reviewer:

Thank you for the comment; The authors would make an adjustment to this statement to make it more clear and perhaps remove, "partial or total," as the two lakes are either connected or they are not.

Reviewer: L24: "persistent freshwater lakes". The Marmara Sea salinity was reconstructed to be brackish (4psu; Aloisi et al., 2015).

Response to reviewer:

We agree with this and should make it clear what we mean by lakes and instead refer to the Sea of Marmara and the Black Sea as brackish lakes with low salinity as by definition, freshwater is that water that has 0 salinity (National Atmospheric and Oceanic Administration 2017).

Reviewer: L36: Remove "bi-polar", DO oscillations are North Hemispheric climatic features. The bipolar seesaw concept does not fit the context of the paragraph.

Response to reviewer:

This is a good suggestion and we would make this alteration in the revised manuscript.

Reviewer: L70: "using a GIA model": Please describe it in the methods along with the parameters you used. The information is currently missing.

Response to reviewer:

The authors used a GIA model to draw conclusions about ESL in previous versions of the manuscript but decided it was a better idea to stick to reporting RSL and just make educated conclusions about ESL given the currently understood distribution of Eurasian ice sheets during MIS 3. The authors need to remove this from the manuscript.

Reviewer: L77: "We show that the two lakes were freshwater". I am afraid that the authors do not provide original data showing that both lakes were fresh. Instead they are building up on many previous works. Please amend this sentence.

Response to reviewer:

This is a good suggestion and in the revised version of the manuscript we will make this change.

Reviewer: L87: These studies do not provide "accumulation". They only provide contents.

Response to reviewer:

We agree with this and will make this change in the revised manuscript.

Reviewer: L91: Vidal et al. (2010) published $\delta$18O from ostracods not from "mollusks".

Response to reviewer:

This is correct and we will make the change in the revised manuscript.

Reviewer: L92-103: The whole paragraph lacks support from the literature. Please cite for example Leng and Marshall (2004) for the mechanism and for example Major et al. (2002), Bahr et al. (2005), Shumilovskikh et al. (2014), Wegwerth et al. (2016) for its observation in the Black Sea.

Response to reviewer:

This is a great suggestion and we do indeed need to support this paragraph with the above citations. We will make sure to make these changes in the revised manuscript.

Reviewer: L97: The Black Sea outflowing into the "small" Marmara Sea as an evidence for explaining the temporal synchronicity is unsupported material. Alternatively the

regional climatic context could explain the temporal synchronicity.

Response to reviewer:

The authors point the reviewer to the discussion above regarding CaCO3 and connection/ disconnection between the Marmara Sea and the Black Sea.

Reviewer: L104-106: The way it is written is misleading. In 2010, Soulet et al. tested the age of the reconnection between 8500 and 9500 based on the available literature (Major et al., 2006) at this time. So the Yanchilina et al (2017) reference for the age of the reconnection should be dropped in this context. Similarly, Aloisi et al set the Marmara Sea reconnection to 14.7 ka based on Vidal et al. (2010) results. Furthermore the authors speak about pore water measurements in the Black Sea (Soulet et al., 2010) and in the Marmara Sea (Aloisi et al., 2015) without citing the original works. L107: This statement does not reflect the reality. 1) Pore water chloride content of the sediment does not reflect a paleo-salinity, as advection-diffusion processes alter both the original concentrations and the age-depth relationship with the sediment. The pore water chloride content profile is modern, not fossil. 2) The advection/diffusion model is actually a quantitative (within the limitations of the model and scenaris tested) reconstruction of the paleo-salinity. The authors state, however the opposite.

Response to reviewer:

We agree that we should make sure to cite Soulet et al. (2010) in this and also clarify that this is the porewater Cl- now, not then, given advection and diffusion within porewater of salts. We completely agree with the fossil vs. modern and the wording should more clearly reflect what the authors want to convey.

Reviewer: L109: "ppt": Part per trillion? "psu" instead?

Response to reviewer:

Yes, psu is correct. We will make this adjustment in the revised manuscript.

Reviewer: L112-114: This is not exactly what the original publications says. Soulet et al. (2010) only tested salinity models not older than 20ka. So it cannot be inferred from this study that the Black Sea was fresh at times older than 20 ka. Instead other studies can be cited. Aloisi et al. (2015) tested salinity models spanning 130 ka, with a salinity decrease at the end of MIS-4, and the salinity value inferred for the Marmara Sea lake during MIS-3 is 4psu (brackish, not fresh).

Response to reviewer:

We agree with this comment and should be more careful in the interpretation of porewater Cl- from the Black Sea. We will adjust our discussion to reflect the work that was actually done.

Reviewer: L128: You may add Shumilovskikh et al. (2014).

Response to reviewer:

This is a great suggestion and we will add this citation in the revised manuscript.

Reviewer: L140: You may add Ankindinova et al. (2019).

Response to reviewer:

This is a great suggestion, we did not know of this paper before resubmitting the manuscript. We will add this citation in the revised manuscript.

Reviewer: L155-156: There is something odd. Are you sure you leached Sr to retrieve the Sr fraction?

Response to reviewer:

Yes, perhaps another word is, "extracted." The mollusks were dissolved in nitric acid following passing the residue through resin that separates out different elements.

Reviewer: L163-164: "Although the original 14C age measurements have been misplaced": Please clarify.

Response to reviewer:

The authors hope to clarify this in the revised version of the manuscript. The mollusk samples were 14C dated originally but this data was lost and was not able to be retrieved.

Reviewer: L169-171: Isn't it the purpose of the paper to infer when the Black Sea was overflowing into the Marmara Sea? From this sentence it seems it was a premise of your work.

Response to reviewer:

For the MIS 3 period but perhaps it is a better idea to give the Marmara Sea samples their own age model from either Vidal et al. (2010) or Çağatay et al. (2015) to avoid this somewhat circulatory methodology.

Reviewer: L171: Supplementary data 1 is missing so one cannot evaluate the quality of the age model and data.

Response to reviewer:

Supplementary data 1 should have been uploaded but by mistake, the authors didn't. The authors will make sure to submit the supplementary data in the revised manuscript.

Reviewer: L175: Literature exits for Gemlik lake: Gasperini et al., 2011; Taviani et al., 2014 ; Filikci et al., 2017

Response to reviewer:

Thank you for this information and we will include these data in the revised manuscript. We were not aware of these data and these publications.

Reviewer: L195: Same conclusions has been reached by Vidal et al. (2010).

Response to reviewer:

This is a correct observation and we should adjust our wording to reflect the primary

contribution of Vidal et al. (2010) to say that the Sea of Marmara must have outflowed to the Mediterranean Sea during this period.

Reviewer: L211-218: Naive or even nonsensical interpretations of pore water chloride profiles, as there is no obvious relationship between the age of the sediment and its pore water composition.

Response to reviewer:

The authors disagree per same comments as above. The porewater Cl- that were measured, were measured at a specific depth that does have an age to it. The authors should make it more clear that the porewater Cl- profiles are always changing, although more on a centennial / millennial scale.

Reviewer: L219: Marmara Sea $\delta 18O$ and Sr records do not extend back to MIS 3 and the Black Sea record is very scarce for this period so conclusions of this sentence are unsupported for the Marmara Sea and weakly supported for the Black Sea.

Response to reviewer:

That is why there is the Sofular Cave $\delta 18O$ record to give additional strong support.

Reviewer: L226-227: Direct proxies have been published for this period in the Black Sea (Shumilovskikh et al., 2014; Wegwerth et al., 2016).

Response to reviewer:

Including data from these works is a great idea and we will consider in the revised manuscript. The authors wanted to focus on geochemical data sets but all available data sets, geochemical and biological should be given full consideration.

Reviewer: L266-267: Chronological data are needed to infer such statement and Vardar et al. (2014) do not provide direct dating of these strata.

Response to reviewer:

The references that the reviewer suggested regarding Gemlik lake and Israeli ridge will provide the chronological framework for this discussion. We should also adjust our wording to give Vardar et al. (2014) proper citation to reflect what the authors said in the publication.

Reviewer: L276-277: "It is shown that the MIS-2 period corresponds to a large transgressive period following the MIS-3 low stand (ÇaÄet al. 2009)." Yet, Çagatay et al. (2009) suggest the opposite: a forced-regression during MIS-2 to -85mbsl that followed a MIS-3 Marmara Sea level at 70mbsl.

Response to reviewer:

The authors are not sure which part of the Çaħatay et al. (2009) study the reviewer refers to but in the abstract, Çaħatay et al. (2009) clearly say, "Ancient shorelines are pervasive at -85 m on the northern shelf and in the region of Prince Islands coincident with the elevation of the modern bedrock sill in the Canakkale (Dardanelles) Strait. At times when global (eustatic) sea level dropped below the sill, the surface of the SoM stabilized at its outlet and freshened. Thus this particular shoreline is interpreted as the edge of the most recent SoM lake that existed from about 75 ka bp to 12 ka bp."

Reviewer: L306: These data do not extend back to MIS-3.

Response to reviewer:

This is a correct observation and we need to adjust this to instead say, "beginning of MIS 2."

Reviewer: L284-286: I don't follow the authors. They write that core data are unavailable to strengthen the interpretation but "hence. . .".

Response to reviewer:

This discussion needs to be clarified. Our description of the results from the seismic profile of Lake Gemlik are to show that there haven't been marine deposits during MIS

2-3-4 and that the lake has only lake deposits during this period.

Reviewer: L326-327: Where are these GIA results? Which results are the author referring to? No method has been described.

Response to reviewer:

As mentioned previously, the earlier version of the manuscript did have the GIA model and the results but the authors decided to eliminate that and instead give much more focus towards RSL data and interpretations.

Review References:

Adkins, J. F., & Schrag, D. P. (2003). Reconstructing Last Glacial Maximum bottom water salinities from deep-sea sediment pore fluid profiles. Earth and Planetary Science Letters, 216(1-2), 109-123.

Aloisi, G., Soulet, G., Henry, P., Wallmann, K., Sauvestre, R., Vallet-Coulomb, C., ... & Bard, E. (2015). Freshening of the Marmara Sea prior to its post-glacial reconnection to the Mediterranean Sea. Earth and Planetary Science Letters, 413, 176-185.

Ankindinova, O., Hiscott, R. N., Aksu, A. E., & Grimes, V. (2019). High-resolution Sr-isotopic evolution of Black Sea water during the Holocene: Implications for reconnection with the global ocean. Marine Geology, 407, 213-228.

Badertscher, S., Fleitmann, D., Cheng, H., Edwards, R. L., Göktürk, O. M., Zumbühl, A., ... & Tüysüz, O. (2011). Pleistocene water intrusions from the Mediterranean and Caspian seas into the Black Sea. Nature Geoscience, 4(4), 236.

Bahr, A., Lamy, F., Arz, H., Kuhlmann, H., & Wefer, G. (2005). Late glacial to Holocene climate and sedimentation history in the NW Black Sea. Marine Geology, 214(4), 309-322. Blaauw, M. (2012). Out of tune: the dangers of aligning proxy archives. Quaternary Science Reviews, 36, 38-49.

ÇaÄ, M. N., Eri Âÿs, K., Ryan, W. B. F., Sancar, Ü., Polonia, A., Akçer, S., ... & Bard,

E. (2009). Late Pleistocene–Holocene evolution of the northern shelf of the Sea of Marmara. Marine Geology, 265(3-4), 87-100.

ÇaÄ, M. N., Wulf, S., Sancar, Ü., Özmaral, A., Vidal, L., Henry, P., ... & Gasperini, L. (2015). The tephra record from the Sea of Marmara for the last ca. 70 ka and its palaeoceanographic implications. Marine Geology, 361, 96-110.

Filikci, B., Eri Âÿs, K. K., ÇaÄ, N., Sabuncu, A., & Polonia, A. (2017). Late glacial to Holocene water level and climate changes in the Gulf of Gemlik, Sea of Marmara: evidence from multi-proxy data. Geo-Marine Letters, 37(5), 501-513.

Fleitmann, D., Cheng, H., Badertscher, S., Edwards, R. L., Mudelsee, M., Göktürk, O. M., ... & Kramers, J. (2009). Timing and climatic impact of Greenland interstadials recorded in stalagmites from northern Turkey. Geophysical Research Letters, 36(19).

Gasperini, L., Polonia, A., ÇaÄ, M. N., Bortoluzzi, G., & Ferrante, V. (2011). Geological slip rates along the North Anatolian Fault in the Marmara region. Tectonics, 30(6).

Krijgsman, W., Tesakov, A., Yanina, T., Lazarev, S., Danukalova, G., Van Baak, C. G., ... & Bruch, A. (2018). Quaternary time scales for the Pontocaspian domain: C9 CPD Interactive comment Printer-friendly version Discussion paper Interbasinal connectivity and faunal evolution. Earth-science reviews.

Leng, M. J., & Marshall, J. D. (2004). Palaeoclimate interpretation of stable isotope data from lake sediment archives. Quaternary Science Reviews, 23(7-8), 811-831.

Major, C. O., Goldstein, S. L., Ryan, W. B., Lericolais, G., Piotrowski, A. M., & Hajdas, I. (2006). The co-evolution of Black Sea level and composition through the last deglaciation and its paleoclimatic significance. Quaternary Science Reviews, 25(17-18), 2031-2047.

Major, C., Ryan, W., Lericolais, G., & Hajdas, I. (2002). Constraints on Black Sea outflow to the Sea of Marmara during the last glacial–interglacial transition. Marine Geology, 190(1-2), 19-34.

Manheim, F. T., & Chan, K. M. (1974). Interstitial Waters of Black Sea Sediments: New Data and Review: Water.

Nowaczyk, N. R., Arz, H. W., Frank, U., Kind, J., & Plessen, B. (2012). Dynamics of the Laschamp geomagnetic excursion from Black Sea sediments. Earth and Planetary Science Letters, 351, 54-69.

Schumilovskikh, L. S., Fleitmann, D., Nowaczyk, N. R., Behling, H., Marret, F., Wegwerth, A., & Arz, H. W. (2014). Orbital and millenial-scale environmental changes between 64 and 25 ka BP recorded in Black Sea sediments. Climate of the Past, 10, 939-945.

Soulet, G., Delaygue, G., Vallet-Coulomb, C., Böttcher, M. E., Sonzogni, C., Lericolais, G., & Bard, E. (2010). Glacial hydrologic conditions in the Black Sea reconstructed using geochemical pore water profiles. Earth and Planetary Science Letters, 296(1-2), 57-66.

Taviani, M., Angeletti, L., ÇaÄ, M. N., Gasperini, L., Polonia, A., & Wesselingh, F. P. (2014). Sedimentary and faunal signatures of the post-glacial marine drowning of the Pontocaspian Gemlik "lake"(Sea of Marmara). Quaternary international, 345, 11-17.

Vardar, D., Öztürk, K., YaltÄsrak, C., Alpar, B., & Tur, H. (2014). Late Pleistocene–' Holocene evolution of the southern Marmara shelf and sub-basins: middle strand of the North Anatolian fault, southern Marmara Sea, Turkey. Marine Geophysical Research, 35(1), 69-85.

Vidal, L., Menot, G., Joly, C., Bruneton, H., Rostek, F., ÇaÄ, M. N., ... & Bard, E. (2010). Hydrology in the Sea of Marmara during the last 23 ka: Implications for timing of Black Sea connections and sapropel deposition. Paleoceanography, 25(1).

Wegwerth, A., Kaiser, J., Dellwig, O., Shumilovskikh, L. S., Nowaczyk, N. R., & Arz, H. W. (2016). Northern hemisphere climate control on the environmental dynamics in the glacial Black Sea "Lake". Quaternary Science Reviews, 135, 41-53.

Yanchilina, A. G., Ryan, W. B., McManus, J. F., Dimitrov, P., Dimitrov, D., Slavova, K., & Filipova-Marinova, M. (2017). Compilation of geophysical, geochronological, and geochemical evidence indicates a rapid Mediterranean-derived submergence of the Black Sea's shelf and subsequent substantial salinification in the early Holocene. Marine Geology, 383, 14-34.

Response References

Aloisi, G., G. Soulet, P. Henry, K. Wallmann, R. Sauvestre, C. Vallet-Coulomb, C. Lécuyer and E. Bard: Freshening of the Marmara Sea prior to its post-glacial reconnection to the Mediterranean Sea, Earth and Planetary Science Letters, 413, 176-85, 10.1016/j.epsl.2014.12.052, 2015. Ankindinova, O., R. N. Hiscott, A. E. Aksu and V. Grimes: High-resolution Sr-isotopic evolution of Black Sea water during the Holocene: Implications for reconnection with the global ocean, Marine Geology, 207, 213-28, 2019. Badertscher, S., D. Fleitmann, H. Cheng, R. L. Edwards, O. M. Göktürk, A. Zumbühl, M. Leuenberger and O. Tüysüz: Pleistocene water intrusions from the Mediterranean and Caspian seas into the Black Sea, Nature Geoscience, 4, 236-39, 10.1038/NGEO1106, 2011. Çağatay, M. N., K. Eriş, W. B. F. Ryan, Ü. Sancar, A. Polonia, S. Akçer, D. Biltekin, L. Gasperini, N. Görür, G. Lericolais and E. Bard: Late Pleistocene-Holocene evolution of the northern shelf of the Sea of Marmara, Marine Geology, 265, 87-100, 10.1016/j.margeo.2009.06.011, 2009. Çağatay, M. N., N. Görür, O. Algan, C. Eastoe, A. Tchapalyga, D. Ongan, T. Kuhn and I. Kuşcu: Late glacial – Holocene palaeoceanography of the Sea of Marmara: Timing of connections with the Mediterranean and the Black seas, Marine Geology, 167, 191-206, doi:10.1016/S0025-3227(00)00031-1, 2000. Çağatay, M. N., S. Wulf, Ü. Sancar, A. Özmaral, L. Vidal, P. Henry, O. Appelt and L. Gasperini: The tephra record from the Sea of Marmara for the last ca. 70 ka and its paleoceanographic implications, Marine Geology, 361, 96-100, 10.1016/j.margeo.2015.01.005, 2015. Filicki, B., K. Kadir Eris, N. Cahatay, A. ASabuncu and A. Polonia: Late glacial to Holocene water level and climate changes in the Gulf of Gemlik, Sea of Marmara:

evidence from multi-proxy data, Geo-Mar Letters, 37, 501-13, 2017. Fleitmann, D., H. Cheng, S. Badertscher, R. L. Edwards, M. Mudelsee, O. M. Göktürk, A. Fankhauser, R. Pickering, C. C. Raible, A. Matter, J. Kramers and O. Tüysüz: Timing and climatic impact of Greenland interstadials recorded in stalagmites from northern Turkey, Geophysical Research Letters, 36, 10.1029/2009GL040050, 2009. Gasperini, L., A. Polonia, M. N. Cagatau, G. Boruluzzi and V. Ferranted: Geological slip rates along the North Anatolian Fault in the Marmara region, Tectonics, 30, 2011. National Oceaniv and Atmospheric Administration: Salinity and Total Dissolved Salts, 2017. Nowaczyk, N. R., H. W. Arz, U. Frank, J. Kind and B. Plessen: Dynamics of the Laschamp geomagnetic excursion from Black Sea sediments, Earth and Planetary Science Letters, 351-352, 54-69, 10.1016/j.epsl.2012.06.050, 2012. Shumilovskikh, L. S., D. Fleitmann, N. R. Nowaczyk, H. Behling, F. Merret, A. Wegwerth and H. W. Arz: Orbital- and millenial-scale environmental changes between 64 and 20 ka BP recorded in Black Sea sediments, Climate of the past, 10, 939-54, 2014. Soulet, G., G. Delaygue, C. Vallet-Coulomb, M. E. Böttcher, C. Sonzogni, G. Lericolais and E. Bard: Glacial hydrologic conditions in the Black Sea reconstructed using geochemical pore water profiles, Earth and Planetary Science Letters, 296, 57-66, 10.1016/j.epsl.2010.04.045, 2010. Taviani, M., L. Angeletti, M. N. Cagatay, L. Gasperini, A. Polonia and F. P. Wesselingh: Sedimentary and faunal signatures of the post-glacial marine drowning of the Pontocaspian Gemlik "lake" (Sea of Marmara), Quaternary International, 345, 11-17, 2014. Vardar, D., K. Öztürk, C. Yaltirak, B. Alpar and H. Tur: Late Pleistocene-Holocene evolution of the southern Marmara shelf and sub-basins: middle strand of the North Anatolian fault, southern Marmara Sea, Turkey, Marine Geophysical Research, 35, 69-85, 2014. Vidal, L., G. Ménot, C. Joly, H. Bruneton, F. Rostek, M. N. Çağatay, C. Major and E. Bard: Hydrology in the Sea of Marmara during the last 23 ka: Implications for timing of Black Sea connections and sapropel deposition, Paleoceanography, 25, 10.1029/2009PA001735, 2010. Wegwerth, A., J. Kaiser, O. Dellwig, L. S. Shumilovskikh, N. R. Nowaczyk and H. W. Arz: Northern hemisphere climate control on the environmentak dynamics in the

glacial Black Sea "Lake", Quaternary Science Reviews, 135, 41-53, 2016. Yanchilina, A. G., W. B. F. Ryan, J. F. McManus, P. Dimitrov, D. Dimitrov, K. Slavova and M. Filipova-Marinova: Compilation of geophysical, geochronological, and geochemical evidence indicates a rapid Mediterranean-derived submergence of the Black Sea's shelf and subsequent substantial salinification in the early Holocene, Marine Geology, 383, 14-34, 10.1016/j.margo.2016.11.001, 2017.

Please also note the supplement to this comment:
https://www.clim-past-discuss.net/cp-2019-30/cp-2019-30-AC5-supplement.pdf